# Beyond Naïve Prompting: Strategies for Improved Context-aided Forecasting with LLMs

**Arjun Ashok**                                                                *arjun.ashok.psg@gmail.com*
*ServiceNow Research, Mila - Québec AI Institute, Université de Montréal*

**Andrew R. Williams**                                                         *andrew.williams@umontreal.ca*
*ServiceNow Research, Mila - Québec AI Institute, Université de Montréal*

**Vincent Zhihao Zheng**                                                       *z.vincent.zheng@gmail.com*
*ServiceNow Research, McGill University*

**Irina Rish**                                                                 *irina.rish@umontreal.ca*
*Mila - Québec AI Institute, Université de Montréal*

**Nicolas Chapados**                                                           *nicolas.chapados@mila.quebec*
*ServiceNow Research, Mila - Québec AI Institute, Polytechnique Montréal*

**Étienne Marcotte**                                                           *etienne.marcotte@gmail.com*
*ServiceNow Research*

**Valentina Zantedeschi**                                                      *valentina.zantedeschi@servicenow.com*
*ServiceNow Research, Université Laval*

**Alexandre Drouin**                                                           *alexandre.drouin@servicenow.com*
*ServiceNow Research, Mila - Québec AI Institute, Université Laval*

**Reviewed on OpenReview:** *https://openreview.net/forum?id=dkjHHFJkVI*

## Abstract

Real-world forecasting requires models to integrate not only historical data but also relevant contextual information provided in textual form. While large language models (LLMs) show promise for context-aided forecasting, critical challenges remain: we lack diagnostic tools to understand failure modes, performance remains far below their potential, and high computational costs limit practical deployment. We introduce a unified framework of four strategies that address these limitations along three orthogonal dimensions: model diagnostics, accuracy, and efficiency. Through extensive evaluation across model families from small open-source models to frontier models including Gemini, GPT, and Claude, we uncover both fundamental insights and practical solutions. Our findings span three key dimensions: diagnostic strategies reveal the "Execution Gap" where models correctly explain how context affects forecasts but fail to apply this reasoning; accuracy-focused strategies achieve substantial performance improvements of 25-50%; and efficiency-oriented approaches show that adaptive routing between small and large models can approach large model accuracy on average while significantly reducing inference costs. These orthogonal strategies can be flexibly integrated based on deployment constraints, providing practitioners with a comprehensive toolkit for practical LLM-based context-aided forecasting. Code is made available at https://github.com/ashok-arjun/beyond-naive-prompting.

# 1  Introduction

Probabilistic time series forecasting is essential for optimal decision-making, involving predicting the evolution of various quantities over time, as well as estimating the likelihood of various scenarios (Hyndman & Athanasopoulos, 2021; Peterson, 2017). This problem has been extensively studied by both the statistical and machine learning communities (Hyndman et al., 2008; Box et al., 2015; Hyndman & Athanasopoulos, 2021), culminating in different methods such as classical methods (Hyndman et al., 2008; Gardner Jr., 1985), deep learning methods (Salinas et al., 2020; Drouin et al., 2022; Ashok et al., 2024), hybrid methods (Oreshkin et al., 2019), and more recently, foundation models (Rasul et al., 2023; Ansari et al., 2024; Woo et al., 2024). Research in forecasting has largely focused on building models that use nu-

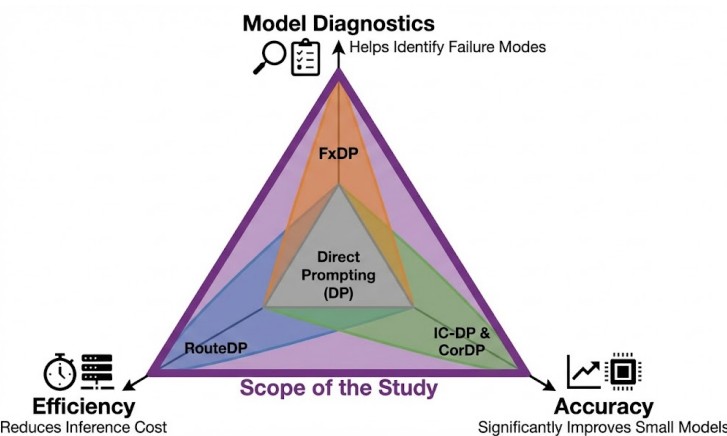

Figure 1: Scope of our study. We propose four complementary strategies that extend naïve Direct Prompting (DP) (Williams et al., 2025) along different dimensions. FxDP (top) enables model diagnostics by eliciting explanations about how context affects forecasts, RouteDP (bottom-left) reduces inference costs through adaptive model routing, and IC-DP and CorDP (bottom-right) substantially improve forecasting accuracy, especially for smaller models.

merical historical observations and engineered covariates, while in the real-world, accurate forecasts rely not only on them but also on contextual information about the problem or task in hand (Hyndman & Athanasopoulos, 2021). With the realistic assumption that such prior information can be expressed flexibly in natural language, a new, multimodal problem setting of *context-aided forecasting* has recently emerged in the literature (Jin et al., 2024; Liu et al., 2024a;c; Kong et al., 2025).

Several methods have been proposed for context-aided forecasting, and can be broadly classified into two types (Zhang et al., 2025): those that rely on training models on specific context-aided forecasting tasks (Jin et al., 2024; Zhang et al., 2023; Xu et al., 2024; Emami et al., 2024; Wang et al., 2024; Liu et al., 2024a; Zhou et al., 2025) and those that do not require training, and purely leverage the capabilities of LLMs for context-aided forecasting (Merrill et al., 2024; Gruver et al., 2024; Requeima et al., 2024; Williams et al., 2025). Among those that use LLMs, only simple strategies have been explored, such as direct prompting (Williams et al., 2025) and autoregressive LLM processes (Requeima et al., 2024) among others. These methods involve simply feeding historical numerical data and textual context into the LLM and generating forecasts timestep-by-timestep. Yet these simple methods provide no mechanism to diagnose failures and offer limited accuracy and efficiency improvements.

In this work, we systematically investigate 4 strategies that address and improve different aspects of forecasting with LLMs (illustrated in Figure 1):

- **FxDP: Direct Prompting with Forecast Effect Explanation** (Section 4) elicits explanations of how context affects forecasts before predictions, enabling diagnosis of failure modes. Evaluating explanation accuracy against gold standards reveals the "Execution Gap": models often explain correctly but fail to apply their reasoning, with this pattern persisting even for frontier models.

- **CorDP: Direct Prompting for Forecast Correction** (Section 5) leverages LLMs to refine existing probabilistic forecasts with context rather than forecasting from scratch. This improves forecast accuracy by up to 50% while enabling practical integration into existing forecasting workflows.

- **IC-DP: In-Context Direct Prompting** (Section 6) embeds historical examples of context-aided forecasting tasks in the prompt. This substantially improves accuracy across all model scales, reducing forecast error by over 25% even for frontier models.

- **RouteDP: Direct Prompting with Model Routing** (Section 7) enables accurate forecasting under resource constraints by using a small model for easy tasks and delegating more difficult ones to a larger model, guided by a router. We observe substantial improvements in forecast accuracy (up to 46%) at a fraction of the cost.

These strategies target three orthogonal dimensions as shown in Figure 1: FxDP provides model diagnostics, CorDP and IC-DP enhance accuracy, and RouteDP improves efficiency. We show that these strategies are complementary and can be deployed individually or in combination based on specific constraints and objectives (Section 8). We evaluate these strategies on diverse context-aided forecasting tasks from the Context-Is-Key (CiK) benchmark (Williams et al., 2025). Across models from the Qwen (Yang et al., 2024) and Llama (Grattafiori et al., 2024) families, as well as frontier models including Gemini (Gemini Team, 2024), GPT (OpenAI, 2024), and Claude (Anthropic, 2024), we demonstrate that these strategies consistently improve over naive direct prompting. Our findings reveal both the significant potential and fundamental limitations of LLMs for context-aided forecasting across model scales.

## 2 Related Work

### 2.1 Large Models for Forecasting

Classical methods such as ETS, ARIMA, and their ensembles have long been central to time series forecasting (Hyndman et al., 2008; Box et al., 2015). Deep learning brought RNN and LSTM based models (Hewamalage et al., 2021; Salinas et al., 2020), then transformer-based methods (Lim et al., 2021; Wu et al., 2021; Zhou et al., 2021; Drouin et al., 2022; Wen et al., 2023; Nie et al.; Ashok et al., 2024). Following the success of pretrained language models (Brown et al., 2020), foundation models for forecasting (Rasul et al., 2023; Goswami et al., 2024; Woo et al., 2024; Ansari et al., 2024) have been proposed; they achieve strong zero-shot performance on unseen datasets, often outperforming dataset-specific models. Research on their capabilities (Potosnak et al., 2024) and limitations (Liang et al., 2024) continues to grow. A separate line of work uses large language models (LLMs) for forecasting. Gruver et al. (2024) propose LLMTime, the first work to show that LLMs can zero-shot extrapolate time series autoregressively; their then-surprising finding that LLMs match or exceed purpose-built forecasting models motivated a growing body of follow-up work, which we describe below. Requeima et al. (2024) propose LLM Processes, showing that prompt formatting and time series scaling both matter. (Liu et al., 2024b) propose LSTPrompt, using chain-of-thought to improve LLM performance on quantitative forecasting. Recent work continues to explore the value and limitations of LLMs for forecasting (Tang et al., 2025). Our work also aims to improve LLM based forecasting, but we focus on a setting where textual context is necessary to succeed: context-aided forecasting (Williams et al., 2025), which poses different challenges and requires different capabilities, which we study with our methods.

### 2.2 Context-aided Forecasting Methods

LLMs can condition on complementary side-information in text for forecasting (Jin et al., 2024; Liu et al., 2024c; Wang et al., 2024; Xu et al., 2024; Liu et al., 2024a). Jin et al. (2024) propose Time-LLM, using an LLM to encode dataset-level metadata and a transformer for the time series, trained jointly. Xu et al. (2024); Liu et al. (2024a) propose similar multimodal architectures for dataset-specific forecasting with per-window text. Liu et al. (2024c) extend to multi-dataset training with objectives that use textual metadata while avoiding "domain confusion". Wang et al. (2025) combine a pretrained time series foundation model with an LLM and train only adapters between them. Wang et al. (2024) fine-tune LLMs (e.g. Qwen-7B) on dataset-specific context-aided forecasting, showing the value of pure-LLM approaches. All of these methods involve training, specializing models to their training data (Zhang et al., 2025). Gruver et al. (2024); Requeima et al. (2024); Williams et al. (2025); Merrill et al. (2024) explore forecasting across diverse contexts and time series, where the focus in this setting is on how well models can use unambiguous, relevant context to succeed in forecasting scenarios, instead of on specializing models to specific scenarios (Wang et al., 2025). Merrill et al. (2024) evaluate LLMs on GPT-4-generated context-aided tasks and find a large human–LLM performance gap. Williams et al. (2025) propose a benchmark of 71 context-aided forecasting tasks across 8 domains, each requiring use of textual context to succeed, and evaluate LLMs with LLMP (Requeima et al., 2024) and a simpler method, Direct Prompt (DP), with promising results. Our work builds on this,

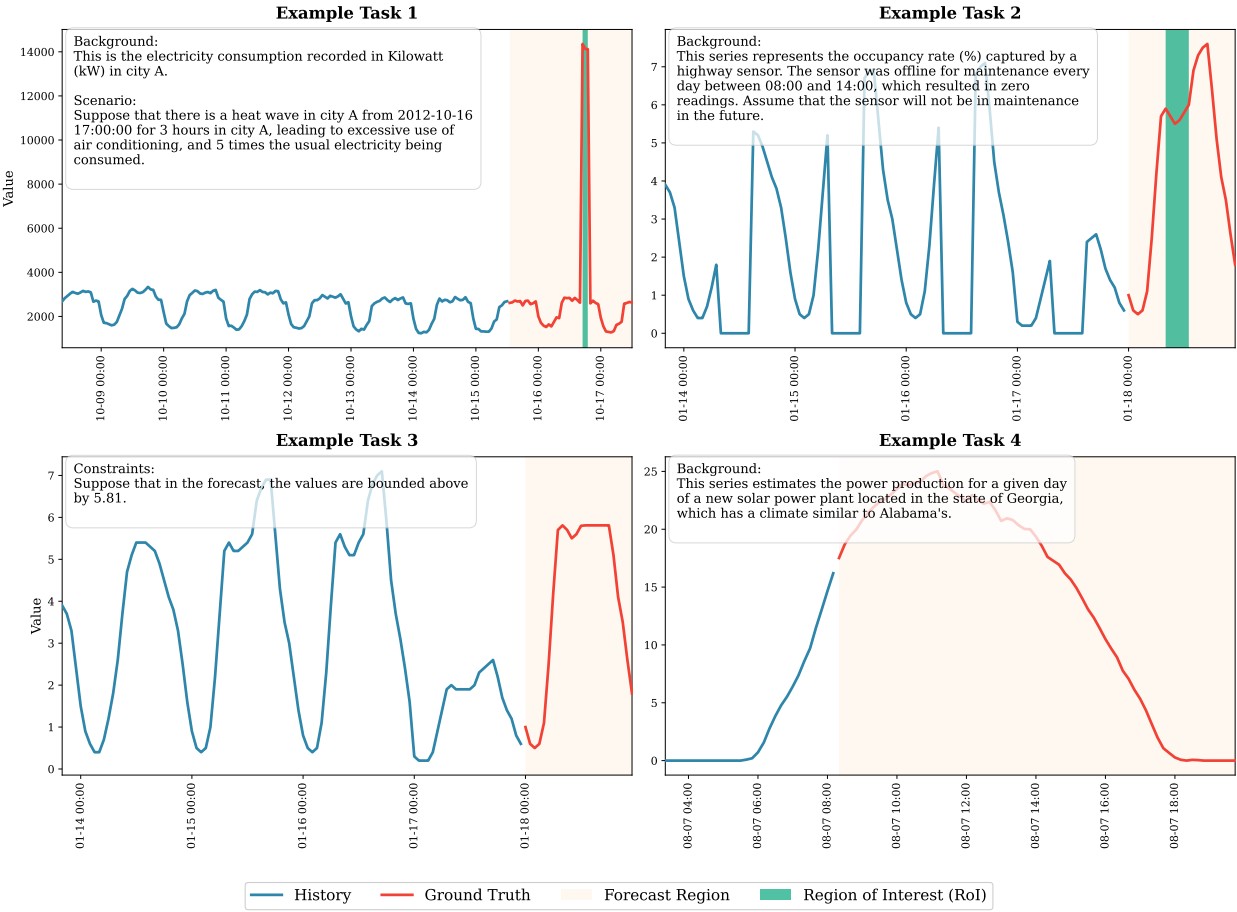

Figure 2: Examples of context-aided forecasting tasks from the Context-is-Key (CiK) benchmark (Williams et al., 2025). CiK comprises 2,644 time series across 7 real-world domains/datasets; it is designed to benchmark context-aided forecasting models, with tasks where the textual context is necessary for accurate forecasts.

going beyond naïve direct prompting (Williams et al., 2025) to explore variants that offer complementary advantages, reveal insights into model capabilities, and significantly improve performance.

## 3 Setup and Methodology

### 3.1 Problem Setting

The goal of context-aided forecasting is to produce statistical forecasts by incorporating relevant side information (i.e., context) (Williams et al., 2025; Wang et al., 2024; Kong et al., 2025; Zhang et al., 2025). We focus on the case where context is available in textual form. Formally, let $\mathbf{X}_H = [X_1, \ldots, X_t]$ denote a sequence of random variables representing historical observations at discrete time steps, with each $X_\tau \in \mathcal{X} \subseteq \mathbb{R}$. The future observations are denoted by $\mathbf{X}_F = [X_{t+1}, \ldots, X_T]$. The textual *context*, $\mathbf{C}$, provides additional information pertinent to forecasting $\mathbf{X}_F$, supplementing the information contained in $\mathbf{X}_H$. The forecasting task is thus to estimate the conditional distribution $P(\mathbf{X}_F \mid \mathbf{X}_H, \mathbf{C})$.

### 3.2 Direct Prompt

Williams et al. (2025) introduce Direct Prompt (DP), a method that instructs an LLM to generate forecasts as structured output for all required timestamps, given historical data and textual context. DP has shown that instruction-tuned LLMs can leverage context to improve forecasts on multiple benchmarks (Williams et al., 2025; Kupferschmidt et al., 2024), providing advantages over quantitative methods that cannot use

textual information (Kong et al., 2025; Zhang et al., 2025). Importantly, DP elicits accurate probabilistic forecasts without the computational overhead of digit-by-digit autoregressive generation (Gruver et al., 2024; Requeima et al., 2024).

### 3.3 Experimental Protocol

Following prior work (Zhang et al., 2025), we use the Context-Is-Key (CiK) benchmark (Williams et al., 2025) to evaluate zero-shot forecasting methods. CiK contains 71 manually designed context-aided forecasting tasks from 2,644 time series across 7 real-world domains: Climatology (Sengupta et al., 2018), Economics (U.S. Bureau of Labor Statistics, 2024), Energy (Godahewa et al., 2021), Mechanics (Gamella et al., 2024), Public Safety (Ville de Montréal, 2020), Transportation (Chen et al., 2001), and Retail (Godahewa et al., 2021). The tasks span sampling frequencies from 10 minutes to monthly intervals (see Figure 2 for examples). Crucially, CiK is the only benchmark where accurate forecasts cannot be achieved without incorporating the context, making it uniquely suitable for evaluating context-aided forecasting capabilities (Zhang et al., 2025). This distinguishes it from other benchmarks (Merrill et al., 2024; Liu et al., 2024a; Wang et al., 2024; 2025) where context may not be always essential for high-quality forecasts (Zhang et al., 2025). Additionally, the tasks in CiK are designed to mitigate memorization effects, making it suitable for evaluating LLMs.

We use the Region-of-interest CRPS (RCRPS) metric to evaluate context-aided forecasting performance (Williams et al., 2025), which prioritizes context-sensitive windows called the region of interest (RoI) and accounts for any hard constraints mentioned in the context (e.g., values cannot be negative). CRPS (Gneiting & Raftery, 2007) measures the squared difference between the predicted cumulative distribution function (CDF) and the empirical CDF of the ground truth; lower values indicate better forecast calibration (full definition in Appendix A). We report average RCRPS across all tasks. Additional details on the metric are in Appendix A.

We experiment with instruction-tuned models spanning many orders of magnitude in size: Qwen-2.5-0.5B, 1.5B, 3B, 7B, 14B, 32B, 72B (Yang et al., 2024) and Llama-3.2-1B, 3B, Llama-3-8B, Llama-3.3-70B, Llama-3.1-405B (Grattafiori et al., 2024), ensuring consistency with prior work (Williams et al., 2025). We also evaluate frontier models available at the time of writing: GPT-5.2 (OpenAI, 2024), Gemini-2.5-Pro (Gemini Team, 2024), and Claude-Sonnet-4.5 (Anthropic, 2024), demonstrating effectiveness on state-of-the-art LLMs. To obtain probabilistic forecasts, we draw 25 samples from each model's output distribution. Implementation details of models are provided in Appendix G.

In the following sections, we introduce our four strategies: FxDP, CorDP, IC-DP, and RouteDP, demonstrating their respective improvements over direct prompting (DP) Code for all methods are provided in the attached supplementary material.

## 4 FxDP: Direct Prompting with Forecast Effect Explanation

### 4.1 The Need for Forecast Effect Explanations

Context-aided forecasting requires models to perform two sequential tasks: first, correctly reasoning how the context would affect the forecast, and second, producing an accurate quantitative forecast that applies said effects. Without FxDP, models perform both steps implicitly in a single forward pass, with contextual reasoning internalized and never surfaced. FxDP explicitly decouples these steps by requiring the model to first verbalize how the context affects the forecast, making intermediate failures diagnosable. Current evaluation approaches focus exclusively on final forecasting accuracy, treating the model as a black box. This limitation becomes particularly problematic when models fail: we cannot determine whether the failure stems from poor reasoning about the context's effect on the forecast or from an inability to apply the effect on the forecast. Such ambiguity hinders our ability to diagnose the limitations of models. By evaluating both the model's explanation of how context affects the forecast and its forecasting performance, we can disentangle these two capabilities and potentially develop targeted model improvements.

## 4.2 FxDP as a Diagnostic Tool

To evaluate the model's reasoning process, we modify Direct Prompt, instructing the LLM to first explain the effect the context would have on the forecast, followed by the numerical forecast itself (see Appendix C.1 for the prompt). We call this explanation the "Forecast Effect Explanation", and the method, Direct Prompt with Forecast Effect Explanation (FxDP). This approach builds on the chain-of-thought prompting literature (Wei et al., 2022), where reasoning traces have been used for improved evaluation (Lightman et al., 2023a) and model diagnosis (Fu et al., 2023). We adapt it to elicit only what we aim to evaluate: a concise explanation of how *context* would affect the *forecast*, not a detailed step-by-step reasoning trace of how the model produces the forecast. Producing a correct explanation may be easier than producing an accurate forecast, since the former is a structured verbal task while the latter requires numerical precision; the diagnostic thus separates verbal reasoning about context from the ability to apply it quantitatively.

Our goal is to diagnose which of three cases holds for a given model and task: (1) the model reasons about the forecast effect correctly and applies it successfully, (2) the model reasons about the forecast effect correctly but fails to apply it to the forecast (which we term the "Execution Gap"), or (3) the model fails to reason about the forecast effect correctly. To support this diagnosis, we build an evaluation protocol with three components.

**Ground Truth Forecast Effects.** We first augment the Context-is-Key (CiK) benchmark (Williams et al., 2025) with ground truth forecast effects that we write for each task. Each effect states what the forecast should reflect given the context. For example, for *Example Task 2* in Figure 2, the ground truth forecast effect is that "the forecast should treat maintenance periods (where historical values are 0) as uninformative for the series level and should extrapolate the underlying trend and seasonality to the prediction window". We design a human evaluation protocol to verify these effects, following prior work that conducts similar human studies on gold-standard explanations and reasoning traces (Lightman et al., 2023b; Lee & Hockenmaier, 2025): we recruit five independent evaluators per task via the Prolific platform, present each effect with its task context, and ask whether the explanation correctly and sufficiently describes how the context affects the forecast; we aggregate responses by majority vote and use attention checks to ensure annotation quality, following standard annotation practice with small expert panels and strict agreement thresholds (Snow et al., 2008; Jamison & Gurevych, 2015) (details in Appendix C.2). We obtain majority agreement across all tasks, achieving at least 4 out of 5 annotator agreement. These verified effects form the gold standard for evaluating model explanations.

**Forecast Effect Evaluation Protocol.** For each model and task, we determine whether key points from the gold standard appear in the model's explanation, rather than requiring exact phrasing. We use a panel of three LLM judges (GPT-5.2, Gemini 2.5 Pro, Claude Sonnet) to perform this comparison at scale. A common limitation of LLM judge evaluation is uncertainty about whether judge ratings align with human judgment; we address this by designing a human evaluation protocol on a subset of items, where five independent annotators per item perform the same comparison task as the judges, using majority vote aggregation and strict agreement thresholds following standard annotation practice (Snow et al., 2008; Jamison & Gurevych, 2015). We find substantial to almost perfect agreement between human evaluators and the LLM judges (91.7–93.3%, Cohen's $\kappa = 0.81$–$0.84$), with high inter-annotator and inter-judge agreement (see Appendix C.3), validating the reliability of our automated evaluation. This approach follows prior work that compares model reasoning traces to gold-standard or expert-labeled references to evaluate correctness (Lightman et al., 2023a; Jacovi et al., 2024). To support reproducibility and future research, we open-source all evaluation artifacts: the verified ground truth forecast effects, the forecast effect explanations generated by all evaluated models, and the corresponding judgments from all three LLM judges and the humans. The inference time and cost of FxDP, and the cost of reasoning correctness evaluation, are reported in Appendix H.2.

**Forecast Improvement Evaluation.** To complete the diagnosis, we must assess whether models that explain correctly also produce better forecasts when context is provided. We measure forecast improvement as the relative improvement in CRPS when context is provided versus when it is not, computed within the region of interest (RoI) - the portion of the forecast that context is expected to affect (e.g. the green shaded portion in *Example Task 1* in Figure 2). We report results using a 50% relative improvement threshold in

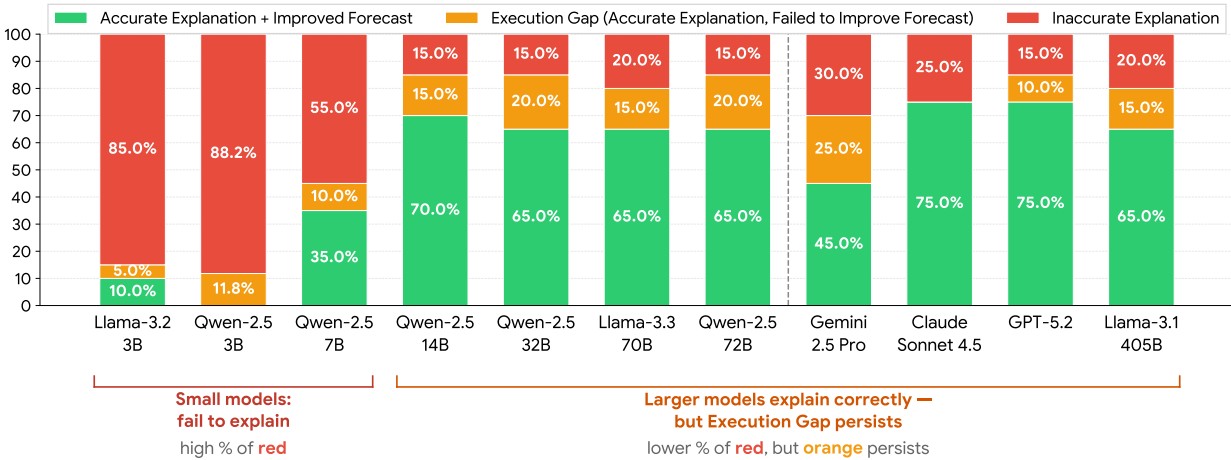

Figure 3: Forecast effect explanation accuracy and forecast improvement across models. Each bar shows the percentage of tasks falling into three categories: accurate explanation with improved forecast (green segment), accurate explanation but no forecast improvement (orange segment, the "Execution Gap"), and inaccurate explanation (red segment). Larger models can both reason about forecast effects correctly and apply them to improve forecasts, while smaller models often explain accurately but fail to translate this into improved forecasts. Results are with the panel of LLM judges; we find that the results are robust to the choice of LLM judge; extended results are in Appendix C.3.4.

the RoI as our primary criterion and provide analyses at additional thresholds (30%–80%) in Appendix C.5 to assess sensitivity to this choice.

These three components together enable the above diagnosis. By diagnosing whether failures stem from poor explanations or poor application, this approach provides actionable insights for targeted model improvements.

### 4.3 Results

**Explanation accuracy scales with model size.** The results in Figure 3 reveal clear patterns in how models process context-aided forecasting tasks across scales. Forecast effect explanation accuracy (in green) improves dramatically with model size: small models (3B) struggle to produce accurate explanations in most tasks, while larger models (14B+) consistently explain forecast effects correctly. A critical threshold emerges at 14B parameters, where models transition from predominantly failing to predominantly succeeding at explaining forecast effects. Mid-sized and large models (14B-72B) achieve strong performance, with the best models being Claude Sonnet 4.5 and GPT-5.2, reaching 75% full success (accurate explanation with improved forecast).

**The Execution Gap persists across model scales.** However, the Execution Gap, where models explain correctly but fail to improve forecasts, emerges as a persistent challenge. As shown in the middle segment (grey portion) of models in Figure 3, this gap affects even the largest models, indicating a fundamental difficulty in applying their reasoning to their forecasts. Notably, Gemini 2.5 Pro underperforms relative to its scale, achieving lower success rates than much smaller models like Qwen 2.5-14B and Llama 3.3-70B. In contrast, the smallest models (3B-7B) are primarily limited by their inability to generate accurate explanations in the first place. A key finding is that explanation accuracy scales more effectively than execution ability: while inaccurate explanations drop dramatically as models scale up, the Execution Gap persists even for frontier models. The robustness of this pattern across different improvement thresholds (30%–80%), detailed in Appendix C.5, confirms this is a fundamental challenge. No incorrect explanation leads to an improved forecast - confirmed by unanimous agreement between human annotators and LLM judges on all such cases (Appendix C.3.4).

The Execution Gap arises from the model's failure to apply its contextual reasoning to the forecast, not from errors in interpreting historical data. This can be inferred from the no-context results in Table 3: models given only historical data produce reasonable quantitative forecasts — frontier models (Claude-Sonnet-4.5:

| Model | Direct Prompt (DP) | Median Corrector (Median-CorDP) | | | SampleWise Corrector (SampleWise-CorDP) | | |
|---|---|---|---|---|---|---|---|
| | | Lag-Llama | Chronos Large | ARIMA | Lag-Llama | Chronos Large | ARIMA |
| Base Quantitative Forecaster | - | 0.382 ± 0.011 | 0.492 ± 0.004 | 0.636 ± 0.014 | 0.382 ± 0.011 | 0.492 ± 0.004 | 0.636 ± 0.014 |
| Llama3.2-1B-Inst | 0.396 ± 0.027 | **0.394 ± 0.004** | 0.515 ± 0.007 | 0.612 ± 0.018 | 0.541 ± 0.009 | 0.634 ± 0.005 | 0.672 ± 0.015 |
| Llama3.2-3B-Inst | 0.687 ± 0.025 | **0.344 ± 0.011** | 0.455 ± 0.009 | 0.573 ± 0.022 | 0.509 ± 0.026 | 0.423 ± 0.007 | 0.663 ± 0.031 |
| Llama3-8B-Inst | 0.543 ± 0.026 | **0.315 ± 0.004** | 0.453 ± 0.005 | 0.571 ± 0.004 | 0.426 ± 0.009 | 0.410 ± 0.004 | 0.636 ± 0.010 |
| Llama3.3-70B-Inst | 0.230 ± 0.006 | 0.281 ± 0.002 | 0.251 ± 0.004 | 0.352 ± 0.006 | 0.223 ± 0.004 | **0.215 ± 0.004** | 0.311 ± 0.007 |
| Llama3.1-405B-Inst | **0.173 ± 0.003** | 0.278 ± 0.009 | 0.226 ± 0.004 | 0.257 ± 0.008 | 0.199 ± 0.006 | 0.194 ± 0.004 | 0.229 ± 0.008 |
| Qwen2.5-0.5B-Inst | 0.592 ± 0.027 | 0.633 ± 0.002 | 0.801 ± 0.003 | 0.761 ± 0.054 | **0.494 ± 0.008** | 0.644 ± 0.076 | 0.655 ± 0.055 |
| Qwen2.5-1.5B-Inst | 0.616 ± 0.018 | **0.426 ± 0.013** | 0.537 ± 0.003 | 0.682 ± 0.006 | 0.522 ± 0.018 | 0.474 ± 0.005 | 0.719 ± 0.013 |
| Qwen2.5-3B-Inst | 0.424 ± 0.017 | 0.490 ± 0.005 | 0.491 ± 0.004 | 0.597 ± 0.009 | **0.398 ± 0.028** | 0.451 ± 0.005 | 0.512 ± 0.032 |
| Qwen2.5-7B-Inst | 0.401 ± 0.006 | 0.419 ± 0.004 | 0.641 ± 0.008 | 0.633 ± 0.008 | **0.382 ± 0.007** | 0.402 ± 0.020 | 0.540 ± 0.011 |
| Qwen2.5-14B-Inst | **0.247 ± 0.006** | 0.315 ± 0.003 | 0.334 ± 0.006 | 0.423 ± 0.004 | 0.364 ± 0.006 | 0.410 ± 0.006 | 0.471 ± 0.009 |
| Qwen2.5-32B-Inst | 0.397 ± 0.008 | **0.248 ± 0.004** | 0.272 ± 0.005 | 0.329 ± 0.008 | 0.310 ± 0.005 | 0.338 ± 0.007 | 0.414 ± 0.009 |
| Qwen-2.5-72B-Inst | **0.202 ± 0.009** | 0.319 ± 0.008 | 0.358 ± 0.010 | 0.428 ± 0.009 | 0.255 ± 0.010 | 0.322 ± 0.010 | 0.386 ± 0.010 |
| GPT-4o | 0.317 ± 0.009 | 0.253 ± 0.004 | 0.240 ± 0.004 | 0.354 ± 0.007 | **0.184 ± 0.004** | 0.196 ± 0.004 | 0.251 ± 0.008 |
| GPT-4o-mini | 0.389 ± 0.010 | 0.364 ± 0.006 | 0.340 ± 0.004 | 0.516 ± 0.005 | 0.302 ± 0.008 | **0.296 ± 0.005** | 0.415 ± 0.011 |
| GPT-5.2 | 0.271 ± 0.001 | 0.246 ± 0.024 | **0.167 ± 0.014** | 0.285 ± 0.017 | 0.201 ± 0.018 | 0.235 ± 0.012 | 0.276 ± 0.015 |
| Claude-Sonnet-4.5 | 0.114 ± 0.001 | 0.299 ± 0.001 | 0.213 ± 0.042 | 0.242 ± 0.001 | 0.162 ± 0.027 | **0.110 ± 0.001** | 0.217 ± 0.003 |
| Gemini-2.5-Pro | **0.108 ± 0.002** | 0.157 ± 0.002 | 0.123 ± 0.002 | 0.166 ± 0.002 | 0.130 ± 0.002 | 0.110 ± 0.002 | 0.145 ± 0.003 |
| **Total Wins** | **4/17** | **5/17** | **1/17** | **0/17** | **4/17** | **3/17** | **0/17** |

Table 1: Aggregate RCRPS results on CiK benchmark (mean ± standard error; lower is better). CorDP augmentation preserves base quantitative model accuracy while incorporating contextual reasoning, achieving best performance on 13/17 evaluated LLMs with improvements of up to 50% over DP. The best performing method for each model is in **bold**. Extended results are in Appendix D.

0.509, GPT-5.2: 0.504, Gemini-2.5-Pro: 0.457) perform comparably to dedicated quantitative forecasters (Chronos-Large: 0.492), demonstrating that their numerical forecasting capability is intact. The Execution Gap thus measures the additional improvement attainable when context is provided, isolating contextual application as the sole variable.

**Implications for model improvement.** These findings provide clear directions for model improvement: smaller models need better reasoning capabilities to reason about forecast effects, while larger models require enhanced ability to translate correct explanations into improved forecasts. Approaches such as RL-based post-training (DeepSeek-AI et al., 2025) may offer pathways to address both challenges.

# 5 CorDP - Direct Prompting for Forecast Correction

## 5.1 Limitations of LLM-based Context-Aided Forecasting

Real-world forecasting requires both the ability to incorporate contextual information when available, as well as high accuracy on standard quantitative forecasting tasks without textual context. Standard quantitative forecasting systems employ specialized, domain-specific models that are carefully tuned to achieve optimal performance for their use cases (Petropoulos et al., 2022; Januschowski et al., 2024), making them difficult to replace without significant performance degradation. Current approaches to context-aided forecasting attempt to replace these specialized models entirely with LLMs (Williams et al., 2025; Requeima et al., 2024), leading to suboptimal performance on quantitative forecasting tasks. Rather than replacing quantitative models, we propose an augmentation approach that preserves their specialized forecasting capabilities while adding contextual reasoning through LLMs. This approach leverages the strengths of both paradigms: the quantitative accuracy of domain-specific models and the contextual reasoning capabilities of LLMs.

## 5.2 Forecast Correction: A Practical Approach to Context-Aided Forecasting

We propose Direct Prompting for Forecast Correction (CorDP), where an LLM acts as a *forecast corrector* rather than a *forecaster* as in prior work (Williams et al., 2025; Requeima et al., 2024; Gruver et al., 2024). The LLM receives probabilistic forecasts from a quantitative model and is instructed to correct them based on the textual context, modifying only the portions where the context provides relevant insights. This approach is inspired by judgmental correction in the forecasting literature, where human forecasters incorporate contextual information post-hoc to improve quantitative forecasts (Hyndman & Athanasopoulos, 2021); CorDP automates this process with LLMs in a zero-shot manner.

We introduce two variants that differ in how they handle forecast distributions:

- **Median-CorDP:** The LLM corrects the median of the forecast distribution multiple times, generating a new context-informed distribution.

- **SampleWise-CorDP:** The LLM corrects each sample from the original probabilistic forecast, preserving the original distribution's structure.

These variants are fundamentally different in how they handle the forecast distribution: Median-CorDP discards the original distribution entirely, using only the base forecast median as a starting point and drawing new LLM samples to form the corrected distribution; SampleWise-CorDP instead preserves the original probabilistic structure by correcting each sample individually. The two variants are therefore complementary by design rather than combinable.

CorDP offers three key advantages: it preserves the quantitative forecasting accuracy of the base model, imposes minimal computational overhead by only correcting existing forecasts, and integrates seamlessly with existing forecasting pipelines without requiring infrastructure changes. CorDP is intended primarily as a practical way to achieve quality context-aided forecasting with much more *efficient* (small) models that struggle with direct prompting, by letting them bootstrap off stronger quantitative forecasts instead of forecasting from scratch. The CorDP prompt is provided in Appendix D.1. Cost and inference time for CorDP are reported in Appendix H.4; CorDP incurs similar cost and time as DP.

### 5.3 Performance of CorDP

**Overall performance.** Table 1 presents results aggregated across all tasks from CiK. CorDP methods achieve the best performance on 13/17 evaluated LLMs, benefiting models across different scales and families, including frontier models (GPT-5.2, Claude-Sonnet-4.5) with improvements of up to 50% over Direct Prompting. Side-by-side comparisons of DP and each CorDP variant per base forecaster are provided in Appendix D.2.

**SampleWise vs Median variants.** Among the 13 models where CorDP outperforms Direct Prompting, SampleWise-CorDP achieves the best performance in 7 models, while Median-CorDP is best for the remaining 6 models. The performance of CorDP methods differs widely depending on the quantitative forecaster used, with all winning methods except one using Lag-Llama's forecasts. This is likely because Lag-Llama is the best performing quantitative forecaster (as shown in the base forecaster row), highlighting the importance of selecting an appropriate base forecaster for CorDP. This points to a practical selection criterion: choose the best available quantitative forecaster as the base model for CorDP, evaluated on standard held-out no-context data, a routine step in any forecasting pipeline that requires no labeled context-aided data.

**CorDP consistently improves upon base forecasters.** Most models (11/13) perform strictly better than the base forecaster when used with CorDP. The remaining two models (Llama3.2-1B-Inst and Qwen2.5-1.5B-Inst) occasionally degrade the base quantitative forecast, yet still achieve substantial improvements over Direct Prompting, demonstrating the value of conditioning on base forecasts even for very small models.

**Direct Prompting remains best for select models.** While CorDP achieves best performance on 13 models, Direct Prompting remains superior for 4 models: Llama3.1-405B-Inst, Qwen2.5-14B, Qwen2.5-72B, and Gemini-2.5-Pro. This suggests that these models may be inherently strong forecasters with direct prompting, or that they leverage different reasoning patterns that do not benefit from bootstrapping off quantitative forecasts.

**Performance varies by task type.** Performance patterns across task types (detailed in Appendix D.3) reveal clear specialization: SampleWise-CorDP excels on partial RoI tasks (where context affects a subset of the prediction window), while Median-CorDP dominates on full-shape tasks and tasks with constraints, sometimes achieving perfect performance with frontier models (GPT-5.2, Claude-Sonnet-4.5). This task-type specialization provides practitioners with a principled selection criterion: prefer SampleWise-CorDP for partial-RoI tasks and Median-CorDP for full-shape or constrained tasks — a decision that can be made from task metadata alone, without labeled context-aided data. Example forecasts with both CorDP methods are provided in Appendix D.5.

**Practical implications and future directions.** These results establish CorDP as a practical solution for context-aided forecasting that leverages existing numerical forecasting tools, augmenting them with the advanced reasoning abilities of LLMs. A key benefit is the ability to achieve quality performance with much more efficient (small) models, where gains over Direct Prompting are largest, rather than requiring large or frontier models for state-of-the-art results. Success depends on careful LLM and base forecaster selection. Future work could explore fine-tuning LLMs with CorDP to better match the forecast distributions of specific quantitative forecasters. Analyzing the distributional properties of LLM-generated versus quantitative forecasts would also be useful for improving CorDP performance and understanding when the method is most effective.

## 6 IC-DP - In-Context Direct Prompting

### 6.1 Leveraging Historical Examples

Real-world forecasting applications typically deal with domain-specific contexts that repeat over time, such as seasonal heat waves in electricity consumption forecasting. Previous work has confirmed this pattern across various domains, where similar contextual events recur with varying impact on forecasts (Wang et al., 2024; 2025). In an ideal case, a model could be trained to understand and forecast with the domain-specific contexts. However, this approach requires significant overhead in model selection, training, and maintenance. In contrast, LLMs can leverage in-context learning to improve performance by learning from examples provided in the prompt (Brown et al., 2020), offering an alternative to domain-specific training. We explore this capability for context-aided forecasting by demonstrating to models how past contexts affected forecasts through in-context examples.

### 6.2 IC-DP: In-Context Forecasting

To test the in-context forecasting capabilities of LLMs, we modify Direct Prompt to include example context-aided forecasting tasks in the prompt, providing their respective histories, contexts, and ground truths (see Appendix E.1 for the prompt). We call this "In-Context Direct Prompt" (IC-DP). Following standard one-shot learning evaluation (Brown et al., 2020), we provide one past instance from the same task type: the time series and textual context differ, but the structural relationship between context and forecast remains identical. This minimal setup evaluates whether LLMs can learn the mapping from context to forecast effect from a single demonstration, mirroring realistic scenarios where limited historical examples of similar contexts are available (examples in Appendix E.4). IC-DP negligibly raises inference time and cost relative to DP, obtaining similar time and cost as DP (details in Appendix H.3).

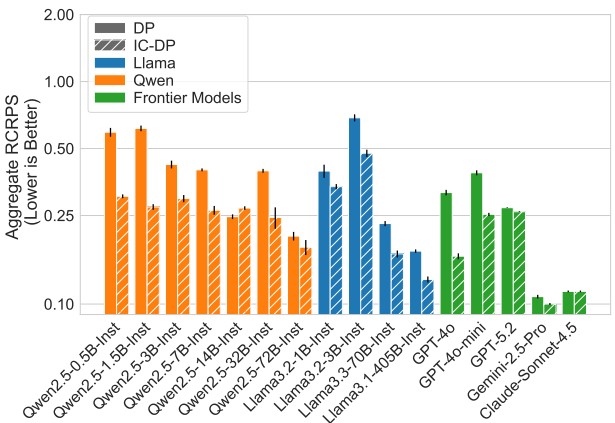

Figure 4: Aggregate RCRPS results comparing Direct Prompting (DP) with In-Context Direct Prompting (IC-DP). IC-DP improves performance for 14/16 models, with gains of 14–56% for small models and 20–40% for mid-size and large models, demonstrating that a single in-context example significantly enhances performance across model scales.

### 6.3 Performance Gains

**Overall improvements across model scales.** Aggregate results in Figure 4 show that IC-DP improves performance for 14/16 tested models, demonstrating that a single in-context example can significantly enhance context-aided forecasting capabilities. Small models benefit most with improvements of 14–56%, while mid-size and large models show 20–40% gains. Notably, IC-DP improves even the largest open-weight model, Llama3.1-405B-Inst, by 25%, and yields exceptional gains for GPT-4o (48% improvement), indicating that in-context examples benefit models across all scales. IC-DP also enables substantial efficiency gains: small models with IC-DP can match or exceed the performance of much larger models with DP alone.

**Task-type patterns reveal specialization.** IC-DP provides outsized improvements on tasks where the entire forecast is shaped by context (full-RoI tasks), with significant gains in RoI CRPS and constraints CRPS, and minor improvements in non-RoI CRPS (detailed results by task group in Table 15 in the appendix). Frontier models show a bimodal pattern: some benefit substantially from IC-DP (GPT-4o: 48% improvement), while others are already near-optimal with DP alone (Claude-Sonnet-4.5: 0% improvement; GPT-5.2, Gemini-2.5-Pro: 4–7% improvement), with benefits concentrated on context-heavy tasks. Qwen-2.5-14B remains an outlier, degrading 9% on average, further evidence for its strong zero-shot forecasting capabilities with DP where modifications may be detrimental. IC-DP differs from CorDP in the task types where it provides most improvements, indicating that the two strategies can be complementary and selected according to application requirements, which we discuss this further in Section 8.

**Practical implications and future directions.** The general success of IC-DP validates that in-context examples can significantly enhance LLM forecasting capabilities. Future work could explore varying the number and similarity of examples to optimize this trade-off, or using synthetic examples to handle contexts completely different from previously seen ones.

# 7 RouteDP: Direct Prompt with Model Routing

## 7.1 Balancing Performance and Efficiency

Work in context-aided forecasting has shown that larger LLMs generally perform better on average (Williams et al., 2025; Zhang et al., 2025; Kupferschmidt et al., 2024), which we also observe with the DP method. However, utilizing large LLMs such as Llama-405B-Inst (Grattafiori et al., 2024) for every task is prohibitively expensive and often unnecessary, as smaller models may suffice for simpler tasks. Model routing strategies (Ong et al., 2024; Madras et al., 2018) address this by allocating tasks adaptively: routing challenging cases to larger, more capable models while directing easier tasks to smaller, more efficient ones. This approach is particularly valuable when operating under fixed compute budgets, where the goal is to maximize average performance by strategically allocating resources across tasks of varying difficulty. We investigate whether LLMs can assess task difficulty to enable effective routing: given a set of context-aided forecasting tasks and available compute, can models rank tasks by difficulty and route the hardest cases to a pre-determined large model while using a small model for easier tasks?

## 7.2 Task Difficulty Ranking and Two-Model Routing

We explore a two-model routing setup where a small model handles most tasks, while a large model is reserved for the most challenging cases. A separate router model is prompted zero-shot to assign a difficulty score between 0 (easiest) and 1 (hardest) to each task based on its context and history (prompt in Appendix F.1). For a given compute budget and $N$ context-aided forecasting tasks, the $k$ most difficult tasks as judged by the router are sent to the large model, while the remaining $N - k$ tasks are handled by the small model. To measure performance at various compute budgets, we vary $k$ from 0 (all tasks to the small model) to $N$ (all tasks to the large model) and measure aggregate performance as a function of $k$. We call this approach RouteDP. While we evaluate RouteDP in a batch setting for clarity, the numerical difficulty scores naturally support per-task routing via a score threshold; we discuss this equivalence in Appendix F.2. We compare RouteDP against two baselines: random routing (assigning $k$ random tasks to the large model) and ideal routing (assigning tasks in the order that maximally improves average RCRPS, providing an upper bound on performance). To quantify router effectiveness, we compute how much performance gain routing with RouteDP provides over random routing at each value of $k$. We aggregate this value across all values of $k$ to compute the area that the router captures between random and ideal routing. Higher values indicate more effective task difficulty ranking. In our protocol, we evaluate Llama-3.1-405B-Inst as the large model and models from the Qwen family as both the small model and router model. Additional details on the protocol are provided in Appendix F. Time and cost for computing difficulty scores with RouteDP are reported in Appendix H.5.

| Router Model | Percentage of tasks sent to large model | | | | | |
|---|---|---|---|---|---|---|
| | **0%** | 20% | 40% | 60% | 80% | **100%** |
| Qwen2.5-0.5B-Inst | $0.592 \pm 0.027$ | **$0.316 \pm 0.027$** | **$0.222 \pm 0.005$** | **$0.206 \pm 0.005$** | $0.199 \pm 0.004$ | $0.173 \pm 0.003$ |
| Qwen2.5-1.5B-Inst | $0.592 \pm 0.027$ | $0.504 \pm 0.009$ | $0.449 \pm 0.007$ | $0.404 \pm 0.004$ | $0.407 \pm 0.004$ | $0.173 \pm 0.003$ |
| Qwen2.5-3B-Inst | $0.592 \pm 0.027$ | $0.507 \pm 0.026$ | $0.490 \pm 0.026$ | $0.393 \pm 0.003$ | $0.282 \pm 0.003$ | $0.173 \pm 0.003$ |
| Qwen2.5-7B-Inst | $0.592 \pm 0.027$ | $0.510 \pm 0.010$ | $0.437 \pm 0.007$ | $0.412 \pm 0.004$ | **$0.181 \pm 0.004$** | $0.173 \pm 0.003$ |
| Qwen2.5-14B-Inst | $0.592 \pm 0.027$ | $0.581 \pm 0.027$ | $0.439 \pm 0.027$ | $0.324 \pm 0.027$ | $0.187 \pm 0.004$ | $0.173 \pm 0.003$ |
| Qwen2.5-32B-Inst | $0.592 \pm 0.027$ | $0.383 \pm 0.010$ | $0.368 \pm 0.008$ | $0.230 \pm 0.006$ | $0.196 \pm 0.004$ | $0.173 \pm 0.003$ |
| Qwen2.5-72B-Inst | $0.592 \pm 0.027$ | $0.509 \pm 0.010$ | $0.395 \pm 0.009$ | $0.287 \pm 0.009$ | $0.243 \pm 0.009$ | $0.173 \pm 0.003$ |

Table 2: Average RCRPS with Qwen2.5-0.5B-Inst as the small model, as different routers direct varying percentages of tasks to the large model (Llama-405B-Inst). Percentage labels are nominal; actual task counts are floored (e.g., 20% → 14 of 71 tasks). Grayed columns (0% and 100%) show baseline performance that is identical across all routers; **bold** indicates best router at each budget. RouteDP achieves substantial improvements: routing just 20% of tasks yields 46.6% better performance than the small model baseline. Qwen2.5-0.5B-Inst performs best as its own router at low-to-moderate routing budgets. Results with other small models in Appendix F.3.

## 7.3 Results with RouteDP

**RouteDP effectively identifies and routes difficult tasks.** RouteDP meaningfully exploits task differences to predict difficulty and route tasks, achieving significantly better performance than random routing. For example, consider the case of using Qwen-2.5-0.5B-Inst as the small model for most tasks, adaptively routing tasks to the large model as required. As shown in Figure 5, RouteDP can improve over random routing, capturing 66% of the total area between random and ideal routing, and sharp improvements in average performance is seen as tasks are routed, indicating that the router can meaningfully capture notions of task difficulty to identify when to use the large model.

**RouteDP provides significant performance gains.** Table 2 shows the performance of Qwen2.5-0.5B-Inst (the main model), as an increasing percentage of tasks are routed to the large model, with different models as the router model. RouteDP serves as a simple yet effective approach to improve performance: routing even just the most difficult 20% of tasks to the large model yields a sharp improvement in average RCRPS, with a 46.6% improvement already, with further improvements following as more tasks are routed. This shows that practitioners can achieve significant improvements in accuracy at a fraction of the cost, which is highly relevant for real-world deployment. This holds across several routers and tested small models (results in Appendix F.3).

**Each model is its own best router.** Each model performs best as its own router: as shown in Table 2, Qwen2.5-0.5B-Inst routing for itself achieves the best results across different compute budgets, a trend that holds for most small models tested (Table 17). Further, among the tested router models,

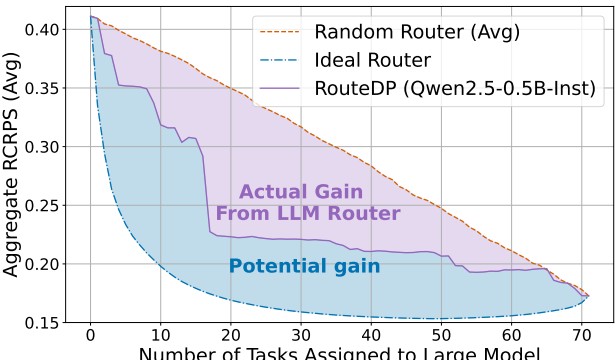

Figure 5: The plot shows the average RCRPS achieved using Qwen2.5-0.5B-Inst as the small model as an increasing percentage of tasks are routed to the large model (Llama-405B-Inst). We use Qwen2.5-0.5B-Inst as the router, and compare this to random and ideal routing. The router can meaningfully capture task difficulty and route tasks to improve aggregate performance. A significant 66% of the possible area between random and ideal routing is captured by RouteDP. Results with other models are in Appendix F.4.

several small models, particularly Qwen2.5-0.5B-Inst, stand out as disproportionately effective routers.

**Smallest models benefit most from routing.** Next, we observe that the smallest models benefit the most from routing. This is expected as the possible improvement with routing decreases with model size: for Qwen2.5-0.5B-Inst, there is a 71% reduction possible (using Llama-3.1-405B-Inst's performance as the

ceiling), while for Qwen2.5-14B-Inst, the possible reduction is only 30%. This is reflected in the performance improvement obtained by routing - for e.g. by routing 20% of the tasks, Qwen2.5-0.5B-Inst achieves a 46.6% improvement while Qwen2.5-14B-Inst only achieves 16.6% with their respective best routers. Area captured by RouteDP for various small models (and various routers) is given in Table 17.

**Practical implications.** RouteDP enables substantial performance gains by leveraging LLMs' ability to assess task difficulty. Even small routers like Qwen2.5-0.5B-Inst capture the majority of possible improvements, offering immediate practical benefits for deployment. Future work could explore training routers to better predict task difficulty or incorporating domain knowledge about inherently difficult task characteristics.

## 8 Integrating the Proposed Strategies

As illustrated in Figure 1, the strategies extend DP along three dimensions: diagnostics (FxDP), accuracy (IC-DP, CorDP), and efficiency (RouteDP). A natural question is whether they are complementary to each other in practice. We explore this complementarity through targeted combination experiments. We find that IC-DP and CorDP can be combined, with the integrated approach achieving further accuracy improvements beyond either method alone (Appendix D.4); similarly, RouteDP (routing) can be used with IC-DP or CorDP to even further improve performance under resource constraints (Appendix F.5). These experiments illustrate two specific combinations, though several other combinations, such as pairing FxDP's diagnostic capabilities with accuracy-focused strategies, or combining all three dimensions where deployment settings permit - may be viable. Each strategy involves different inference cost and latency profiles, but the improvements they enable may justify these trade-offs in practice (details in Appendix H). Because these dimensions are largely orthogonal, practitioners can flexibly adopt and combine strategies based on their specific deployment constraints and objectives.

## 9 Discussion and Future Work

This work addresses the challenge of context-aided time series forecasting with large language models (LLMs). We propose a framework of four orthogonal strategies that extend naïve direct prompting: Direct Prompting with Forecast Effect Explanation (FxDP), which elicits explanations to diagnose failure modes; Direct Prompting for Forecast Correction (CorDP), which refines existing forecasts with context; In-Context Direct Prompting (IC-DP), which leverages historical examples; and Direct Prompting with Model Routing (RouteDP), which adaptively allocates models based on task difficulty. Through extensive evaluation across model families: Qwen, Llama, Gemini, GPT, and Claude, we analyze the performance, limitations, and trade-offs of each strategy. Our findings reveal both the capabilities and fundamental limitations of LLMs for context-aided forecasting: FxDP uncovers the "Execution Gap" where models explain forecast effects correctly but fail to apply their reasoning; IC-DP and CorDP demonstrate substantial accuracy improvements (up to 25-50% relative CRPS improvement); and RouteDP shows that adaptive routing between small and large models can approach large model accuracy while significantly reducing inference costs. Through targeted combination experiments, we demonstrate that these strategies can be integrated to address multiple deployment objectives simultaneously, providing practitioners with a flexible toolkit for real-world applications.

These findings open several directions for future research in context-aided forecasting. First, LLMs as orchestrators (Ye et al., 2025) or agents (Garza & Rosillo, 2025) could determine when to apply specific forecasting techniques or adjust their outputs; understanding when such approaches justify their computational cost would enhance their practical appeal. Notably, CorDP already takes a step in this direction by decoupling numerical forecasting from contextual reasoning - mirroring what a tool-use agent would do architecturally, and can serve as a strong prompting-based baseline for future agentic methods in this setting. Second, while we focus on the DP method, extending these strategies to other approaches such as LLMP (Requeima et al., 2024) and LLMTime (Gruver et al., 2024) would broaden their applicability. Third, evaluating these strategies in more unconstrained settings, where context may be irrelevant or excessively long, is an interesting direction, however requires developing appropriate benchmarks first. We note that CiK, while uniquely suited for controlled evaluation of context-aided forecasting, does not cover these unconstrained scenarios, and results should be interpreted within this scope. Fourth, moving beyond zero-shot and few-shot methods to training-based methods could expand the scope of these strategies, enabling trained routers in RouteDP or models that leverage arbitrary base forecasters in CorDP. Notably, FxDP already provides concrete

guidance on where fine-tuning would help most: small models primarily fail at the reasoning stage, while larger models fail at the execution stage, giving a clear target for fine-tuning efforts at each model scale. Finally, the high cost of LLMs relative to canonical forecasting methods remains a barrier to deployment; these findings should inform the development of more efficient context-aided forecasting models from the ground-up, keeping the requirements of the respective forecasting application in mind.

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

# Appendix

## Table of Contents

# A  Additional Details on the Context-Is-Key (CiK) benchmark

## A.1  The RCRPS metric

We use the Region-of-Interest CRPS (RCRPS) metric to evaluate context-aided forecasting performance (Williams et al., 2025), which modifies the CRPS metric (Gneiting & Raftery, 2007) to prioritize context-sensitive windows and accounts for constraint satisfaction. Given an inferred forecast distribution $\widetilde{\mathbf{X}}_F$ and a ground truth $\mathbf{x}_F$, the RCRPS metric is defined as:

$$\mathrm{RCRPS}(\widetilde{\mathbf{X}}_F, \mathbf{X}_F) \;=\; \alpha \cdot \left[\frac{1}{2|\mathcal{I}|}\sum_{i \in \mathcal{I}} \mathrm{CRPS}(\widetilde{X}_i, x_i) + \frac{1}{2|\neg\mathcal{I}|}\sum_{i \in \neg\mathcal{I}} \mathrm{CRPS}(\widetilde{X}_i, x_i) \;+\; \beta \cdot \mathrm{CRPS}(v_{\mathbf{C}}(\widetilde{\mathbf{X}}_F), 0)\right],$$

where the CRPS for a univariate forecast $\widetilde{X}$ and ground-truth realization $x$ is defined as:

$$\mathrm{CRPS}(\widetilde{X}, x) = \int_{-\infty}^{\infty} \left[\Phi_{\widetilde{X}}(y) - \mathbb{1}(y \geq x)\right]^2 dy, \tag{1}$$

where $\Phi_{\widetilde{X}}(y)$ is the CDF of $\widetilde{X}$ and $\mathbb{1}$ is the indicator function.

where the terms respectively account for the CRPS inside the RoI, the CRPS outside of the RoI, and the constraint violation penalty. The $\alpha$ term is a task-dependent normalization factor to make the RCRPS scale-independent, while $\beta$ is a scaling factor that controls the impact of constraint violation on the score; we use $\beta = 10$ in our experiments following prior work (Williams et al., 2025; Zhang et al., 2025).

# B  Additional Results with the Direct Prompt (DP) Method

We select Direct Prompting (DP) as our foundational baseline over other methods like LLMTime(Gruver et al., 2024) and LLM Processes(Requeima et al., 2024) because DP avoids the prohibitive computational overhead of digit-by-digit autoregressive generation. Prior work on multiple benchmarks (Williams et al., 2025; Kupferschmidt et al., 2024) has already established DP as a highly efficient and competitive baseline for context-aided tasks, making it the ideal foundation for our diagnostic and efficiency-focused extensions.

## B.1  Aggregate Results of models

Results of various models with Direct Prompt (DP), with and without context are given in Table 3.

## B.2  Results of models on various kinds of tasks

Results of various models with Direct Prompt (DP), with and without context, partitioned by different kinds of tasks are given in Table 4.

# C  Additional Details on FxDP

## C.1  FxDP Prompt

We use the following prompt for the FxDP method, where **{history}** is replaced by the respective numerical history for the task instance in the format (timestamp, value), **{context}** is replaced by the respective textual context for the task instance, and ((**pred_time**)) is replaced with the prediction timesteps.

| Model | Without Context | With Context |
|---|---|---|
| Qwen2.5-0.5B-Inst | **0.404 ± 0.028** | 0.592 ± 0.027 |
| Qwen2.5-1.5B-Inst | 0.631 ± 0.039 | **0.616 ± 0.018** |
| Qwen2.5-3B-Inst | 0.513 ± 0.039 | **0.424 ± 0.017** |
| Qwen2.5-7B-Inst | 0.610 ± 0.011 | **0.401 ± 0.006** |
| Qwen2.5-14B-Inst | 0.551 ± 0.007 | **0.247 ± 0.006** |
| Qwen2.5-32B-Inst | 0.607 ± 0.008 | **0.397 ± 0.008** |
| Qwen2.5-72B-Inst | 0.549 ± 0.009 | **0.202 ± 0.009** |
| Llama3.2-1B-Inst | 0.481 ± 0.028 | **0.396 ± 0.027** |
| Llama3.2-3B-Inst | 0.950 ± 0.041 | **0.687 ± 0.025** |
| Llama3-8B-Inst | 0.758 ± 0.009 | **0.543 ± 0.026** |
| Llama3.3-70B-Inst | 0.700 ± 0.009 | **0.230 ± 0.006** |
| Llama3.1-405B-Inst | 0.686 ± 0.011 | **0.173 ± 0.003** |
| GPT-4o | 0.665 ± 0.004 | **0.317 ± 0.009** |
| GPT-4o-mini | 0.648 ± 0.012 | **0.389 ± 0.010** |
| GPT-5.2 | 0.504 ± 0.002 | **0.271 ± 0.001** |
| Gemini-2.5-Pro | 0.457 ± 0.005 | **0.108 ± 0.002** |
| Claude-Sonnet-4.5 | 0.509 ± 0.005 | **0.114 ± 0.001** |
| Chronos-Large | 0.492 ± 0.004 | - |
| Lag-Llama | 0.382 ± 0.011 | - |
| Arima | 0.636 ± 0.014 | - |

Table 3: Aggregate Results (RCRPS) of models on the CiK benchmark.

| Model | ROI Without Context | ROI With Context | non-ROI Without Context | non-ROI With Context | Full ROI Without Context | Full ROI With Context | Constraints Without Context | Constraints With Context |
|---|---|---|---|---|---|---|---|---|
| Llama3.2-1B-Inst | 0.357 ± 0.018 | **0.336 ± 0.026** | **0.236 ± 0.018** | 0.248 ± 0.026 | 0.607 ± 0.045 | **0.467 ± 0.041** | 0.604 ± 0.064 | **0.275 ± 0.092** |
| Llama3.2-3B-Inst | 0.832 ± 0.118 | **0.281 ± 0.013** | 0.769 ± 0.030 | **0.162 ± 0.013** | 1.022 ± 0.048 | **1.004 ± 0.040** | **0.613 ± 0.075** | 1.030 ± 0.090 |
| Llama3-8B-Inst | 0.336 ± 0.017 | **0.255 ± 0.008** | 0.239 ± 0.017 | **0.163 ± 0.008** | 1.078 ± 0.009 | **0.771 ± 0.043** | 0.460 ± 0.199 | **0.169 ± 0.172** |
| Qwen2.5-1.5B-Inst | 0.327 ± 0.009 | **0.317 ± 0.020** | **0.142 ± 0.009** | 0.224 ± 0.020 | 0.900 ± 0.065 | **0.851 ± 0.026** | **0.379 ± 0.242** | 0.706 ± 0.147 |
| Qwen2.5-7B-Inst | 0.520 ± 0.008 | **0.285 ± 0.006** | **0.157 ± 0.008** | 0.164 ± 0.006 | 0.794 ± 0.018 | **0.521 ± 0.009** | 0.476 ± 0.041 | **0.470 ± 0.078** |
| Qwen2.5-14B-Inst | 0.376 ± 0.008 | **0.162 ± 0.005** | 0.155 ± 0.008 | **0.146 ± 0.005** | 0.745 ± 0.010 | **0.310 ± 0.010** | 0.473 ± 0.019 | **0.039 ± 0.015** |
| Qwen2.5-32B-Inst | 0.537 ± 0.003 | **0.116 ± 0.001** | 0.152 ± 0.003 | **0.140 ± 0.001** | 0.786 ± 0.014 | **0.580 ± 0.013** | 0.503 ± 0.031 | **0.479 ± 0.019** |
| Llama3.3-70B-Inst | 0.531 ± 0.010 | **0.105 ± 0.003** | **0.147 ± 0.010** | 0.182 ± 0.003 | 0.945 ± 0.014 | **0.289 ± 0.011** | 0.475 ± 0.031 | **0.000 ± 0.024** |
| Llama3.1-405B-Inst | 0.537 ± 0.002 | **0.126 ± 0.004** | **0.147 ± 0.002** | 0.150 ± 0.004 | 0.920 ± 0.019 | **0.196 ± 0.005** | 0.478 ± 0.038 | **0.004 ± 0.009** |
| Qwen2.5-3B-Inst | 0.280 ± 0.006 | **0.269 ± 0.015** | 0.155 ± 0.006 | 0.186 ± 0.015 | 0.713 ± 0.065 | **0.558 ± 0.027** | **0.087 ± 0.147** | 0.234 ± 0.056 |
| Qwen2.5-72B-Inst | 0.530 ± 0.001 | **0.115 ± 0.004** | 0.141 ± 0.001 | **0.138 ± 0.004** | 0.695 ± 0.015 | **0.253 ± 0.015** | 0.513 ± 0.034 | **0.032 ± 0.028** |
| Qwen2.5-0.5B-Inst | **0.249 ± 0.005** | 0.339 ± 0.010 | 0.149 ± 0.005 | **0.129 ± 0.010** | **0.544 ± 0.046** | 0.836 ± 0.046 | 0.557 ± 0.104 | **0.243 ± 0.103** |
| GPT-4o | 0.524 ± 0.003 | **0.123 ± 0.004** | 0.148 ± 0.003 | **0.106 ± 0.004** | 0.888 ± 0.006 | **0.455 ± 0.014** | 0.477 ± 0.009 | **0.455 ± 0.029** |
| GPT-4o-mini | 0.495 ± 0.008 | **0.263 ± 0.005** | **0.102 ± 0.008** | 0.150 ± 0.005 | 0.885 ± 0.019 | **0.513 ± 0.017** | 0.461 ± 0.042 | **0.001 ± 0.032** |
| GPT-5.2 | 0.564 ± 0.004 | **0.104 ± 0.001** | 0.107 ± 0.004 | **0.095 ± 0.001** | 0.618 ± 0.003 | **0.387 ± 0.001** | 0.470 ± 0.004 | **0.000 ± 0.002** |
| Gemini-2.5-Pro | 0.509 ± 0.003 | **0.112 ± 0.001** | 0.097 ± 0.003 | **0.081 ± 0.001** | 0.561 ± 0.008 | **0.116 ± 0.003** | 0.462 ± 0.006 | **0.000 ± 0.005** |
| Claude-Sonnet-4.5 | 0.535 ± 0.001 | **0.108 ± 0.001** | 0.109 ± 0.001 | **0.090 ± 0.001** | 0.636 ± 0.008 | **0.124 ± 0.002** | 0.488 ± 0.010 | **0.000 ± 0.003** |
| Chronos-Large | 0.536 ± 0.003 | | 0.115 ± 0.003 | | 0.605 ± 0.006 | | 0.487 ± 0.010 | |
| Lag-Llama | 0.224 ± 0.005 | | 0.202 ± 0.005 | | 0.497 ± 0.018 | | 0.204 ± 0.037 | |
| Arima | 0.272 ± 0.004 | | 0.159 ± 0.004 | | 0.921 ± 0.023 | | 0.843 ± 0.050 | |

Table 4: Aggregate Results (RCRPS) of models on various groups of tasks from the CiK benchmark, with the DP method.

```
I have a time series forecasting task for you.

Here is some context about the task. Make sure to factor in any background knowledge,
satisfy any constraints, and respect any scenarios.
<context>
{context}
</context>

Here is a historical time series in (timestamp, value) format:
<history>
{history}
</history>

You are tasked with predicting the value at the following timestamps: {pred_time}.

First, within <reason> and </reason> tags, walk-through how you would incorporate each piece
    of the context to improve your forecast. If you think any of the context is irrelevant,
    please indicate. At the end, state the effect of the context on the forecast by stating
    'Therefore, the effect of the context on the forecast would be that' continued by the
    effect of the context on the forecast.

Next, return your forecast in (timestamp, value) format in between <forecast> and </forecast>
    tags.
Do not include any other information (e.g., comments) in the forecast.
```

One could use constrained decoding tools such as lm-format-enforcer (Gat & contributors, 2024) and XGrammar (Dong et al.) to constrain the output format, however we found that using using constrained decoding with free-form text (between the <reasoning> and </reasoning>) was very slow, taking several hours for a single instance and at times not completing. Therefore, we do not use any constrained decoding and instead retry 15 times if a model fails to output in the specified format. We find that all models successfully produce outputs in the required format without requiring 15 retries.

## C.2 Ground Truth Forecast Explanations: Protocol and Validation

Following the CiK benchmark design (Williams et al., 2025), we focus on tasks where context effects are localized to specific forecast regions. This subset is particularly well-suited for our diagnostic approach, as the ground truth forecast effect can be precisely specified and the impact of applying that effect can be clearly measured within the Region of Interest (RoI).

### C.2.1 Human Evaluation - Verifying ground truth forecast effects

We use the Prolific platform (`https://www.prolific.com`) to conduct a human evaluation, following prior work that validates ground-truth explanations and reasoning traces with human annotators (Lightman et al., 2023b; Lee & Hockenmaier, 2025).

**Study Name**: Check Whether Time Series Forecast Explanations Make Sense

**Study Description**:

This study asks you to evaluate short explanations about time series forecasting scenarios.

For each task, you will see:

A context describing a real-world situation (e.g., weather, traffic, or system behaviour)

A short explanation claiming how that context should affect a future forecast

Your role is to decide whether the explanation correctly and sufficiently explains how the
    context affects the forecast.

You will respond using:

Yes---if the explanation is correct and sufficient

No---if the explanation is incorrect or missing an essential detail

Additional guidance:
No numerical forecasting is required.
Focus on correctness, not writing quality.
Some tasks may look very similar---please read each carefully.

If you answer No, you may optionally add a brief note explaining why.

Examples are given below.

The study is straightforward and takes approximately 20 minutes to complete.

Example 1

Context: The weather was unusually warm in NYC on July 4th 2026, but this will not repeat.

Forecast Explanation:

The effect of the context on the forecast would be that the forecast has to exclude the data
    from the unusually warm day on the 4th of July 2026, and extrapolate the underlying trend
    and seasonality to predict future temperatures."

Your Answer: Yes

(The forecast explanation correctly explains the effect of the context on the forecast)

Example 2

Context: For 4 days from 2026-07-02, traffic to the website would be 20% higher

Forecast Explanation: The effect of the context on the forecast would be that from 2026-07-02
    to 2026-07-07, values would be 20% higher.

Your Answer: No

Reason: Wrong end date---should be 2026-07-05 (4 days from 2026-07-02)

We recruit **5** participants per item via the Prolific platform[1] (Palan & Schitter, 2018), following standard practices in NLP evaluation (Zhou et al., 2024; Zhang et al., 2024; Ethayarajh et al., 2024) and annotation protocol design with small expert panels (Snow et al., 2008; Jamison & Gurevych, 2015). We use the screening criterion that participants are fluent in English and have passed Prolific's Qualified AI Taskers assessment (demonstrating advanced reasoning, fact-checking, and writing skills). We determine correctness via majority vote, a widely-used approach for aggregating crowdsourced annotations (Snow et al., 2008; Artstein & Poesio, 2008). Participants are compensated 2.5 pounds for an estimated 20 minutes of work. Our total spend for human evaluation is 17.86 pounds including Prolific's fees.

**Screenshots of the interface**  Screenshots of the interface are provided in Figure 6 and Figure 7.

**Attention Checks and Quality Control**: To ensure annotation quality and identify participants who are not carefully reading the instructions or task content, we include control tasks (attention checks) where the correct answer is unambiguous. These are standard practice in crowdsourced annotation to filter low-quality responses (Snow et al., 2008).

We include 2 control tasks with deliberately incorrect forecast effect explanations, where the expected answer is "No" (the explanation does not match the context):

1. **RestaurantValentinesDayOrders (quantitative error)**:

   *Context:* "Background: This is the hourly number of food orders at a popular downtown restaurant. Constraints: None. Scenario: Suppose that Valentine's Day falls on 2024-02-14, and the restaurant expects 3 times the usual number of orders from 17:00:00 to 23:00:00 (6 hours)."

   *Provided explanation:* "The effect of the context on the forecast would be that the forecasted values for 6 hours, i.e. 2024-02-14 17:00:00, 2024-02-14 18:00:00, 2024-02-14 19:00:00, 2024-02-14 20:00:00, 2024-02-14 21:00:00, 2024-02-14 22:00:00, would be 2 times the usual value (i.e. multiplied by 2), to account for the Valentine's Day rush."

   The context states 3 times, but the explanation incorrectly says 2 times. This tests whether annotators catch quantitative mismatches.

2. **GymClosedChristmasHoliday (directional error)**:

   *Context:* "Background: This is the daily number of gym members checking in at FitLife Fitness Center. Constraints: None. Scenario: The gym is closed for the Christmas holiday from 2024-12-24 to 2024-12-26 (3 days)."

   *Provided explanation:* "The effect of the context on the forecast would be that the forecasted values for 3 days, i.e. 2024-12-24, 2024-12-25, 2024-12-26, would be significantly higher than usual, to account for people exercising more during the holiday season."

   The context states the gym is closed (check-ins should be zero), but the explanation incorrectly claims check-ins would be higher. This tests whether annotators catch directional errors.

All participants (5/5) correctly identified both control tasks as incorrect, indicating high attention and task comprehension.

**Results**:

We obtain unanimous agreement (5/5) on 45% of items and 4/5 agreement on the remaining 55% of items. Our task is binary verification against objective criteria (time windows, direction, and magnitude of effects specified in the context), and the high agreement rates (80-100% per item) indicate that the ground truth forecast effects are unambiguous and annotators can reliably verify whether explanations match them.

---

[1] https://www.prolific.com

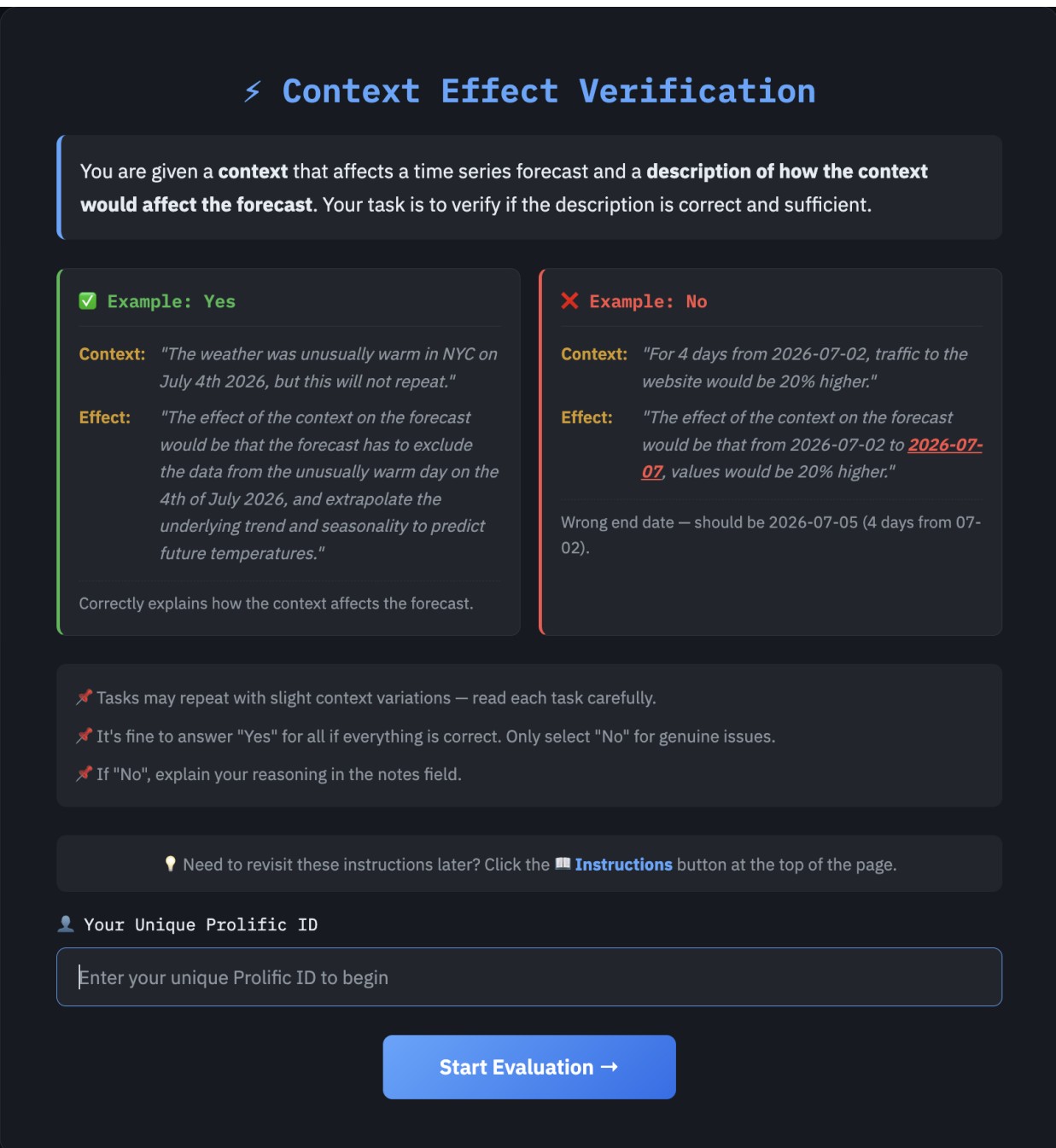

Figure 6: Welcome Page for the Prolific Human Evaluation

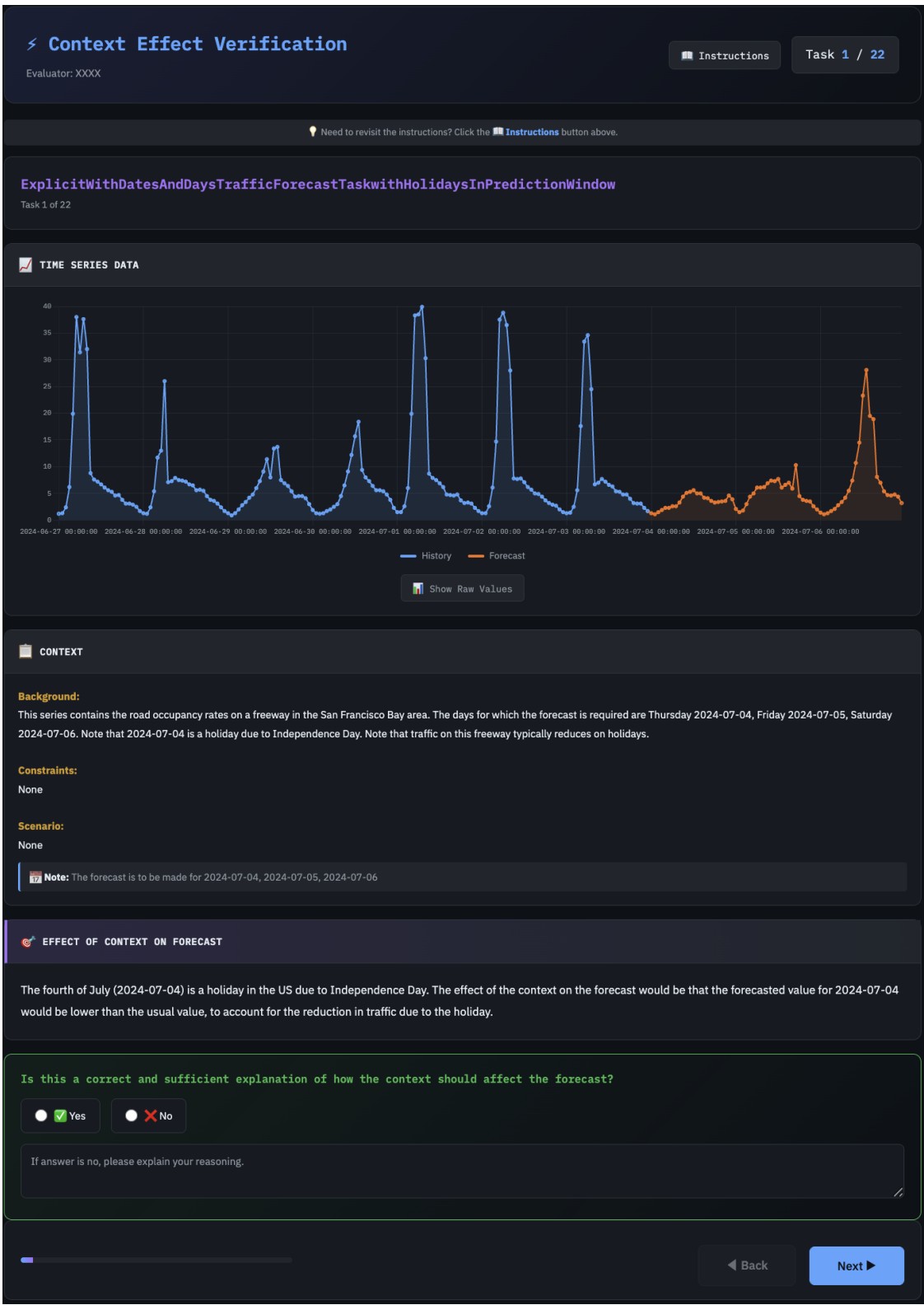

Figure 7: Page shown for each Task in the Human Evaluation

### C.3 Evaluating the Forecast Effect Explanations of Models: Protocol and Validation

Following prior work that compares model reasoning traces to gold-standard or expert-labeled references to evaluate correctness (Lightman et al., 2023a; Jacovi et al., 2024), we evaluate whether model-generated forecast effect explanations contain the key points from our verified ground truth effects.

#### C.3.1 LLM Judge Panel

To compare a model's forecast effect explanation produced with FxDP with the ground truth, we use the following prompt.

```
You will evaluate whether model-generated reasoning traces contain the key points from
    expert-written ground truth reasoning.

For each sample, you will see:

- Task Context: Background information about a forecasting scenario
- Ground Truth Reasoning: Expert explanation of how context affects the forecast
- Model Reasoning: Model-generated reasoning trace to evaluate

Your task: Does the model reasoning contain the key points from the ground truth?
i.e. Would you be able to precisely make the same forecast with the model reasoning as you
    would with the ground truth reasoning?
Note: The model reasoning should not bluntly repeat the context, but rather mention the effect
    that the context would have on the forecast.
Note: Double-check the dates. The context or the model's reasoning may refer to a 2 hour
    window affecting timestamps 1:00:00 and 2:00:00, while the ground truth reasoning may
    refer to the same window as "2 hours from 1:00:00 to 3:00:00". They are interchangeable,
    as the forecast is always made for the whole period starting with the stated timestamp.
    Therefore, the model reasoning may use either of the two formats.
Here is your task:

Context:
{context}

Model Reasoning:
{model_reasoning}

Ground Truth Reasoning:
{ground_truth_reasoning}

Answer with exactly one word: YES or NO between <answer> and </answer> tags, followed by a
    very concise explanation between <explanation> and </explanation> tags.
```

#### C.3.2 Human Evaluation

We use the Prolific platform (https://www.prolific.com) to conduct a human evaluation.

**Study Name**: Compare AI Reasoning to Expert Explanations

**Study Description**:

```
Task Information

We are testing how well AI models understand forecasting scenarios. In this task, you will
    evaluate if a model's reasoning matches expert-written "Ground Truth."

Your task is expected to take 30 minutes.

For each sample, you will be presented with three items:
Task Context: Background info on a specific scenario.
Ground Truth Forecast Effect: An expert explanation of how that scenario affects the
    forecast.
Model Reasoning: The AI's attempt to explain the effect.

Your Task:

You must answer one main question for each sample:
"Does the model reasoning contain the key points from the ground truth?"
YES: The key points are present.
NO: The key points are missing. (If No, please explain why in a few words).

CRITICAL GUIDELINES: Please keep these two rules in mind while grading:

 Extra Info is Okay: The model's reasoning trace may contain extra information not
     specified in the ground truth. This is fine. Your only job is to check if the Ground
     Truth key points are present anywhere in the model's text.

 No Blunt Repetition: The model should not simply repeat the "Task Context" word-for-word.
     It must explicitly mention the effect the context would have on the forecast.

NO FORECASTING REQUIRED: You do not need to perform any forecasting, math, or data
    analysis yourself. Your only task is to compare two short pieces of text.

Example 1:

Context: This is hourly food orders at a downtown restaurant. Valentine's Day falls on
    2024-02-14, and the restaurant expects 3 times the usual orders from 17:00 to 23:00 (6
    hours).

Ground Truth Reasoning: The forecast should show 3x the usual orders for 6 hours from
    17:00 to 23:00 on February 14th, to account for the Valentine's Day dinner rush.

Model Reasoning: Valentine's Day is a major dining occasion. Based on the context, I will
    forecast orders at triple the normal rate during the evening hours (17:00-23:00) on
    2024-02-14, then return to typical levels afterward.

Your Answer: Yes

 Why YES? The model correctly captures all key points:

- Correct multiplier (3x / triple)
- Correct time window (17:00-23:00)
- Correct date (February 14th)

 (Continued on next page...)
```

```
Example 2:

Context: This is hourly food orders at a downtown restaurant. Valentine's Day falls on
    2024-02-14, and the restaurant expects 3 times the usual orders from 17:00 to 23:00 (6
    hours).

Ground Truth Reasoning: The forecast should show 3x the usual orders for 6 hours from
    17:00 to 23:00 on February 14th, to account for the Valentine's Day dinner rush.

Model Reasoning: Valentine's Day is coming up, which means the restaurant will be shut
    down. The restaurant is closed on Valentine's Day, so the forecast should be 0 for the
    entire day.

Your Answer: No

Why NO? The model misunderstood the context completely, and the reasoning is not correct
    (it predicted 0 instead of 3x).
```

**Sampling Strategy**   From a total of 200 model-task pairs (10 models × 20 tasks), we sample 60 items for human evaluation using a stratified approach to ensure comprehensive coverage: (1) 20 items with one sample per task to ensure full task coverage, (2) 20 items with two additional samples from each of the 10 models to balance model representation, and (3) 20 items randomly sampled from the remaining pairs. This ensures all tasks and all models are represented in the evaluation set. The same 60-item set is used consistently across all annotators.

**Annotation Setup**   We recruit 15 annotators via Prolific, organized into 3 batches of 5 annotators each, following standard annotation practice with small expert panels and strict agreement thresholds (Snow et al., 2008; Jamison & Gurevych, 2015). Each batch evaluates 20 items (one-third of the 60-item set), with each annotator completing exactly one batch to reduce fatigue. This results in 5 independent annotations per item and 300 total annotations.

**Annotation Task**   For each item, annotators answer: *"Does the model reasoning trace correctly capture the same key effect(s) as the verified ground-truth explanation, without contradictions or missing essential information?"* Response options are Yes/No, with an optional short explanation if No is selected. Annotators are blinded to model identity.

**Quality Control**   Each batch includes two control tasks with clearly incorrect explanations, where the expected answer is No. These control items are excluded from the analysis. Two annotators failed the control tasks and were excluded from the analysis; all LLM judges correctly identified the control tasks.

The two control tasks are:

1. **RestaurantValentinesDayOrders (quantitative error):**

   *Context:* "Background: This is the hourly number of food orders at a popular downtown restaurant. Constraints: None. Scenario: Suppose that Valentine's Day falls on 2024-02-14, and the restaurant expects 3 times the usual number of orders from 17:00:00 to 23:00:00 (6 hours)."

   *Ground Truth Reasoning:* "The effect of the context on the forecast would be that the forecasted values for 6 hours, from 2024-02-14 17:00:00 to 2024-02-14 23:00:00, would be 3 times the usual value (i.e. multiplied by 3), to account for the Valentine's Day rush."

   *Model Reasoning (Incorrect):* "The effect of the context on the forecast would be that the forecasted values for 6 hours, from 2024-02-14 17:00:00 to 2024-02-14 23:00:00, would be 2 times the usual value (i.e. multiplied by 2), to account for the Valentine's Day rush."

The context states 3 times, but the model reasoning incorrectly says 2 times. This tests whether annotators catch quantitative mismatches.

2. **GymClosedChristmasHoliday (directional error):**

*Context:* "Background: This is the daily number of gym members checking in at FitLife Fitness Center. Constraints: None. Scenario: The gym is closed for the Christmas holiday from 2024-12-24 to 2024-12-26 (3 days)."

*Ground Truth Reasoning:* "The effect of the context on the forecast would be that the forecasted values for 3 days, from 2024-12-24 to 2024-12-26, would be zero, to account for the gym being closed during the Christmas holiday."

*Model Reasoning (Incorrect):* "The effect of the context on the forecast would be that the forecasted values for 3 days, from 2024-12-24 to 2024-12-26, would be significantly higher than usual, to account for people exercising more during the holiday season."

The context states the gym is closed (check-ins should be zero), but the model reasoning incorrectly claims check-ins would be higher. This tests whether annotators catch directional errors.

All annotators correctly identified both control tasks as incorrect.

Participants were compensated 4.5 pounds for an estimated 30-minute task, with a total study cost of 102.85 pounds including platform fees. We release the complete set of 60 samples and all participant annotations to support reproducibility and future research.

### C.3.3   Screenshots

Screenshots of the Prolific human evaluation interface are provided in Figure 8 and Figure 9.

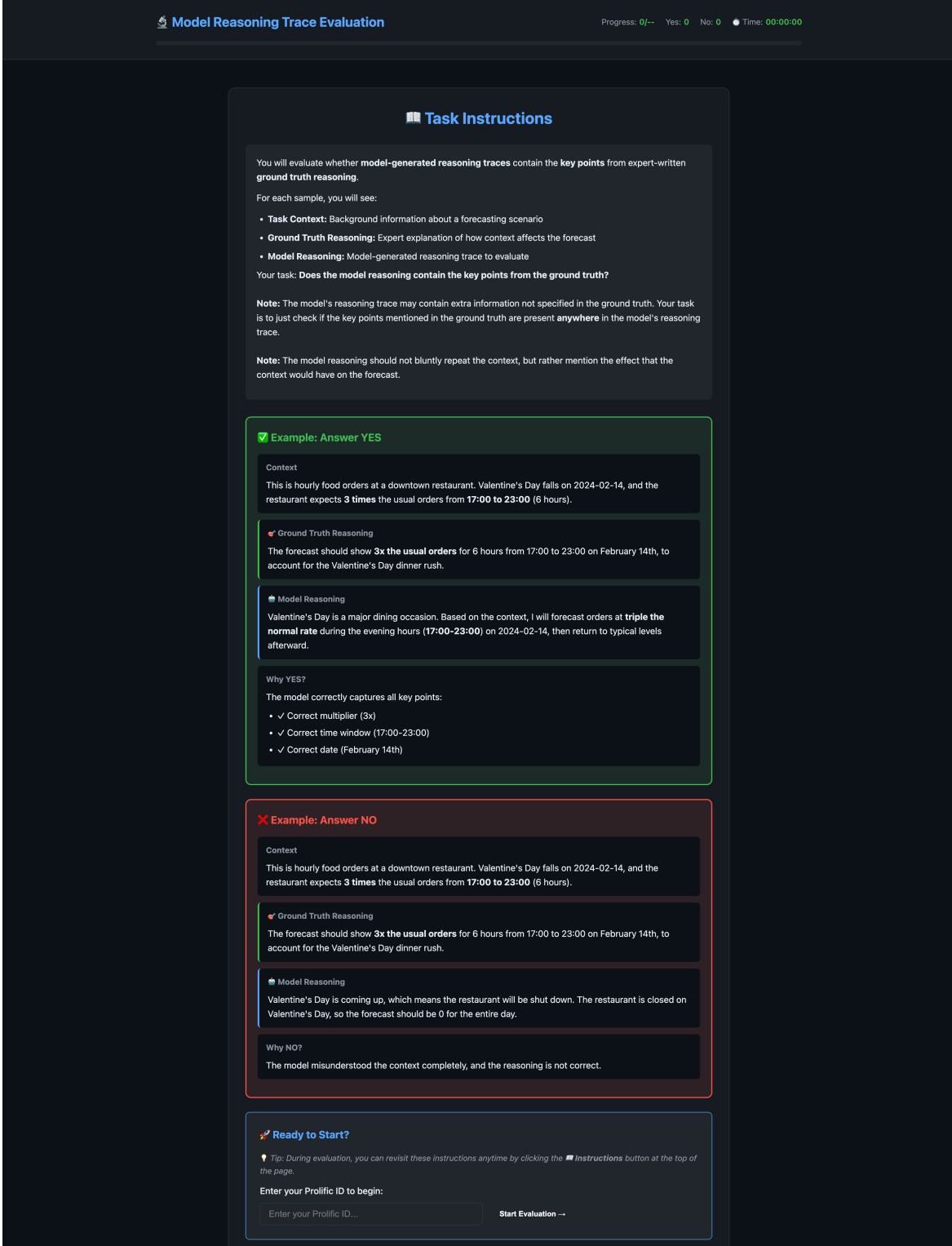

Figure 8: Welcome Page for the Prolific Human Evaluation

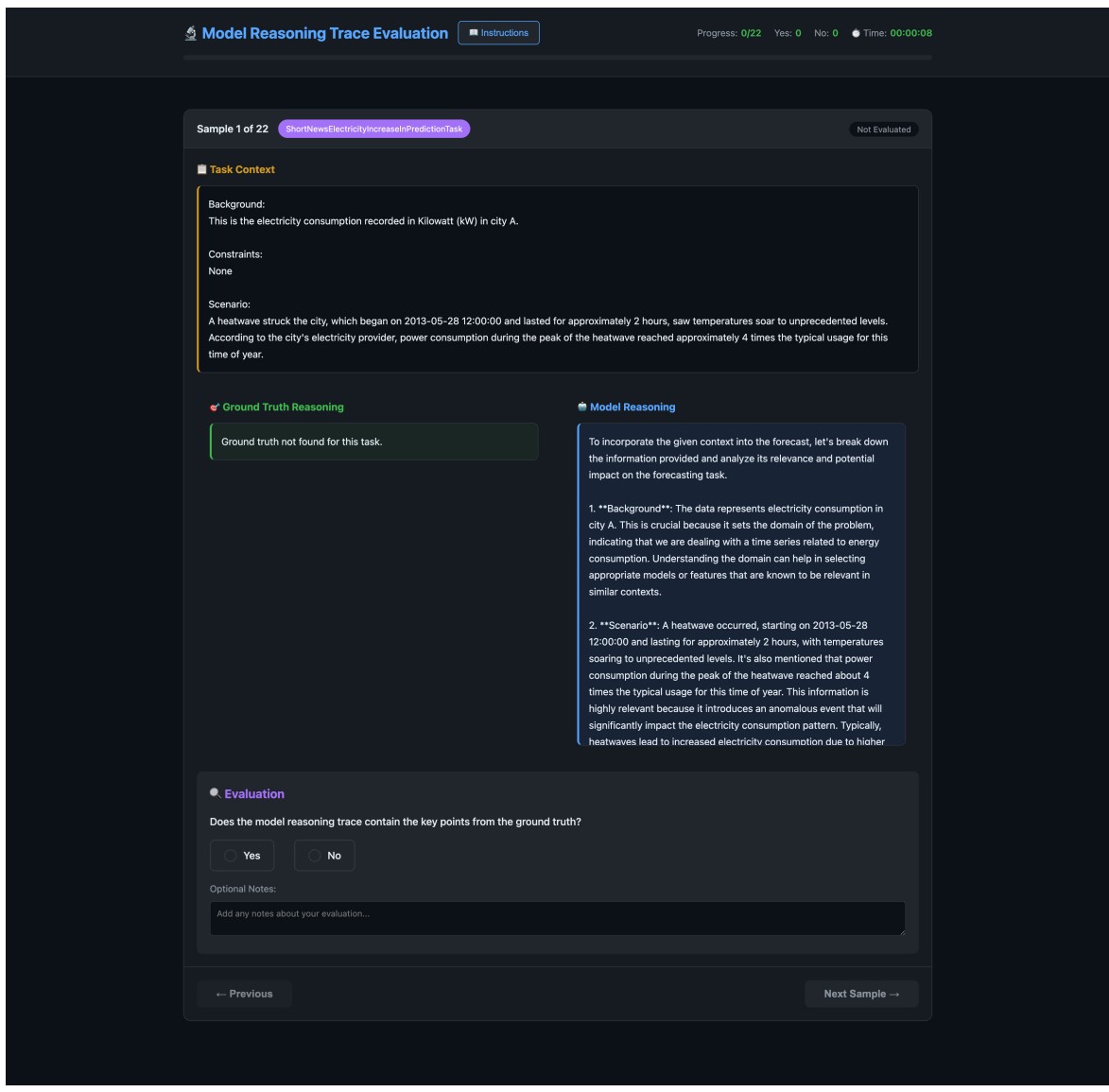

Figure 9: Page shown for each Task in the Human Evaluation

### C.3.4 Analysis

**LLM Judge Validation Against Human Evaluators** To validate that LLM judges can reliably evaluate forecast effect explanations, we compare their judgments against human majority votes on the 60-item human evaluation subset. Figure 10 shows that all three LLM judges achieve substantial to almost perfect agreement with human evaluators: GPT-5.2 and Claude Sonnet 4.5 both reach 91.7% agreement (Cohen's $\kappa = 0.81$), while Gemini 2.5 Pro achieves 93.3% agreement ($\kappa = 0.84$). Results are also given in Table 5. These results demonstrate that LLM judges can effectively replicate human evaluation judgments, enabling scalable assessment across the full model-task set.

| Judge | Samples | Agreement | Cohen's $\kappa$ | 95% CI | Human YES | LLM YES |
|---|---|---|---|---|---|---|
| GPT-5.2 | 60 | 91.7% | 0.810 | [0.711–0.909] | 68.3% | 66.7% |
| Claude Sonnet 4.5 | 60 | 91.7% | 0.810 | [0.711–0.909] | 68.3% | 66.7% |
| Gemini 2.5 Pro | 60 | 93.3% | 0.841 | [0.743–0.939] | 68.3% | 71.7% |
| **LLM Majority** | **60** | **98.3%** | **0.962** | **[0.864–1.000]** | **68.3%** | **66.7%** |

Table 5: LLM Judge vs Human Evaluation Agreement (60 Samples)

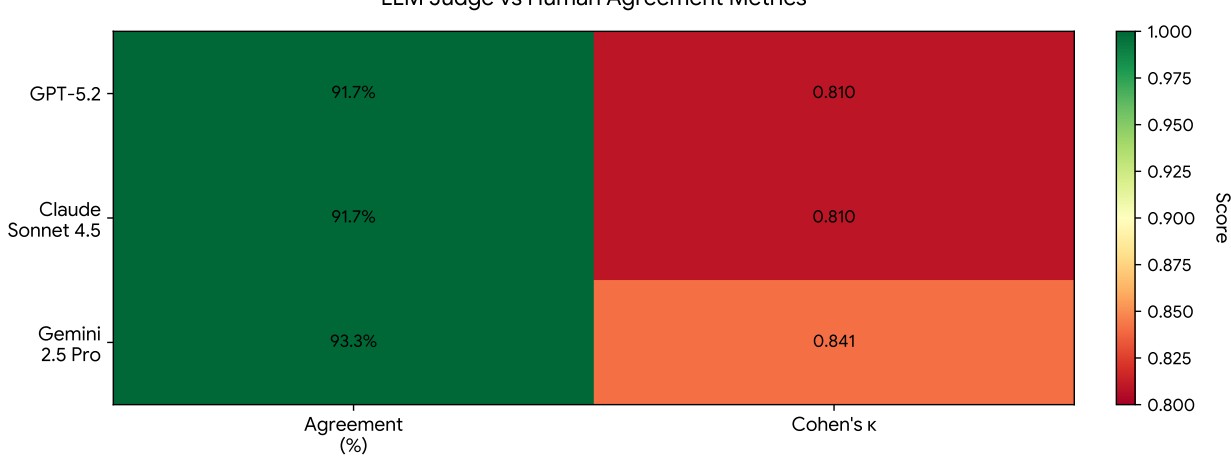

Figure 10: LLM judge validation against human evaluators (n=60). All three judges show substantial to almost perfect agreement with humans: GPT-5.2 and Claude Sonnet 4.5 both achieve 91.7% agreement (Cohen's $\kappa = 0.81$), while Gemini 2.5 Pro achieves 93.3% agreement ($\kappa = 0.84$). The high Cohen's $\kappa$ values (0.81–0.84) account for base rate differences (68.3% human YES rate) and confirm that agreement is well above chance, with confidence intervals (95% CI) ranging from [0.71–0.91] to [0.74–0.94]. These results validate the use of LLM judges for scalable evaluation of forecast effect explanations across the full 180-item model-task set.

**Inter-Annotator Agreement** We assess inter-annotator reliability on the 60-item human evaluation subset. As shown in Figure 11, the vast majority of items (95%) achieve at least 4 out of 5 annotator agreement. This high task-level consensus demonstrates that the ground truth forecast effects are unambiguous and that annotators can reliably verify whether model explanations match the specified criteria (time windows, direction, and magnitude).

**Model Performance from Human Evaluations** Beyond validating the LLM judges, the human evaluation subset provides direct insight into model explanation quality. Figure 12 shows the percentage of correct explanations (based on majority vote) for each tested model. Larger models consistently produce higher-quality explanations: Qwen 14B-32B and Llama 70B-405B achieve 75–89% correctness, while smaller models struggle significantly (Qwen 7B: 50%, Llama 3B: 22%). These human-verified results corroborate the

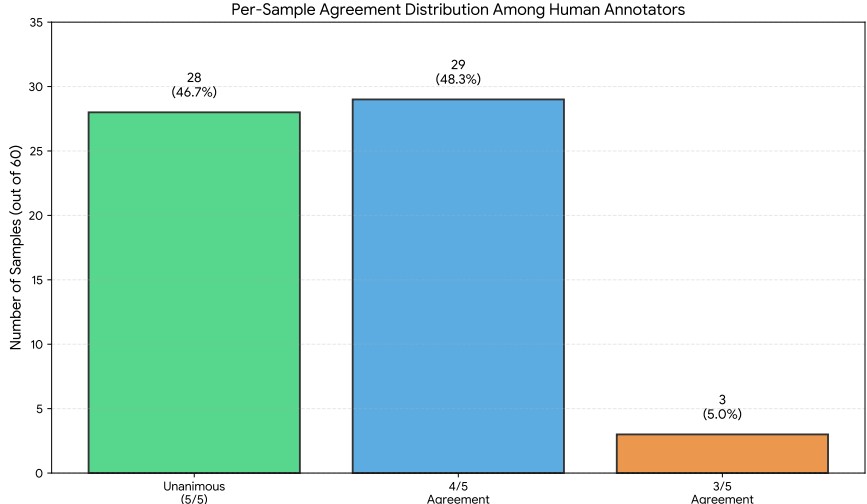

Figure 11: Distribution of per-sample agreement among human annotators (n=60). The vast majority of items (95%, CI: [86.3%–98.3%]) achieve at least 4/5 agreement, with 28 items (46.7%) achieving unanimous (5/5) agreement and 29 items (48.3%) achieving 4/5 agreement. Only 3 items (5%) have 3/5 agreement (split decisions). The high task-level consensus (average 88.3% agreement per sample) indicates that the binary verification task has clear, objective criteria. While traditional inter-annotator metrics (Fleiss' $\kappa$ = 0.502, moderate) reflect differences in annotator overall tendencies (YES rates range 40–90%), the per-sample agreement directly measures task-level reliability and demonstrates strong consensus on individual evaluation decisions.

LLM judge assessments and confirm that explanation quality scales with model size, though with diminishing returns at the frontier.

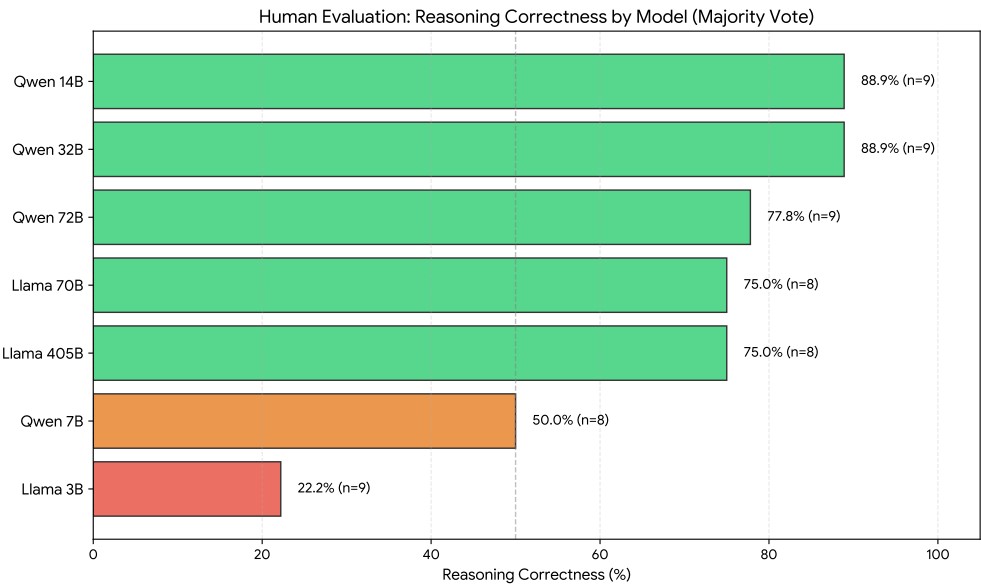

Figure 12: Reasoning correctness by model based on human majority vote (n=60 subset, 7 models). Larger models achieve substantially higher correctness: Qwen 14B-32B (89%), Llama 70B-405B (75%), and Qwen 72B (78%) demonstrate strong explanation quality, while smaller models show marked declines (Qwen 7B: 50%, Llama 3B: 22%). These human-verified results confirm that explanation quality scales with model capacity, consistent with the LLM judge assessments on the full set.

**Inter-LLM Judge Agreement** To assess whether the evaluation criteria are well-defined and consistently interpretable, we analyze agreement among the three LLM judges on the 60-item human evaluation subset. As shown in Figure 13, all judge pairs achieve substantial agreement: GPT-5.2 and Claude Sonnet 4.5 show the highest pairwise agreement (86.7%, Cohen's $\kappa = 0.70$), while both pairs involving Gemini 2.5 Pro achieve 85.0% agreement ($\kappa = 0.65$). The judges reach unanimous agreement on 78.3% of items (Fleiss' $\kappa = 0.666$), with only 21.7% showing 2/3 split decisions. These high agreement rates demonstrate that the evaluation protocol provides clear, consistent criteria that different LLM judges interpret similarly.

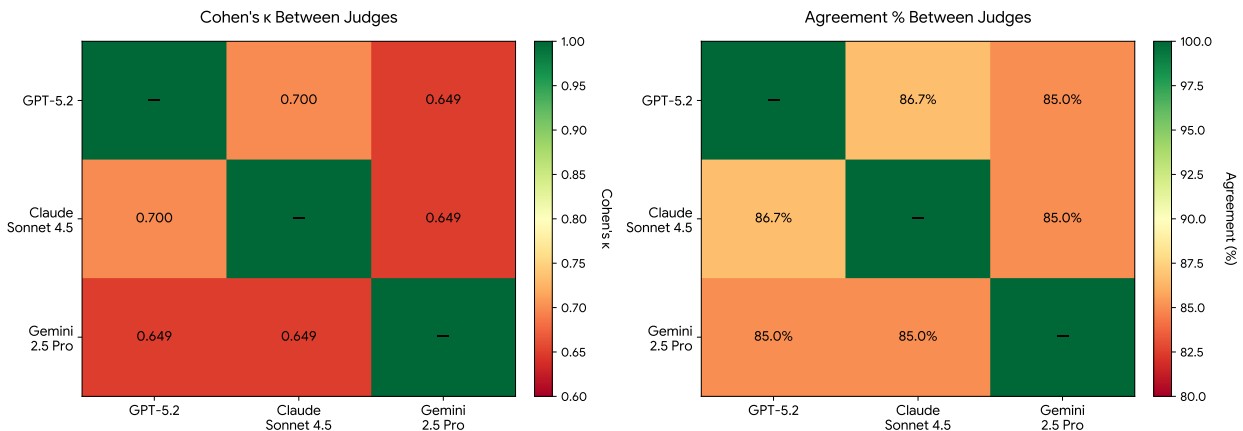

Figure 13: Pairwise inter-LLM judge agreement (n=60) showing Cohen's $\kappa$ (left) and percentage agreement (right). All judge pairs achieve substantial agreement: GPT-5.2 vs Claude Sonnet 4.5 (86.7%, $\kappa = 0.70$), GPT-5.2 vs Gemini 2.5 Pro (85.0%, $\kappa = 0.65$), and Claude Sonnet 4.5 vs Gemini 2.5 Pro (85.0%, $\kappa = 0.65$). The judges reach unanimous agreement (3/3) on 78.3% of items (35 unanimous YES, 12 unanimous NO) with a Fleiss' $\kappa$ of 0.666 (substantial), confirming that the evaluation criteria are well-defined. Individual judge YES rates are similar: GPT-5.2 (66.7%), Claude Sonnet 4.5 (66.7%), and Gemini 2.5 Pro (71.7%), indicating consistent application of standards across judges.

**Failure Mode Analysis** We present here an analysis of the cases where LLM judges marked explanations as incorrect reveals systematic patterns in model limitations. The most common failure type is missing quantitative details (34% of failures), where models understand the context qualitatively but fail to specify exact multipliers or magnitudes (e.g., stating "significantly higher" instead of "4 times"). The second most prevalent issue is maintenance data handling (28% of failures), where even frontier models often miss the critical methodological step of excluding historical maintenance periods when fitting trend and seasonality patterns, instead only noting that future predictions should not include maintenance. Smaller models (3B-7B) additionally exhibit fundamental interpretation errors (25–50% of their failures), including percentage calculation mistakes and incorrect time period identification. Interestingly, frontier models show a systematic gap in statistical preprocessing methodology rather than comprehension, with 40–78% of their failures related to historical data treatment/exclusion. These patterns suggest targeted improvement opportunities: small models need better core reasoning capabilities, mid-sized models need quantitative precision training, and large models would benefit from instruction tuning on time series methodology and definitive (non-hedging) language. We release all LLM judge evaluations along with their detailed explanations for each decision to support future analysis of model failure modes and development of targeted training approaches for context-aided forecasting tasks.

**Summary of Forecast Effect Explanation Evaluation** Our comprehensive evaluation protocol, validated through both LLM judges and human annotators, yields several key findings:

- **LLM judges are reliable:** All three LLM judges (GPT-5.2, Claude Sonnet 4.5, Gemini 2.5 Pro) show substantial agreement with human evaluators (91.7–93.3%, Cohen's $\kappa = 0.81$–$0.84$), validating their use for scalable evaluation across the full model-task set.

- **Human annotators show high consensus:** 95% of items achieve at least 4/5 annotator agreement, indicating the evaluation task is well-defined with clear criteria. All annotators passed attention checks, confirming annotation quality.

- **Inter-LLM agreement is high:** The three judges achieve 78.3% unanimous agreement and substantial pairwise agreement ($\kappa = 0.65$–$0.70$), demonstrating that the evaluation criteria yield consistent judgments even across different LLM architectures.

- **Results are robust to threshold choice:** Sensitivity analysis across thresholds (30%–80%) shows that while absolute performance decreases with stricter thresholds, the relative ranking of models remains consistent. The 50% threshold provides a balanced criterion that distinguishes model capabilities while focusing on substantial improvements.

- **Explanation quality scales with model size:** Larger models (>14B parameters) consistently produce accurate forecast effect explanations (85% correctness), while smaller models (<7B) struggle (15–45% correctness). This pattern holds across all thresholds.

- **The Execution Gap persists across scales:** Even when models explain correctly, they fail to improve forecasts in 10–30% of tasks depending on model size and threshold. This gap persists even for the best-performing models (GPT-5.2, Claude Sonnet 4.5), indicating a fundamental challenge in translating explanations into improved forecasts.

These findings establish FxDP as a reliable diagnostic tool for understanding where models succeed and fail in context-aided forecasting, with validation from both automated (LLM judge) and human evaluation.

## C.4 FxDP vs DP: Forecasting Accuracy Comparison

Providing explicit forecast effect explanations does not meaningfully change forecasting accuracy. The aggregate RCRPS results of FxDP are within the standard error of DP across all evaluated models, confirming that FxDP's value lies in diagnosis rather than accuracy improvement. This is consistent with the chain-of-thought literature, where reasoning traces do not always translate to accuracy gains without additional training (Wei et al., 2022).

| Model | DP | FxDP |
|---|---|---|
| Llama-3.2-3B | $0.687 \pm 0.025$ | $0.694 \pm 0.029$ |
| Qwen-2.5-3B | $0.424 \pm 0.017$ | $0.431 \pm 0.011$ |
| Qwen-2.5-7B | $0.401 \pm 0.006$ | $0.404 \pm 0.001$ |
| Qwen-2.5-14B | $0.247 \pm 0.006$ | $0.250 \pm 0.002$ |
| Qwen-2.5-32B | $0.397 \pm 0.008$ | $0.401 \pm 0.001$ |
| Llama-3.3-70B | $0.230 \pm 0.006$ | $0.228 \pm 0.009$ |
| Qwen-2.5-72B | $0.202 \pm 0.009$ | $0.207 \pm 0.007$ |
| Llama-3.1-405B | $0.173 \pm 0.003$ | $0.175 \pm 0.005$ |
| GPT-5.2 | $0.271 \pm 0.001$ | $0.272 \pm 0.001$ |
| Gemini-2.5-Pro | $0.108 \pm 0.002$ | $0.109 \pm 0.004$ |
| Claude-Sonnet-4.5 | $0.114 \pm 0.001$ | $0.114 \pm 0.008$ |

Table 6: Aggregate RCRPS comparison of DP and FxDP (mean ± standard error; lower is better). FxDP results are within the standard error of DP across all models.

## C.5 Improvement Threshold Ablation

To assess robustness to the choice of improvement threshold, we evaluate model performance across six threshold values (30%, 40%, 50%, 60%, 70%, 80% relative CRPS improvement in the Region of Interest). As shown in Figures 14 and 15, both overall performance and the Execution Gap vary with threshold. Performance (correct explanation + improved forecast) decreases as the threshold increases, but the relative

ranking of models remains consistent across all thresholds. Frontier models (GPT-5.2, Claude Sonnet 4.5) maintain the highest success rates even at stringent thresholds (80%), while small models (3B) show minimal success even at lenient thresholds (30%). Critically, the Execution Gap persists across all thresholds, confirming that it is not an artifact of the 50% threshold choice used in the main text.

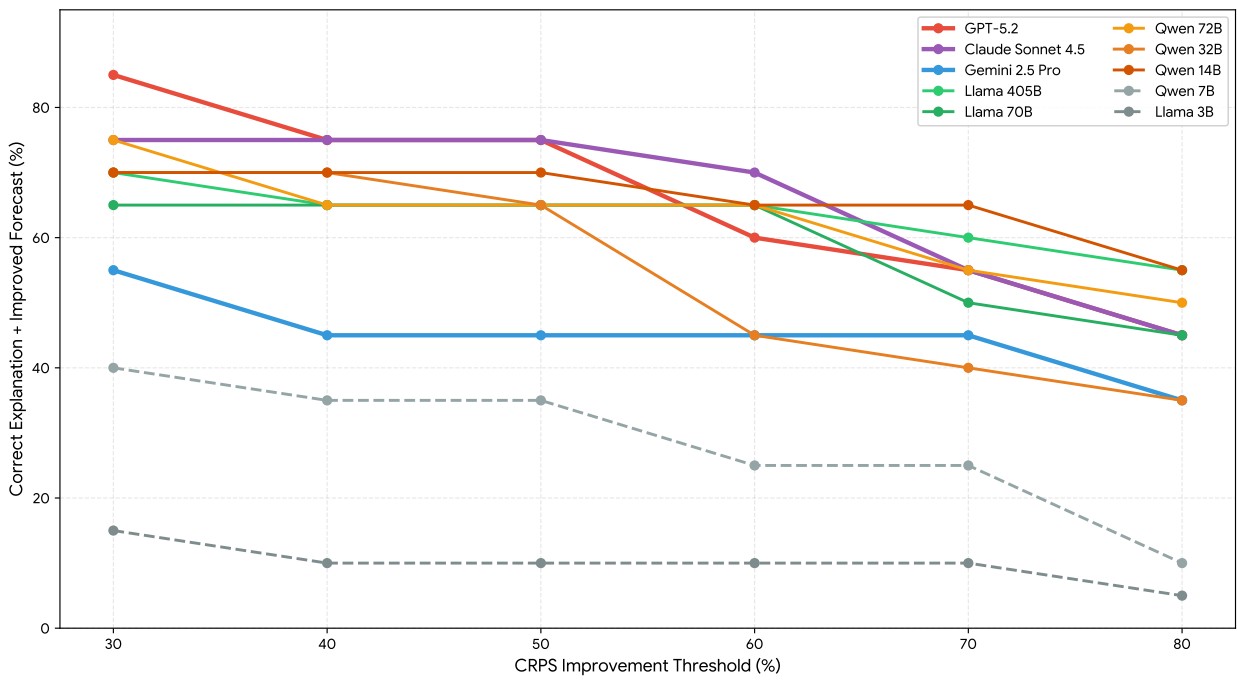

Figure 14: Sensitivity analysis: Percentage of tasks achieving both correct forecast effect explanation and improved forecast across different improvement thresholds (30%–80%). Performance decreases as the threshold increases (frontier models drop from ∼75% at 30% to ∼50% at 80%), but the relative ranking of models remains stable across all thresholds. Frontier models (GPT-5.2, Claude Sonnet 4.5, Llama 405B) maintain the highest performance at all thresholds, while small models (Llama 3B, Qwen 3B-7B) show minimal success even at the most lenient 30% threshold. Mid-sized models (Qwen 14B-72B, Llama 70B) occupy the middle ground. This consistency across thresholds validates that the 50% threshold used in the main text represents a balanced choice for identifying meaningful forecast improvements.

Figure 15: Execution Gap sensitivity analysis: Percentage of tasks with correct explanations but insufficient forecast improvement across different thresholds (30%–80%). The Execution Gap (correct explanation without adequate improvement) persists across all models and thresholds, demonstrating that it is not an artifact of threshold choice. The gap is most pronounced for mid-sized models (Qwen 32B-72B, Llama 70B) and varies with threshold stringency: at lenient thresholds (30%), gaps are smaller (0–15%), while at stringent thresholds (80%), gaps increase substantially (20–55%) as models produce correct explanations but fail to meet higher improvement criteria. Even frontier models (GPT-5.2, Claude Sonnet 4.5) show persistent gaps of 10–35% depending on threshold. Gemini 2.5 Pro exhibits consistently high gap rates (15–30%) across all thresholds, confirming its tendency to explain correctly without translating to proportional forecast improvements.

## C.6 Examples of Accurate and Inaccurate Forecast Effect Explanations

We present three examples of unanimous agreement (YES) and three of unanimous agreement (NO) from the LLM judge evaluations (GPT-5.2, Claude-Sonnet-4.5, Gemini-2.5-Pro), including each judge's explanation.

### C.6.1 Unanimous YES

**Example 1: *ExplicitTrafficForecastTaskwithHolidaysInPredictionWindow*, Llama3.2-3B-Inst**

*Context:* Road occupancy rates on a freeway in the San Francisco Bay area. 2024-07-04 is Independence Day; traffic typically reduces on holidays.

*GT forecast effect:* The forecasted value for 2024-07-04 would be lower than the usual value, to account for the reduction in traffic due to the holiday.

*Model forecast effect (verbatim):*

> To incorporate the context into the time series forecasting task, we need to consider the following factors:
>
> 1. **Holidays**: The context mentions that 2024-07-04 is a holiday due to Independence Day. This typically reduces traffic on the freeway. We should expect lower occupancy rates on this day.

2. **Day of the week**: The context does not provide explicit day of the week information. However, we can infer that the historical data is mostly from the end of June, which is a weekday. This information can help us identify patterns in the data, but we should be cautious not to overemphasize it.

3. **Background knowledge**: The context provides general information about the freeway and the region, which can help us understand the underlying factors that affect the occupancy rates. However, we should be aware that this information may not directly impact the forecasting task.

4. **Seasonality**: The historical data does not provide explicit seasonal patterns, but we can observe that the occupancy rates seem to increase as the day progresses and then decrease at the end of the day. This could be a seasonal pattern.

*Judge explanations:* **GPT-5.2:** The model reasoning identifies July 4th as a holiday and explicitly expects lower freeway occupancy/traffic that day, matching the ground truth's key adjustment. **Claude-Sonnet-4.5:** The model correctly identifies that 2024-07-04 is a holiday (Independence Day), recognizes that this typically reduces traffic on the freeway, and explicitly states lower occupancy rates should be expected; this captures the key point from the ground truth. **Gemini-2.5-Pro:** The model correctly identifies that July 4th is a holiday, which leads to reduced traffic and therefore lower expected occupancy rates, matching the key points of the ground truth.

### Example 2: *ElectricityIncreaseInPredictionWithDistractorWithDates*, Llama3.2-3B-Inst

*Context:* Electricity consumption in city A. A festival in neighboring cities B and C caused $10\times$ consumption there (does not affect city A). A heat wave in city A from 2013-05-28 12:00:00 for 2 hours leads to $4\times$ usual consumption.

*GT forecast effect:* Forecasted values for 2013-05-28 12:00:00 and 13:00:00 would be 4 times the usual value (multiplied by 4) due to the heat wave.

*Model forecast effect (verbatim):*

Incorporating the context into the forecasting task, we need to consider the unusual increase in electricity consumption in neighboring cities B and C due to the festival, which should not affect city A. However, there is a heat wave in city A from 2013-05-28 12:00:00 for 2 hours, leading to excessive use of air conditioning, which should increase electricity consumption.

To incorporate this context, we can create a separate time series for the heat wave event and account for its impact on the electricity consumption. The heat wave event is represented by the hourly electricity consumption from 12:00:00 to 14:00:00 on May 28, which is 4 times the usual electricity being consumed.

We can also use this information to adjust the model for the normal hourly electricity consumption, allowing the model to learn from the historical data and capture the normal patterns and trends.

Additionally, the festival in neighboring cities B and C should not affect city A, so we need to account for this by excluding these data points from the model. The festival is represented by the hourly electricity consumption from 12:00:00 to 14:00:00 on May 28, which is 10 times the usual electricity being consumed.

*Judge explanations:* **GPT-5.2:** It identifies that city A's load should be $4\times$ normal during 2013-05-28 12:00–14:00 due to the heat wave, and notes the neighboring-city festival should not affect city A. **Claude-Sonnet-4.5:** The model correctly identifies the heat wave in city A for 2 hours should result in 4 times the usual consumption, and correctly notes the festival in B and C does not affect city A; it captures the essential forecast adjustment. **Gemini-2.5-Pro:** The model correctly identifies that the heat wave will cause a $4\times$ increase for the specified 2-hour period and correctly dismisses the festival in other cities as irrelevant.

### Example 3: *CashDepletedinATMScenarioTask*, Qwen-2.5-32B-Inst

*Context:* Cash withdrawals from an ATM in England. Cash is depleted from 1996-11-24 00:00:00 for 10 days, resulting in no withdrawals.

*GT forecast effect:* Forecasted values for 10 days (1996-11-24 to 1996-12-03 inclusive) would be zero.

*Model forecast effect (verbatim):*

> To incorporate the provided context into the forecast, we need to take several steps:
>
> 1. **Background Information**: The data represents cash withdrawals from an ATM. The variability in the number of withdrawals can be influenced by many factors such as business days, weekends, holidays, or unusual events that affect withdrawal patterns.
>
> 2. **Scenario**: There's a specific scenario where the ATM was empty from 1996-11-24 to 1996-12-03, meaning no withdrawals were recorded during this period. We must consider this as a significant disruption to the usual pattern of withdrawals.
>
> - For the days when the ATM was empty (1996-11-24 to 1996-12-03), the forecast will show zero withdrawals. - After the ATM is replenished, we can expect a return to normal withdrawal levels. However, there might be an immediate spike in withdrawals as users who had been unable to withdraw make up for missed transactions. This could mean that the first few days after the empty period might see higher-than-normal withdrawal numbers.
>
> 3. **Historical Data Analysis**: To forecast accurately beyond the empty period, it's essential to look at the typical withdrawal behavior before and after similar disruptions if they exist in the history. Since there are no previous disruptions noted, we can infer based on general behavior around the dates immediately preceding the empty period.
>
> Therefore, the effect of the context on the forecast would be that the withdrawal counts for the period from 1996-11-24 to 1996-12-03 will be predicted as zero. Following this period, we should anticipate potentially elevated withdrawal counts due to catch-up behavior, followed by a return to normal levels based on prior patterns.

*Judge explanations:* **GPT-5.2:** It states withdrawals should be zero from 1996-11-24 to 1996-12-03 due to the ATM being empty, matching the ground truth's key point. **Claude-Sonnet-4.5:** The model correctly identifies that the forecast should show zero withdrawals for the period from 1996-11-24 to 1996-12-03 due to the ATM being empty; this matches the key point from the ground truth. **Gemini-2.5-Pro:** The model's reasoning correctly identifies that the forecast for the 10-day period starting from 1996-11-24 should be zero due to cash depletion, which is the core instruction from the ground truth.

### C.6.2 Unanimous NO

**Example 1:** *CashDepletedinATMScenarioTask*, **Llama3.2-3B-Inst**

*Context:* Cash withdrawals from an ATM in England. Cash is depleted from 1996-11-24 00:00:00 for 10 days, resulting in no withdrawals.

*GT forecast effect:* Forecasted values for 10 days (1996-11-24 to 1996-12-03 inclusive) would be zero.

*Model forecast effect (verbatim):*

> Walk-through step-by-step how I would incorporate each piece of the context to improve the forecast:
>
> 1. Background context: The ATM cash is depleted from 1996-11-24 00:00:00 for 10 days. This information is crucial for identifying a seasonal trend in the historical time series data. By considering the depletion of cash, the forecast can be improved by accounting for the temporary absence of cash, which can lead to a reduction in cash withdrawals for a short period.
>
> 2. Historical time series data: This data serves as the basis for the forecasting model. It provides information on the trend, seasonality, and patterns in the cash withdrawal data. By analyzing the historical data, the forecasting model can learn to recognize and adapt to these patterns.
>
> 3. Irrelevant context: The specific location in England (i.e., "arbitrary location in England") is not relevant to the forecasting task. The geographical location may impact the cash withdrawal patterns due to local economic and demographic factors, but it is not necessary to incorporate this information for this specific forecasting task.

4. Seasonal trend: The historical data shows a seasonal trend, with higher cash withdrawals during certain periods of the year (e.g., summer and late fall). This trend should be considered when creating the forecasting model to improve the accuracy of the forecast.

5. Temporary absence of cash: The temporary absence of cash during the depletion period (1996-11-24 00:00:00 to 1996-11-30 00:00:00) should be accounted for in the forecasting model. This absence of cash can lead to a temporary reduction in cash withdrawals, which should be reflected in the forecast.

6. Irrelevant context: The background information on the background context is not relevant to the forecasting task. The focus should be on analyzing the historical data and creating a forecasting model that accounts for the temporary absence of cash and seasonal trends.

7. Time series forecasting techniques: Various time series forecasting techniques, such as ARIMA, SARIMA, or machine learning models (e.g., LSTM, GRU), can be used to create a forecasting model that incorporates the seasonal trends and temporary absence of cash. These techniques can learn to recognize patterns in the historical data and adapt to new data.

*Judge explanations:* **GPT-5.2:** The model notes a "temporary reduction" and uses the wrong end date (1996-11-30), but does not state withdrawals must be zero for the full 10-day period 1996-11-24 to 1996-12-03 inclusive. **Claude-Sonnet-4.5:** The model incorrectly identifies the depletion period as ending on 1996-11-30 instead of 1996-12-03 (only 7 days); it mentions "temporary reduction" but does not explicitly state the forecast should be zero; the ground truth's essential instruction (zero for 10 days) is missing. **Gemini-2.5-Pro:** The ground truth states the forecast should be zero for 10 days. The model is imprecise, only mentioning a "reduction," and incorrectly specifies the depletion ends on 1996-11-30 (only 6 days, not 10).

**Example 2:** *ExplicitWithDatesAndDaysTrafficForecastTaskwithHolidaysInPredictionWindow*, **Llama3.2-3B-Inst**

*Context:* Road occupancy rates on a freeway. Forecast required for Thursday 2024-07-04, Friday 2024-07-05, Saturday 2024-07-06. 2024-07-04 is Independence Day; traffic typically reduces on holidays.

*GT forecast effect:* The forecasted value for 2024-07-04 would be lower than the usual value due to the holiday.

*Model forecast effect (verbatim):*

> First, I will incorporate the historical data into my model. The historical data will be used to train and test my forecasting model.

*Judge explanations:* **GPT-5.2:** The model reasoning does not mention the holiday effect or that July 4th traffic/occupancy should be lower than usual. **Claude-Sonnet-4.5:** The model reasoning only describes a generic approach of using historical data for training and testing, without mentioning the critical holiday effect; the ground truth specifically explains that forecasted values for July 4th should be lower than usual due to Independence Day. **Gemini-2.5-Pro:** The model's reasoning is entirely generic and does not mention the key information: that the forecast for July 4th should be lower than usual due to the holiday.

**Example 3:** *ElectricityIncreaseInPredictionWithDistractorText*, **Llama3.2-3B-Inst**

*Context:* Electricity consumption in city A. Heat wave from 2013-05-28 12:00:00 for 2 hours leads to 4× usual consumption. A technical issue 2 weeks ago caused a 75% dip (not expected to recur).

*GT forecast effect:* Forecasted values for 2013-05-28 12:00:00 and 13:00:00 would be 4 times the usual value.

*Model forecast effect (verbatim):*

> The historical data provided is in the format of (timestamp, value) and represents the electricity consumption in city A.
>
> To improve the forecast, the context about the heat wave from 2013-05-28 12:00:00 for 2 hours and the excessive use of air conditioning should be incorporated. This can be done by identifying the

time period when the heat wave occurred and accounting for the increased electricity consumption during this time.

Additionally, the technical issue in the electricity grid that occurred 2 weeks ago should be taken into consideration. This could have led to a decrease in electricity consumption, and we should ensure that this trend is not repeated this week. However, the context states that the issue is not expected to happen again this week, so we can assume that the trend is not relevant for our forecast.

The effect of the context on the forecast would be that the forecast is likely to be affected by an increase in electricity consumption due to the heat wave, but there is a possibility that the forecast may remain stable if the technical issue does not occur again this week.

*Judge explanations:* **GPT-5.2:** The model notes an increase from the heat wave but does not specify that only the two hours (12:00 and 13:00) should be multiplied by 4, which is the key ground-truth adjustment. **Claude-Sonnet-4.5:** The model identifies that the heat wave will increase consumption but fails to specify that values should be multiplied by exactly 4 for the specific timestamps; it only vaguely mentions "accounting for increased electricity consumption" without the precise $4\times$ multiplier needed. **Gemini-2.5-Pro:** The model fails to mention the key quantitative detail: that consumption will be "4 times" the usual value. It only vaguely mentions an "increase."

# D    Additional Details on CorDP

## D.1    CorDP Prompt

We use the following prompt for the CorDP method, where **{base_forecasts}** are replaced by the forecasts of the quantitative forecaster in the format: (timestep1, value1), (timestep2, value2), ... (timestepN, valueN) where N is the prediction length. **{history}** is replaced by the respective numerical history for the task instance in the format (timestamp, value), **{context}** is replaced by the respective textual context for the task instance, and **((pred_time))** is replaced with the prediction timesteps.

For Median-CorDP, the **{base_forecasts}** part is replaced with the median forecast from the base forecaster, and the LLM is sampled 25 times independently to get the forecast distribution.

For SampleWise-CorDP, we perform inference 25 times independently, each time providing a different sample from the base forecaster. Because we sample from the LLM exactly once per forward pass, just as we do for the 25 independent passes in Median-CorDP. Thereby, both methods incur the identical computational and token cost.

```
I have a time series forecasting task for you.

Here is some context about the task. Make sure to factor in any background knowledge,
satisfy any constraints, and respect any scenarios.
<context>
{context}
</context>

Here is a historical time series in (timestamp, value) format:
<history>
{history}
</history>

And these are the forecasts of my statistical forecasting model in (timestamp, value) format:
<base_forecast>
{base_forecasts}
</base_forecast>

My statistical forecasting model does not support taking in context as part of its input. I
    would like you to correct its forecasts to incorporate the context wherever necessary, and
    return the corrected context-aware forecast.
Return the corrected forecast in (timestamp, value) format in between <corrected_forecast> and
    </corrected_forecast> tags.
Do not include any other information (e.g., comments) in the forecast.
```

| Model | Direct Prompt (DP) | Median Corrector (Median-CorDP) | | | SampleWise Corrector (SampleWise-CorDP) | | |
|---|---|---|---|---|---|---|---|
| | | LAG-LLAMA | CHRONOS LARGE | ARIMA | LAG-LLAMA | CHRONOS LARGE | ARIMA |
| Qwen2.5-0.5B-Inst | 0.339 ± 0.010 | 0.302 ± 0.001 | 0.553 ± 0.000 | 0.336 ± 0.001 | **0.235 ± 0.006** | 0.438 ± 0.014 | 0.272 ± 0.004 |
| Qwen2.5-1.5B-Inst | 0.317 ± 0.020 | 0.296 ± 0.002 | 0.538 ± 0.005 | 0.323 ± 0.002 | **0.232 ± 0.005** | 0.478 ± 0.007 | 0.278 ± 0.006 |
| Qwen2.5-3B-Inst | 0.269 ± 0.015 | 0.391 ± 0.004 | 0.420 ± 0.004 | 0.274 ± 0.005 | **0.219 ± 0.005** | 0.388 ± 0.008 | 0.243 ± 0.004 |
| Qwen2.5-7B-Inst | 0.285 ± 0.006 | **0.125 ± 0.002** | 0.198 ± 0.006 | 0.182 ± 0.004 | 0.135 ± 0.004 | 0.180 ± 0.006 | 0.146 ± 0.004 |
| Qwen2.5-14B-Inst | **0.162 ± 0.005** | 0.288 ± 0.002 | 0.247 ± 0.002 | 0.236 ± 0.005 | 0.206 ± 0.007 | 0.221 ± 0.004 | 0.205 ± 0.005 |
| Qwen2.5-32B-Inst | **0.116 ± 0.001** | 0.213 ± 0.002 | 0.156 ± 0.002 | 0.187 ± 0.002 | 0.145 ± 0.005 | 0.132 ± 0.002 | 0.137 ± 0.005 |
| Qwen2.5-72B-Inst | **0.115 ± 0.004** | 0.158 ± 0.003 | 0.169 ± 0.002 | 0.141 ± 0.003 | 0.138 ± 0.006 | 0.140 ± 0.004 | 0.125 ± 0.004 |
| Llama-3.2-1B-Inst | 0.336 ± 0.026 | 0.281 ± 0.004 | 0.414 ± 0.013 | 0.311 ± 0.002 | **0.234 ± 0.006** | 0.507 ± 0.003 | 0.269 ± 0.005 |
| Llama-3.2-3B-Inst | 0.281 ± 0.013 | 0.243 ± 0.003 | 0.368 ± 0.006 | 0.262 ± 0.004 | **0.214 ± 0.005** | 0.362 ± 0.007 | 0.243 ± 0.004 |
| Llama-3-8B-Inst | 0.255 ± 0.008 | 0.167 ± 0.005 | 0.189 ± 0.004 | 0.164 ± 0.003 | **0.149 ± 0.005** | 0.176 ± 0.006 | 0.150 ± 0.005 |
| Llama3.3-70B-Inst | **0.105 ± 0.003** | 0.211 ± 0.001 | 0.163 ± 0.001 | 0.205 ± 0.001 | 0.164 ± 0.005 | 0.126 ± 0.003 | 0.152 ± 0.003 |
| Llama3.1-405B-Inst | 0.126 ± 0.004 | 0.212 ± 0.004 | 0.146 ± 0.003 | 0.168 ± 0.003 | 0.131 ± 0.006 | **0.117 ± 0.003** | 0.144 ± 0.003 |
| GPT-4o | 0.123 ± 0.004 | 0.212 ± 0.005 | 0.118 ± 0.001 | 0.185 ± 0.002 | 0.156 ± 0.006 | **0.108 ± 0.003** | 0.124 ± 0.002 |
| GPT-4o-mini | 0.263 ± 0.005 | 0.277 ± 0.002 | 0.270 ± 0.002 | 0.317 ± 0.001 | **0.224 ± 0.006** | 0.241 ± 0.003 | 0.255 ± 0.003 |
| GPT-5.2 | **0.104 ± 0.001** | 0.160 ± 0.001 | **0.104 ± 0.001** | 0.137 ± 0.001 | 0.119 ± 0.004 | 0.112 ± 0.002 | 0.123 ± 0.004 |
| Claude-Sonnet-4.5 | 0.108 ± 0.001 | 0.167 ± 0.001 | 0.106 ± 0.001 | 0.154 ± 0.001 | 0.134 ± 0.004 | **0.095 ± 0.001** | 0.138 ± 0.005 |
| Gemini-2.5-Pro | **0.112 ± 0.001** | 0.176 ± 0.003 | 0.128 ± 0.001 | 0.154 ± 0.001 | 0.137 ± 0.004 | **0.112 ± 0.002** | 0.126 ± 0.004 |
| Base Quantitative Forecaster | - | 0.224 ± 0.005 | 0.536 ± 0.003 | 0.272 ± 0.004 | 0.224 ± 0.005 | 0.536 ± 0.003 | 0.272 ± 0.004 |

Table 7: RoI CRPS within the context-sensitive region for partial-RoI tasks in CiK (mean ± std. err.; lower is better). SampleWise-CorDP achieves the best performance in the majority of models (especially smaller ones), while DP remains best for several larger models (e.g., Qwen2.5-14B/72B, Llama3.3-70B, frontier APIs). Best per model in **bold**.

| Model | Direct Prompt (DP) | Median Corrector (Median-CorDP) | | | SampleWise Corrector (SampleWise-CorDP) | | |
|---|---|---|---|---|---|---|---|
| | | LAG-LLAMA | CHRONOS LARGE | ARIMA | LAG-LLAMA | CHRONOS LARGE | ARIMA |
| Qwen2.5-0.5B-Inst | $0.129 \pm 0.010$ | $0.283 \pm 0.001$ | $0.142 \pm 0.000$ | $0.206 \pm 0.001$ | $0.211 \pm 0.006$ | $\mathbf{0.111 \pm 0.014}$ | $0.159 \pm 0.004$ |
| Qwen2.5-1.5B-Inst | $0.224 \pm 0.020$ | $0.268 \pm 0.002$ | $0.140 \pm 0.005$ | $0.198 \pm 0.002$ | $0.193 \pm 0.005$ | $\mathbf{0.113 \pm 0.007}$ | $0.160 \pm 0.006$ |
| Qwen2.5-3B-Inst | $0.186 \pm 0.015$ | $0.251 \pm 0.004$ | $0.129 \pm 0.004$ | $0.179 \pm 0.005$ | $0.179 \pm 0.005$ | $\mathbf{0.114 \pm 0.008}$ | $0.134 \pm 0.004$ |
| Qwen2.5-7B-Inst | $0.164 \pm 0.006$ | $0.225 \pm 0.002$ | $0.137 \pm 0.006$ | $0.182 \pm 0.004$ | $0.167 \pm 0.004$ | $\mathbf{0.127 \pm 0.006}$ | $0.146 \pm 0.004$ |
| Qwen2.5-14B-Inst | $\mathbf{0.146 \pm 0.005}$ | $0.306 \pm 0.002$ | $0.212 \pm 0.002$ | $0.219 \pm 0.005$ | $0.210 \pm 0.007$ | $0.188 \pm 0.004$ | $0.200 \pm 0.005$ |
| Qwen2.5-32B-Inst | $0.140 \pm 0.001$ | $0.238 \pm 0.002$ | $0.143 \pm 0.002$ | $0.194 \pm 0.002$ | $0.164 \pm 0.005$ | $\mathbf{0.112 \pm 0.002}$ | $0.131 \pm 0.005$ |
| Qwen2.5-72B-Inst | $\mathbf{0.138 \pm 0.004}$ | $0.265 \pm 0.003$ | $0.192 \pm 0.002$ | $0.200 \pm 0.003$ | $0.181 \pm 0.006$ | $0.155 \pm 0.004$ | $0.158 \pm 0.004$ |
| Llama-3.2-1B-Inst | $0.248 \pm 0.026$ | $0.260 \pm 0.004$ | $0.107 \pm 0.013$ | $0.191 \pm 0.002$ | $0.191 \pm 0.006$ | $\mathbf{0.104 \pm 0.003}$ | $0.159 \pm 0.005$ |
| Llama-3.2-3B-Inst | $0.162 \pm 0.013$ | $0.213 \pm 0.003$ | $0.116 \pm 0.006$ | $0.152 \pm 0.004$ | $0.177 \pm 0.005$ | $\mathbf{0.107 \pm 0.007}$ | $0.136 \pm 0.004$ |
| Llama-3-8B-Inst | $\mathbf{0.163 \pm 0.008}$ | $0.257 \pm 0.005$ | $0.238 \pm 0.004$ | $0.208 \pm 0.003$ | $0.232 \pm 0.005$ | $0.198 \pm 0.006$ | $0.189 \pm 0.005$ |
| Llama3.3-70B-Inst | $0.182 \pm 0.003$ | $0.277 \pm 0.001$ | $0.157 \pm 0.001$ | $0.194 \pm 0.001$ | $0.205 \pm 0.005$ | $\mathbf{0.132 \pm 0.003}$ | $0.154 \pm 0.003$ |
| Llama3.1-405B-Inst | $0.150 \pm 0.004$ | $0.248 \pm 0.004$ | $0.170 \pm 0.003$ | $0.174 \pm 0.003$ | $0.163 \pm 0.006$ | $\mathbf{0.133 \pm 0.003}$ | $0.141 \pm 0.003$ |
| GPT-4o | $\mathbf{0.106 \pm 0.004}$ | $0.246 \pm 0.005$ | $0.140 \pm 0.001$ | $0.190 \pm 0.002$ | $0.159 \pm 0.006$ | $0.114 \pm 0.003$ | $0.145 \pm 0.002$ |
| GPT-4o-mini | $0.150 \pm 0.005$ | $0.282 \pm 0.002$ | $0.141 \pm 0.002$ | $0.198 \pm 0.001$ | $0.198 \pm 0.006$ | $\mathbf{0.117 \pm 0.003}$ | $0.150 \pm 0.003$ |
| GPT-5.2 | $\mathbf{0.095 \pm 0.001}$ | $0.177 \pm 0.001$ | $0.131 \pm 0.001$ | $0.161 \pm 0.001$ | $0.121 \pm 0.004$ | $0.107 \pm 0.002$ | $0.137 \pm 0.004$ |
| Claude-Sonnet-4.5 | $\mathbf{0.090 \pm 0.001}$ | $0.167 \pm 0.001$ | $0.106 \pm 0.001$ | $0.154 \pm 0.001$ | $0.134 \pm 0.004$ | $0.095 \pm 0.001$ | $0.138 \pm 0.005$ |
| Gemini-2.5-Pro | $\mathbf{0.081 \pm 0.001}$ | $0.164 \pm 0.003$ | $0.102 \pm 0.001$ | $0.158 \pm 0.001$ | $0.123 \pm 0.004$ | $0.089 \pm 0.002$ | $0.122 \pm 0.004$ |
| Base Quantitative Forecaster | - | $0.202 \pm 0.005$ | $0.115 \pm 0.003$ | $0.159 \pm 0.004$ | $0.202 \pm 0.005$ | $0.115 \pm 0.003$ | $0.159 \pm 0.004$ |

Table 8: Non-RoI CRPS (outside the context-sensitive region) for partial-RoI tasks in CiK (mean ± std. err.; lower is better). CorDP variants, especially SampleWise-CorDP with Chronos, often outperform DP outside the RoI as well, indicating that correction does not harm, and can improve forecasts in context-free regions. Best per model in **bold**.

| Model | Direct Prompt (DP) | Median Corrector (Median-CorDP) | | | SampleWise Corrector (SampleWise-CorDP) | | |
|---|---|---|---|---|---|---|---|
| | | LAG-LLAMA | CHRONOS LARGE | ARIMA | LAG-LLAMA | CHRONOS LARGE | ARIMA |
| Qwen2.5-0.5B-Inst | $0.836 \pm 0.046$ | $0.864 \pm 0.003$ | $1.110 \pm 0.006$ | $1.094 \pm 0.090$ | $\mathbf{0.679 \pm 0.013}$ | $0.895 \pm 0.127$ | $0.953 \pm 0.092$ |
| Qwen2.5-1.5B-Inst | $0.851 \pm 0.026$ | $\mathbf{0.525 \pm 0.021}$ | $0.672 \pm 0.005$ | $0.969 \pm 0.011$ | $0.733 \pm 0.030$ | $0.595 \pm 0.007$ | $1.059 \pm 0.021$ |
| Qwen2.5-3B-Inst | $0.558 \pm 0.027$ | $0.606 \pm 0.008$ | $0.638 \pm 0.006$ | $0.849 \pm 0.014$ | $\mathbf{0.533 \pm 0.048}$ | $0.587 \pm 0.007$ | $0.731 \pm 0.053$ |
| Qwen2.5-7B-Inst | $\mathbf{0.521 \pm 0.009}$ | $0.584 \pm 0.006$ | $0.964 \pm 0.013$ | $0.939 \pm 0.013$ | $0.538 \pm 0.011$ | $0.571 \pm 0.034$ | $0.808 \pm 0.019$ |
| Qwen2.5-14B-Inst | $\mathbf{0.310 \pm 0.010}$ | $0.328 \pm 0.004$ | $0.406 \pm 0.009$ | $0.556 \pm 0.007$ | $0.470 \pm 0.009$ | $0.551 \pm 0.009$ | $0.654 \pm 0.015$ |
| Qwen2.5-32B-Inst | $0.580 \pm 0.013$ | $\mathbf{0.263 \pm 0.007}$ | $0.355 \pm 0.009$ | $0.423 \pm 0.013$ | $0.416 \pm 0.008$ | $0.486 \pm 0.011$ | $0.604 \pm 0.014$ |
| Qwen2.5-72B-Inst | $\mathbf{0.253 \pm 0.015}$ | $0.392 \pm 0.014$ | $0.479 \pm 0.017$ | $0.603 \pm 0.015$ | $0.320 \pm 0.016$ | $0.441 \pm 0.016$ | $0.552 \pm 0.017$ |
| Llama-3.2-1B-Inst | $0.467 \pm 0.041$ | $0.477 \pm 0.007$ | $0.687 \pm 0.008$ | $0.857 \pm 0.030$ | $0.765 \pm 0.014$ | $0.857 \pm 0.008$ | $0.983 \pm 0.025$ |
| Llama-3.2-3B-Inst | $1.004 \pm 0.040$ | $\mathbf{0.422 \pm 0.018}$ | $0.600 \pm 0.014$ | $0.821 \pm 0.037$ | $0.722 \pm 0.043$ | $0.551 \pm 0.012$ | $0.985 \pm 0.052$ |
| Llama-3-8B-Inst | $0.771 \pm 0.043$ | $\mathbf{0.385 \pm 0.006}$ | $0.615 \pm 0.008$ | $0.833 \pm 0.007$ | $0.586 \pm 0.015$ | $0.561 \pm 0.006$ | $0.953 \pm 0.016$ |
| Llama3.3-70B-Inst | $0.289 \pm 0.011$ | $0.306 \pm 0.004$ | $0.313 \pm 0.006$ | $0.456 \pm 0.010$ | $\mathbf{0.249 \pm 0.006}$ | $0.273 \pm 0.006$ | $0.419 \pm 0.011$ |
| Llama3.1-405B-Inst | $\mathbf{0.196 \pm 0.005}$ | $0.310 \pm 0.014$ | $0.272 \pm 0.006$ | $0.316 \pm 0.012$ | $0.235 \pm 0.009$ | $0.241 \pm 0.006$ | $0.288 \pm 0.013$ |
| GPT-4o | $0.455 \pm 0.014$ | $0.270 \pm 0.006$ | $0.316 \pm 0.007$ | $0.468 \pm 0.012$ | $\mathbf{0.201 \pm 0.007}$ | $0.254 \pm 0.007$ | $0.330 \pm 0.014$ |
| GPT-4o-mini | $0.513 \pm 0.017$ | $0.422 \pm 0.011$ | $0.431 \pm 0.006$ | $0.692 \pm 0.009$ | $\mathbf{0.363 \pm 0.014}$ | $0.375 \pm 0.008$ | $0.559 \pm 0.019$ |
| GPT-5.2 | $0.387 \pm 0.001$ | $0.299 \pm 0.040$ | $0.201 \pm 0.023$ | $0.377 \pm 0.028$ | $\mathbf{0.256 \pm 0.031}$ | $0.321 \pm 0.020$ | $0.375 \pm 0.025$ |
| Claude-Sonnet-4.5 | $0.124 \pm 0.002$ | $0.387 \pm 0.002$ | $0.279 \pm 0.071$ | $0.299 \pm 0.002$ | $0.188 \pm 0.045$ | $\mathbf{0.115 \pm 0.002}$ | $0.277 \pm 0.003$ |
| Gemini-2.5-Pro | $\mathbf{0.116 \pm 0.003}$ | $0.149 \pm 0.002$ | $0.128 \pm 0.003$ | $0.173 \pm 0.003$ | $0.130 \pm 0.003$ | $0.117 \pm 0.002$ | $0.159 \pm 0.003$ |
| Base Quantitative Forecaster | - | $0.497 \pm 0.018$ | $0.605 \pm 0.006$ | $0.921 \pm 0.023$ | $0.497 \pm 0.018$ | $0.605 \pm 0.006$ | $0.921 \pm 0.023$ |

Table 9: RCRPS on full-RoI tasks in CiK, where context influences the entire forecast (mean ± std. err.; lower is better). Median-CorDP achieves the best performance in roughly half the models; DP is strong and often best for larger models (Qwen2.5-14B/72B, Llama3.3-70B, frontier APIs). Best per model in **bold**.

| Model | Direct Prompt (DP) | Median Corrector (Median-CorDP) | | | SampleWise Corrector (SampleWise-CorDP) | | |
|---|---|---|---|---|---|---|---|
| | | LAG-LLAMA | CHRONOS LARGE | ARIMA | LAG-LLAMA | CHRONOS LARGE | ARIMA |
| Qwen2.5-0.5B-Inst | $0.243 \pm 0.103$ | $\mathbf{0.116 \pm 0.007}$ | $0.501 \pm 0.008$ | $0.675 \pm 0.025$ | $0.236 \pm 0.028$ | $0.861 \pm 0.204$ | $0.716 \pm 0.044$ |
| Qwen2.5-1.5B-Inst | $0.706 \pm 0.147$ | $\mathbf{0.185 \pm 0.047}$ | $0.488 \pm 0.008$ | $0.680 \pm 0.022$ | $0.794 \pm 0.065$ | $0.485 \pm 0.010$ | $1.185 \pm 0.043$ |
| Qwen2.5-3B-Inst | $\mathbf{0.234 \pm 0.056}$ | $0.483 \pm 0.008$ | $0.478 \pm 0.005$ | $0.469 \pm 0.024$ | $0.418 \pm 0.107$ | $0.474 \pm 0.005$ | $0.422 \pm 0.118$ |
| Qwen2.5-7B-Inst | $0.470 \pm 0.078$ | $0.507 \pm 0.009$ | $0.947 \pm 0.004$ | $0.547 \pm 0.026$ | $0.523 \pm 0.015$ | $\mathbf{0.146 \pm 0.065}$ | $0.537 \pm 0.036$ |
| Qwen2.5-14B-Inst | $0.039 \pm 0.015$ | $0.001 \pm 0.003$ | $\mathbf{0.000 \pm 0.005}$ | $0.051 \pm 0.009$ | $0.457 \pm 0.015$ | $0.455 \pm 0.010$ | $0.466 \pm 0.030$ |
| Qwen2.5-32B-Inst | $0.479 \pm 0.019$ | $0.001 \pm 0.009$ | $\mathbf{0.000 \pm 0.005}$ | $0.000 \pm 0.027$ | $0.758 \pm 0.012$ | $0.455 \pm 0.008$ | $0.455 \pm 0.028$ |
| Qwen2.5-72B-Inst | $0.032 \pm 0.028$ | $0.304 \pm 0.006$ | $\mathbf{0.000 \pm 0.006}$ | $0.003 \pm 0.007$ | $0.004 \pm 0.008$ | $0.000 \pm 0.008$ | $0.001 \pm 0.024$ |
| Llama-3.2-1B-Inst | $0.275 \pm 0.092$ | $\mathbf{0.084 \pm 0.011}$ | $0.482 \pm 0.015$ | $0.499 \pm 0.068$ | $0.905 \pm 0.027$ | $0.924 \pm 0.013$ | $1.168 \pm 0.053$ |
| Llama-3.2-3B-Inst | $1.030 \pm 0.090$ | $\mathbf{0.112 \pm 0.032}$ | $0.519 \pm 0.018$ | $0.502 \pm 0.081$ | $0.884 \pm 0.091$ | $0.487 \pm 0.016$ | $1.003 \pm 0.116$ |
| Llama-3-8B-Inst | $0.169 \pm 0.172$ | $\mathbf{0.061 \pm 0.011}$ | $0.481 \pm 0.015$ | $0.438 \pm 0.034$ | $0.609 \pm 0.029$ | $0.476 \pm 0.012$ | $0.943 \pm 0.033$ |
| Llama3.3-70B-Inst | $\mathbf{0.000 \pm 0.024}$ | $0.000 \pm 0.003$ | $0.001 \pm 0.007$ | $0.000 \pm 0.022$ | $0.002 \pm 0.010$ | $0.000 \pm 0.006$ | $0.000 \pm 0.022$ |
| Llama3.1-405B-Inst | $0.004 \pm 0.009$ | $0.060 \pm 0.031$ | $0.303 \pm 0.008$ | $0.006 \pm 0.025$ | $0.042 \pm 0.016$ | $\mathbf{0.000 \pm 0.009}$ | $0.228 \pm 0.027$ |
| GPT-4o | $0.455 \pm 0.029$ | $\mathbf{0.000 \pm 0.008}$ | $0.000 \pm 0.010$ | $0.000 \pm 0.021$ | $0.001 \pm 0.008$ | $0.000 \pm 0.010$ | $0.000 \pm 0.028$ |
| GPT-4o-mini | $\mathbf{0.001 \pm 0.032}$ | $0.006 \pm 0.004$ | $0.018 \pm 0.006$ | $0.002 \pm 0.003$ | $0.245 \pm 0.008$ | $0.019 \pm 0.009$ | $0.017 \pm 0.034$ |
| GPT-5.2 | $\mathbf{0.000 \pm 0.002}$ | $0.000 \pm 0.002$ | $0.000 \pm 0.002$ | $0.000 \pm 0.017$ | $0.000 \pm 0.004$ | $0.000 \pm 0.004$ | $0.000 \pm 0.032$ |
| Claude-Sonnet-4.5 | $\mathbf{0.000 \pm 0.003}$ | $0.000 \pm 0.002$ | $0.000 \pm 0.002$ | $0.000 \pm 0.003$ | $0.000 \pm 0.004$ | $0.000 \pm 0.003$ | $0.000 \pm 0.006$ |
| Base Quantitative Forecaster | - | $0.204 \pm 0.037$ | $0.487 \pm 0.010$ | $0.843 \pm 0.050$ | $0.204 \pm 0.037$ | $0.487 \pm 0.010$ | $0.843 \pm 0.050$ |

Table 10: Constraint violation CRPS on constraint tasks in CiK (mean ± std. err.; lower is better). Median-CorDP dominates: many models (including GPT-4o, GPT-5.2, Claude-Sonnet-4.5, Llama3.3-70B) achieve near-zero or zero constraint violation with CorDP, whereas DP incurs higher violation. Best per model in **bold**.

| Model | Lag-Llama | | Chronos Large | | ARIMA | |
|---|---|---|---|---|---|---|
| | DP | Med-CorDP | DP | Med-CorDP | DP | Med-CorDP |
| Llama-3.2-1B | 0.396 ± 0.027 | 0.394 ± 0.004 | 0.396 ± 0.027 | 0.515 ± 0.007 | 0.396 ± 0.027 | 0.612 ± 0.018 |
| Llama-3.2-3B | 0.687 ± 0.025 | 0.344 ± 0.011 | 0.687 ± 0.025 | 0.455 ± 0.009 | 0.687 ± 0.025 | 0.573 ± 0.022 |
| Llama-3-8B | 0.543 ± 0.026 | 0.315 ± 0.004 | 0.543 ± 0.026 | 0.453 ± 0.005 | 0.543 ± 0.026 | 0.571 ± 0.004 |
| Llama-3.3-70B | 0.230 ± 0.006 | 0.281 ± 0.002 | 0.230 ± 0.006 | 0.251 ± 0.004 | 0.230 ± 0.006 | 0.352 ± 0.006 |
| Llama-3.1-405B | 0.173 ± 0.003 | 0.278 ± 0.009 | 0.173 ± 0.003 | 0.226 ± 0.004 | 0.173 ± 0.003 | 0.257 ± 0.008 |
| Qwen-2.5-0.5B | 0.592 ± 0.027 | 0.633 ± 0.002 | 0.592 ± 0.027 | 0.801 ± 0.003 | 0.592 ± 0.027 | 0.761 ± 0.054 |
| Qwen-2.5-1.5B | 0.616 ± 0.018 | 0.426 ± 0.013 | 0.616 ± 0.018 | 0.537 ± 0.003 | 0.616 ± 0.018 | 0.682 ± 0.006 |
| Qwen-2.5-3B | 0.424 ± 0.017 | 0.490 ± 0.005 | 0.424 ± 0.017 | 0.491 ± 0.004 | 0.424 ± 0.017 | 0.597 ± 0.009 |
| Qwen-2.5-7B | 0.401 ± 0.006 | 0.419 ± 0.004 | 0.401 ± 0.006 | 0.641 ± 0.008 | 0.401 ± 0.006 | 0.633 ± 0.008 |
| Qwen-2.5-14B | 0.247 ± 0.006 | 0.315 ± 0.003 | 0.247 ± 0.006 | 0.334 ± 0.006 | 0.247 ± 0.006 | 0.423 ± 0.004 |
| Qwen-2.5-32B | 0.397 ± 0.008 | 0.248 ± 0.004 | 0.397 ± 0.008 | 0.272 ± 0.005 | 0.397 ± 0.008 | 0.329 ± 0.008 |
| Qwen-2.5-72B | 0.202 ± 0.009 | 0.319 ± 0.008 | 0.202 ± 0.009 | 0.358 ± 0.010 | 0.202 ± 0.009 | 0.428 ± 0.009 |
| GPT-4o | 0.317 ± 0.009 | 0.253 ± 0.004 | 0.317 ± 0.009 | 0.240 ± 0.004 | 0.317 ± 0.009 | 0.354 ± 0.007 |
| GPT-4o-mini | 0.389 ± 0.010 | 0.364 ± 0.006 | 0.389 ± 0.010 | 0.340 ± 0.004 | 0.389 ± 0.010 | 0.516 ± 0.005 |
| GPT-5.2 | 0.271 ± 0.001 | 0.246 ± 0.024 | 0.271 ± 0.001 | 0.167 ± 0.014 | 0.271 ± 0.001 | 0.285 ± 0.017 |
| Claude-Sonnet-4.5 | 0.114 ± 0.001 | 0.299 ± 0.001 | 0.114 ± 0.001 | 0.213 ± 0.042 | 0.114 ± 0.001 | 0.242 ± 0.001 |
| Gemini-2.5-Pro | 0.108 ± 0.002 | 0.157 ± 0.002 | 0.108 ± 0.002 | 0.123 ± 0.002 | 0.108 ± 0.002 | 0.166 ± 0.002 |

Table 11: Median-CorDP vs Direct Prompt (DP). Each base forecaster group shows DP alongside the corresponding Median-CorDP result. Green = CorDP beats DP (lower RCRPS is better).

| Model | Lag-Llama | | Chronos Large | | ARIMA | |
|---|---|---|---|---|---|---|
| | DP | SW-CorDP | DP | SW-CorDP | DP | SW-CorDP |
| Llama-3.2-1B | 0.396 ± 0.027 | 0.541 ± 0.009 | 0.396 ± 0.027 | 0.634 ± 0.005 | 0.396 ± 0.027 | 0.672 ± 0.015 |
| Llama-3.2-3B | 0.687 ± 0.025 | 0.509 ± 0.026 | 0.687 ± 0.025 | 0.423 ± 0.007 | 0.687 ± 0.025 | 0.663 ± 0.031 |
| Llama-3-8B | 0.543 ± 0.026 | 0.426 ± 0.009 | 0.543 ± 0.026 | 0.410 ± 0.004 | 0.543 ± 0.026 | 0.636 ± 0.010 |
| Llama-3.3-70B | 0.230 ± 0.006 | 0.223 ± 0.004 | 0.230 ± 0.006 | 0.215 ± 0.004 | 0.230 ± 0.006 | 0.311 ± 0.007 |
| Llama-3.1-405B | 0.173 ± 0.003 | 0.199 ± 0.006 | 0.173 ± 0.003 | 0.194 ± 0.004 | 0.173 ± 0.003 | 0.229 ± 0.008 |
| Qwen-2.5-0.5B | 0.592 ± 0.027 | 0.494 ± 0.008 | 0.592 ± 0.027 | 0.644 ± 0.076 | 0.592 ± 0.027 | 0.655 ± 0.055 |
| Qwen-2.5-1.5B | 0.616 ± 0.018 | 0.522 ± 0.018 | 0.616 ± 0.018 | 0.474 ± 0.005 | 0.616 ± 0.018 | 0.719 ± 0.013 |
| Qwen-2.5-3B | 0.424 ± 0.017 | 0.398 ± 0.028 | 0.424 ± 0.017 | 0.451 ± 0.005 | 0.424 ± 0.017 | 0.512 ± 0.032 |
| Qwen-2.5-7B | 0.401 ± 0.006 | 0.382 ± 0.007 | 0.401 ± 0.006 | 0.402 ± 0.020 | 0.401 ± 0.006 | 0.540 ± 0.011 |
| Qwen-2.5-14B | 0.247 ± 0.006 | 0.364 ± 0.006 | 0.247 ± 0.006 | 0.410 ± 0.006 | 0.247 ± 0.006 | 0.471 ± 0.009 |
| Qwen-2.5-32B | 0.397 ± 0.008 | 0.310 ± 0.005 | 0.397 ± 0.008 | 0.338 ± 0.007 | 0.397 ± 0.008 | 0.414 ± 0.009 |
| Qwen-2.5-72B | 0.202 ± 0.009 | 0.255 ± 0.010 | 0.202 ± 0.009 | 0.322 ± 0.010 | 0.202 ± 0.009 | 0.386 ± 0.010 |
| GPT-4o | 0.317 ± 0.009 | 0.184 ± 0.004 | 0.317 ± 0.009 | 0.196 ± 0.004 | 0.317 ± 0.009 | 0.251 ± 0.008 |
| GPT-4o-mini | 0.389 ± 0.010 | 0.302 ± 0.008 | 0.389 ± 0.010 | 0.296 ± 0.005 | 0.389 ± 0.010 | 0.415 ± 0.011 |
| GPT-5.2 | 0.271 ± 0.001 | 0.201 ± 0.018 | 0.271 ± 0.001 | 0.235 ± 0.012 | 0.271 ± 0.001 | 0.276 ± 0.015 |
| Claude-Sonnet-4.5 | 0.114 ± 0.001 | 0.162 ± 0.027 | 0.114 ± 0.001 | 0.110 ± 0.001 | 0.114 ± 0.001 | 0.217 ± 0.003 |
| Gemini-2.5-Pro | 0.108 ± 0.002 | 0.130 ± 0.002 | 0.108 ± 0.002 | 0.110 ± 0.002 | 0.108 ± 0.002 | 0.145 ± 0.003 |

Table 12: SampleWise-CorDP vs Direct Prompt (DP). Each base forecaster group shows DP alongside the corresponding SampleWise-CorDP result. Green = CorDP beats DP (lower RCRPS is better).

## D.2 Side-by-Side Comparison of CorDP vs DP

Table 11 and Table 12 provide a side-by-side comparison of DP and each CorDP variant per base forecaster. Green cells indicate that CorDP beats DP (lower RCRPS is better).

## D.3 Results on various groups of tasks

We now look into results aggregated across the various kinds of tasks in the CiK benchmark: Table 7, Table 8 showcases performance of methods within and outside the region of interest (RoI) respectively for tasks that have an RoI, Table 9 shows performance across tasks where the entire prediction window is the RoI, and

Table 10 shows constraint RCRPS across tasks with constraints. We find that SampleWise-CorDP has an advantage on tasks with an RoI, achieving the best performance in most models, both within and outside the RoI. Median-CorDP however has a clear advantage on tasks where the shape of the entire forecast is influenced by the context, which make up most of the benchmark, achieving the best performance in half the models, and trailing closely with DP in the other. These results also indicate that DP methods are still consistently strong in tasks where the entire prediction is influenced by the context. Median-CorDP overwhelmingly outperforms DP and bags the best performance in tasks with constraints, sometimes achieving perfect performance with large models. This shows that when choosing between CorDP methods, the kind of tasks that will be encountered is an important factor to consider.

### D.4 IC-CorDP: CorDP with an in-context example

To evaluate if the proposed CorDP and IC-DP methods can be combined to yield further gains, we evaluate a hybrid method, which we call IC-CorDP: this method uses CorDP as the foundation: the goal is to output a forecast given the history, context and a base forecast; when combined with IC-DP, it uses an in-context example that contains the history, context, base forecast and ground truth of the example. We abbreviate this hybrid method as IC-CorDP (In-Context Corrector Direct Prompt). We use the Median-CorDP for this experiment (and hence call this method IC-Median-CorDP), and run experiments with a subset of LLMs. As in CorDP, we test it with multiple base forecasters.

The results are in Table 13. IC-Median-CorDP improves performance compared to Median-CorDP across LLMs across all sizes, and across multiple base quantitative forecasters that the LLM bootstraps over. The levels of gains achieved with IC-Median-CorDP depend on the LLM and the base quantitative forecaster. This shows that there is clear potential in combining the two strategies to improve performance.

| Model | Direct Prompt (DP) | Median Corrector (Median-CorDP) | | | In-Context Median Corrector (IC-Median-CorDP) | | |
|---|---|---|---|---|---|---|---|
| | | LAG-LLAMA | CHRONOS LARGE | ARIMA | LAG-LLAMA | CHRONOS LARGE | ARIMA |
| Llama3.2-1B-Inst | $0.396 \pm 0.027$ | $0.394 \pm 0.004$ | $0.515 \pm 0.007$ | $0.612 \pm 0.018$ | $\mathbf{0.315 \pm 0.004}$ | $0.390 \pm 0.031$ | $0.480 \pm 0.010$ |
| Llama3.2-3B-Inst | $0.687 \pm 0.025$ | $0.344 \pm 0.011$ | $0.455 \pm 0.009$ | $0.573 \pm 0.022$ | $\mathbf{0.334 \pm 0.008}$ | $0.354 \pm 0.011$ | $0.478 \pm 0.016$ |
| Qwen2.5-0.5B-Inst | $0.592 \pm 0.027$ | $0.633 \pm 0.002$ | $0.801 \pm 0.003$ | $0.761 \pm 0.054$ | $\mathbf{0.358 \pm 0.005}$ | $1.734 \pm 0.008$ | $0.548 \pm 0.010$ |
| Qwen2.5-1.5B-Inst | $0.616 \pm 0.018$ | $0.426 \pm 0.013$ | $0.537 \pm 0.003$ | $0.682 \pm 0.006$ | $\mathbf{0.305 \pm 0.004}$ | $0.390 \pm 0.028$ | $0.334 \pm 0.009$ |
| Qwen2.5-3B-Inst | $0.424 \pm 0.017$ | $0.490 \pm 0.005$ | $0.491 \pm 0.004$ | $0.597 \pm 0.009$ | $\mathbf{0.326 \pm 0.008}$ | $0.475 \pm 0.009$ | $0.399 \pm 0.013$ |
| Qwen2.5-7B-Inst | $0.401 \pm 0.006$ | $0.419 \pm 0.004$ | $0.641 \pm 0.008$ | $0.633 \pm 0.008$ | $\mathbf{0.322 \pm 0.008}$ | $0.334 \pm 0.009$ | $0.449 \pm 0.010$ |
| Qwen2.5-14B-Inst | $\mathbf{0.247 \pm 0.006}$ | $0.315 \pm 0.003$ | $0.334 \pm 0.006$ | $0.423 \pm 0.004$ | $0.256 \pm 0.006$ | $0.293 \pm 0.006$ | $0.336 \pm 0.010$ |
| Qwen2.5-32B-Inst | $0.397 \pm 0.008$ | $\mathbf{0.248 \pm 0.004}$ | $0.272 \pm 0.005$ | $0.329 \pm 0.008$ | $0.261 \pm 0.005$ | $0.261 \pm 0.007$ | $0.383 \pm 0.009$ |
| Qwen2.5-72B-Inst | $0.202 \pm 0.009$ | $0.319 \pm 0.008$ | $0.358 \pm 0.010$ | $0.428 \pm 0.009$ | $0.233 \pm 0.005$ | $\mathbf{0.180 \pm 0.005}$ | $0.400 \pm 0.008$ |
| Llama3.1-405B-Inst | $\mathbf{0.173 \pm 0.003}$ | $0.278 \pm 0.009$ | $0.226 \pm 0.004$ | $0.257 \pm 0.008$ | $0.227 \pm 0.006$ | $0.308 \pm 0.006$ | $0.243 \pm 0.012$ |
| Base Quantitative Forecaster | - | $0.382 \pm 0.011$ | $0.492 \pm 0.004$ | $0.636 \pm 0.014$ | $0.382 \pm 0.011$ | $0.492 \pm 0.004$ | $0.636 \pm 0.014$ |

Table 13: Aggregate RCRPS of IC-Median-CorDP (CorDP with one in-context example) vs. Median-CorDP and DP on CiK (mean ± std. err.; lower is better). IC-Median-CorDP improves over Median-CorDP across model sizes and base forecasters, showing that in-context learning and forecast correction combine effectively. Best per model in **bold**.

### D.5 Example Forecasts

#### D.5.1 Task: *ElectricityIncreaseInPredictionWithSplitContext*

> **Context:**
> Background: This is the electricity consumption recorded in Kilowatt (kW) in city A.
> Constraints: None
> Scenario: Suppose that there is a heat wave in city A from 2013-05-28 12:00:00 for 2 hours, which would typically lead to excessive use of air conditioning, and 10 times the usual electricity being consumed. But in this case, residents sought to conserve energy and used lesser air conditioning, resulting in excessive usage of only 4 times the usual electricity.

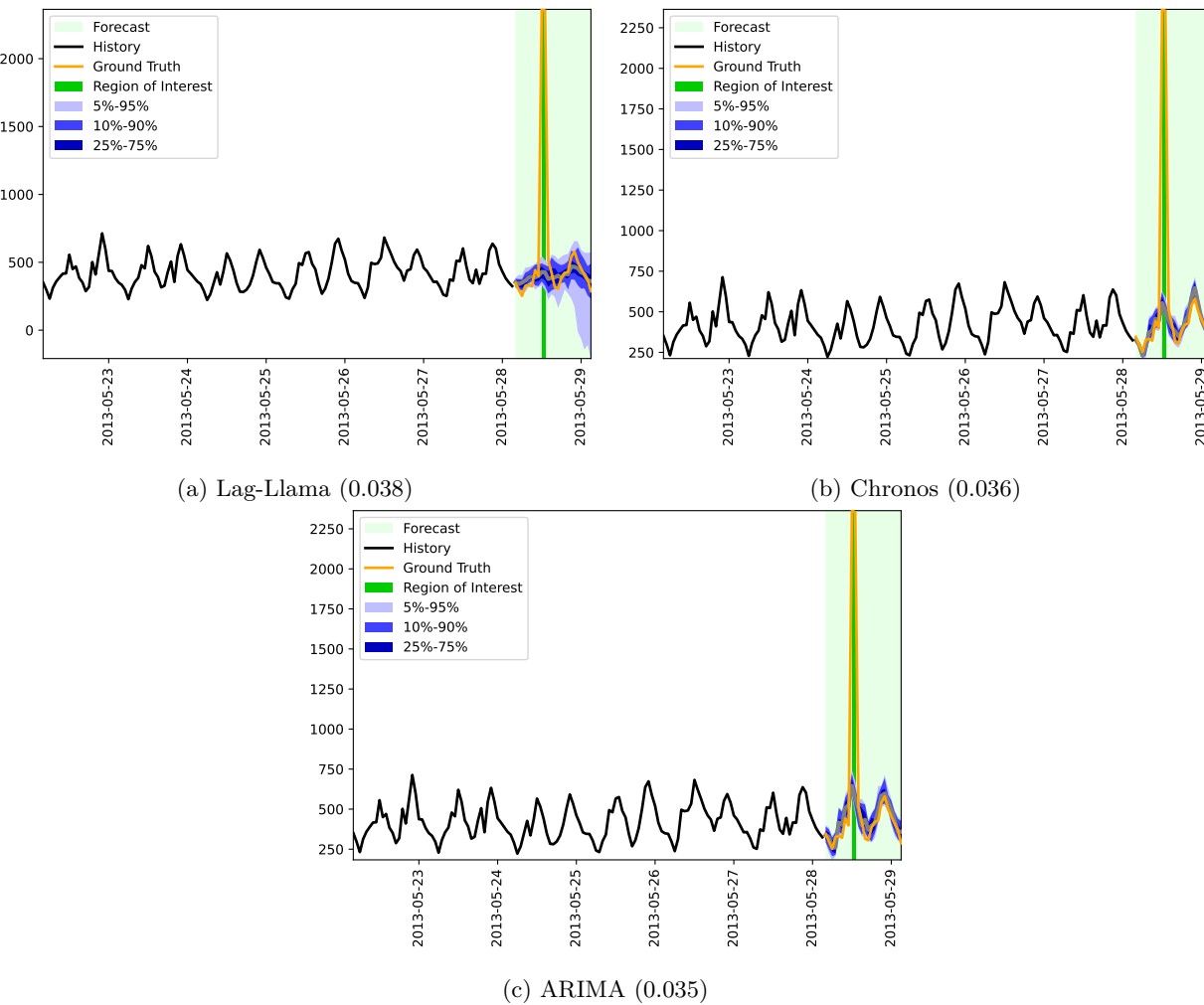

(a) Lag-Llama (0.038)

(b) Chronos (0.036)

(c) ARIMA (0.035)

Figure 16: Forecasts of Lag-Llama, Chronos, and ARIMA on the *ElectricityIncreaseInPredictionWithSplit-Context* task (with RCRPS in brackets)

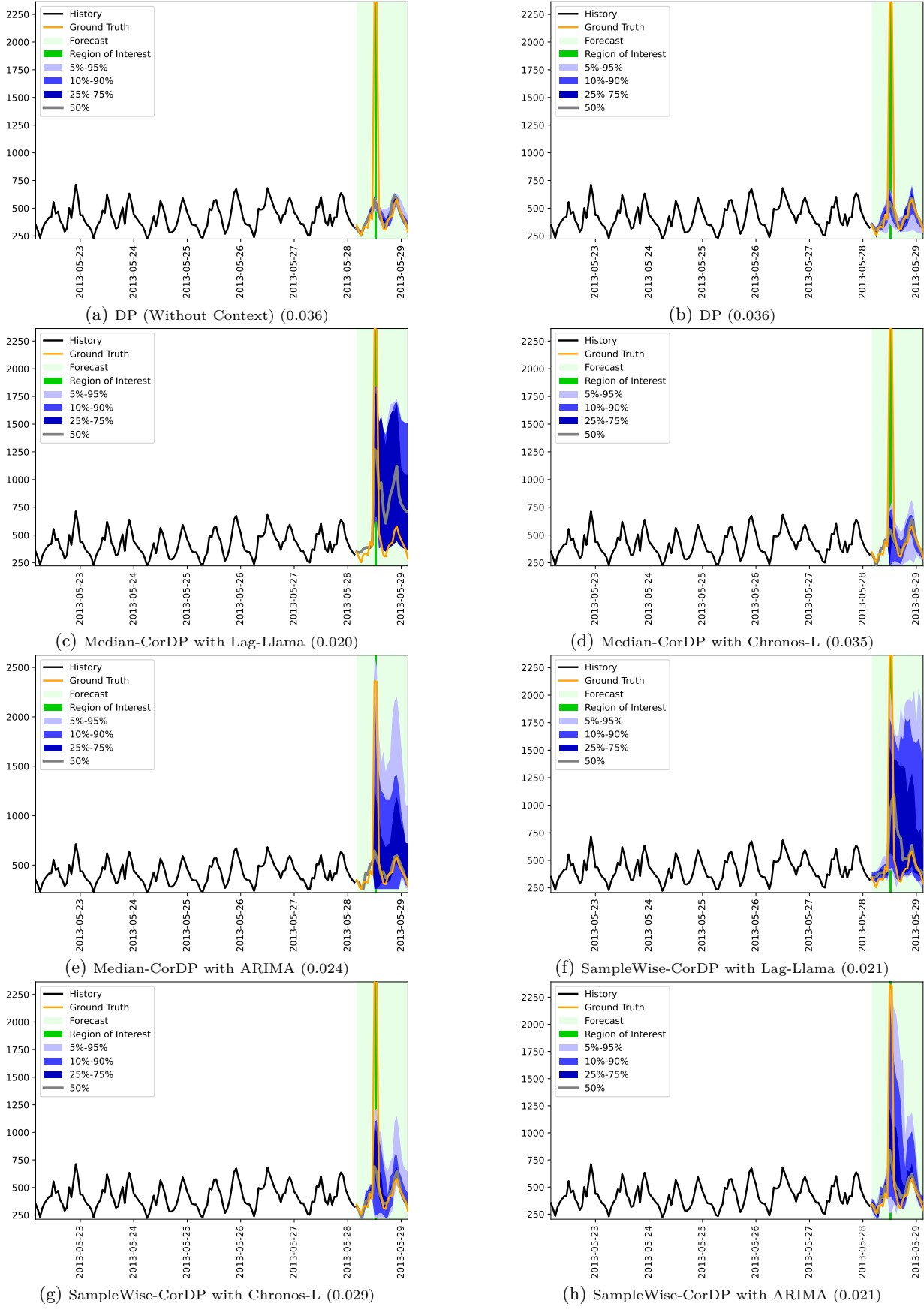

Figure 17: Forecasts of model Qwen2.5-7B-Inst on task *ElectricityIncreaseInPredictionWithSplitContext* (with RCRPS in brackets)

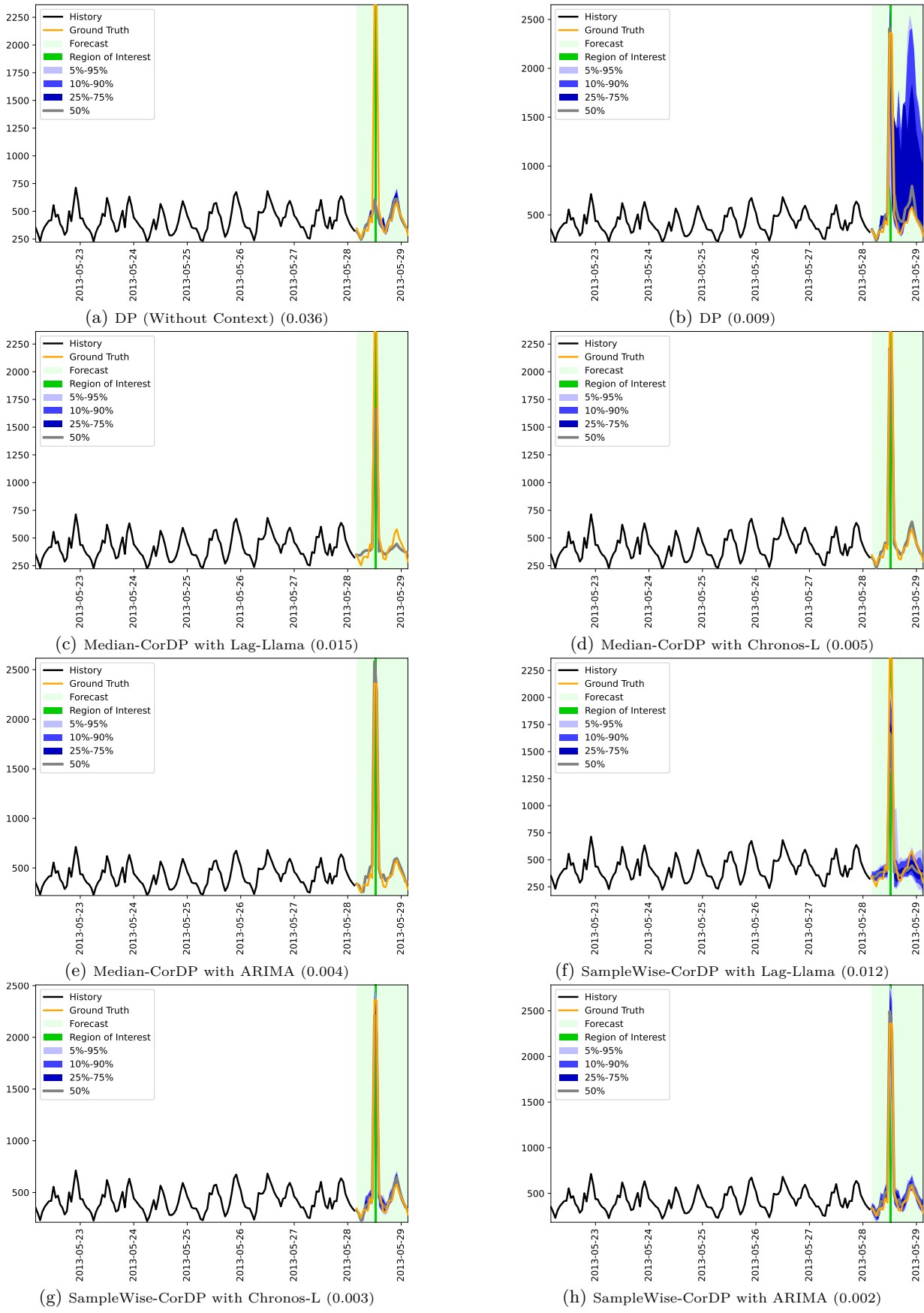

Figure 18: Forecasts of model Qwen2.5-32B-Inst on task *ElectricityIncreaseInPredictionWithSplitContext* (with RCRPS in brackets)

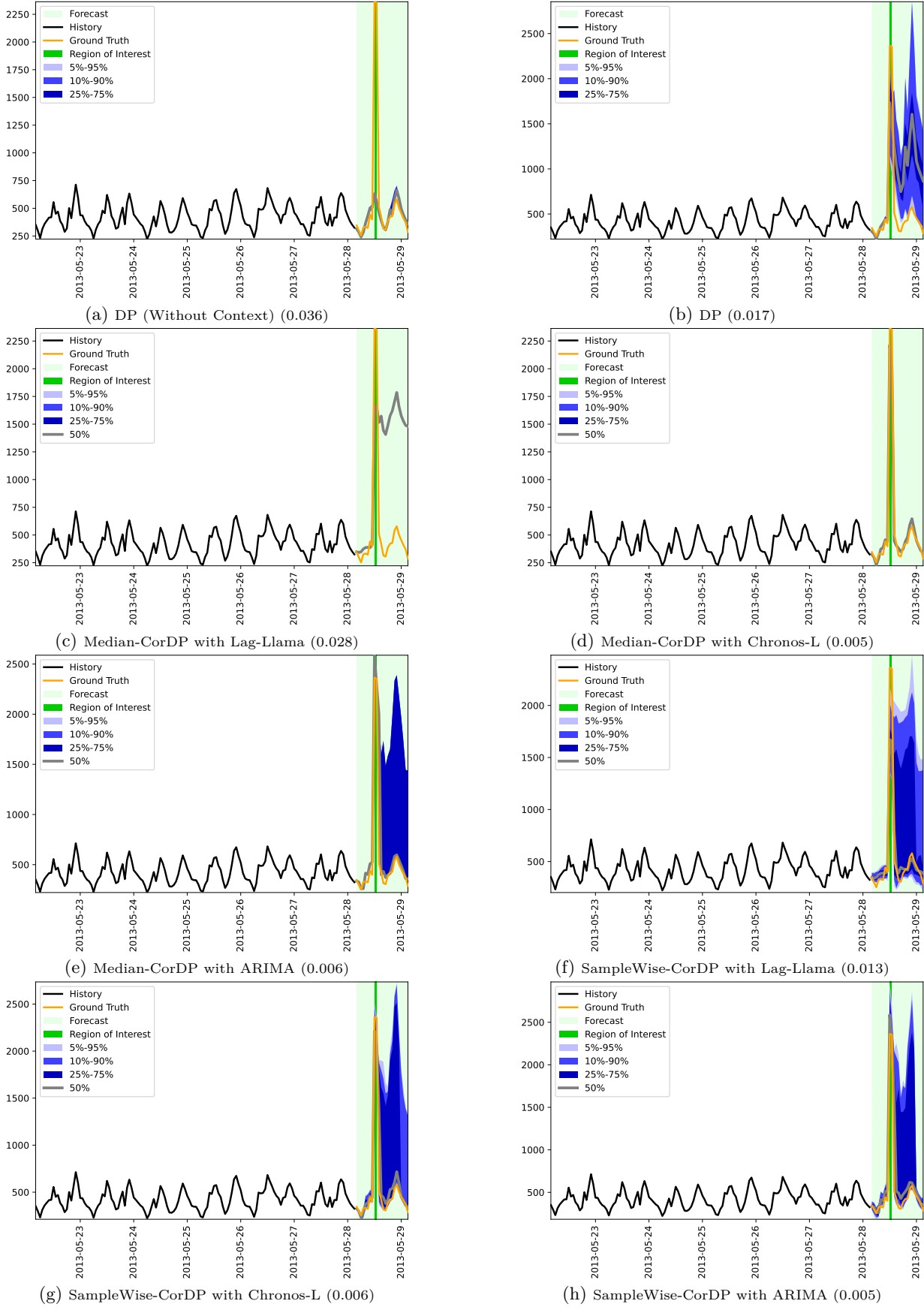

Figure 19: Forecasts of model Llama3.3-70B-Inst on task *ElectricityIncreaseInPredictionWithSplitContext* (with RCRPS in brackets)

**D.5.2 Task: *IncreasedWithdrawalScenario***

**Context:**
Background: This is the number of cash withdrawals from an automated teller machine (ATM) in an arbitrary location in England.
Constraints: None
Scenario: Suppose that there is a carnival from 1996-11-22 00:00:00, for 11 days leading to more people in the area, and 4 times the number of usual withdrawals during that period.

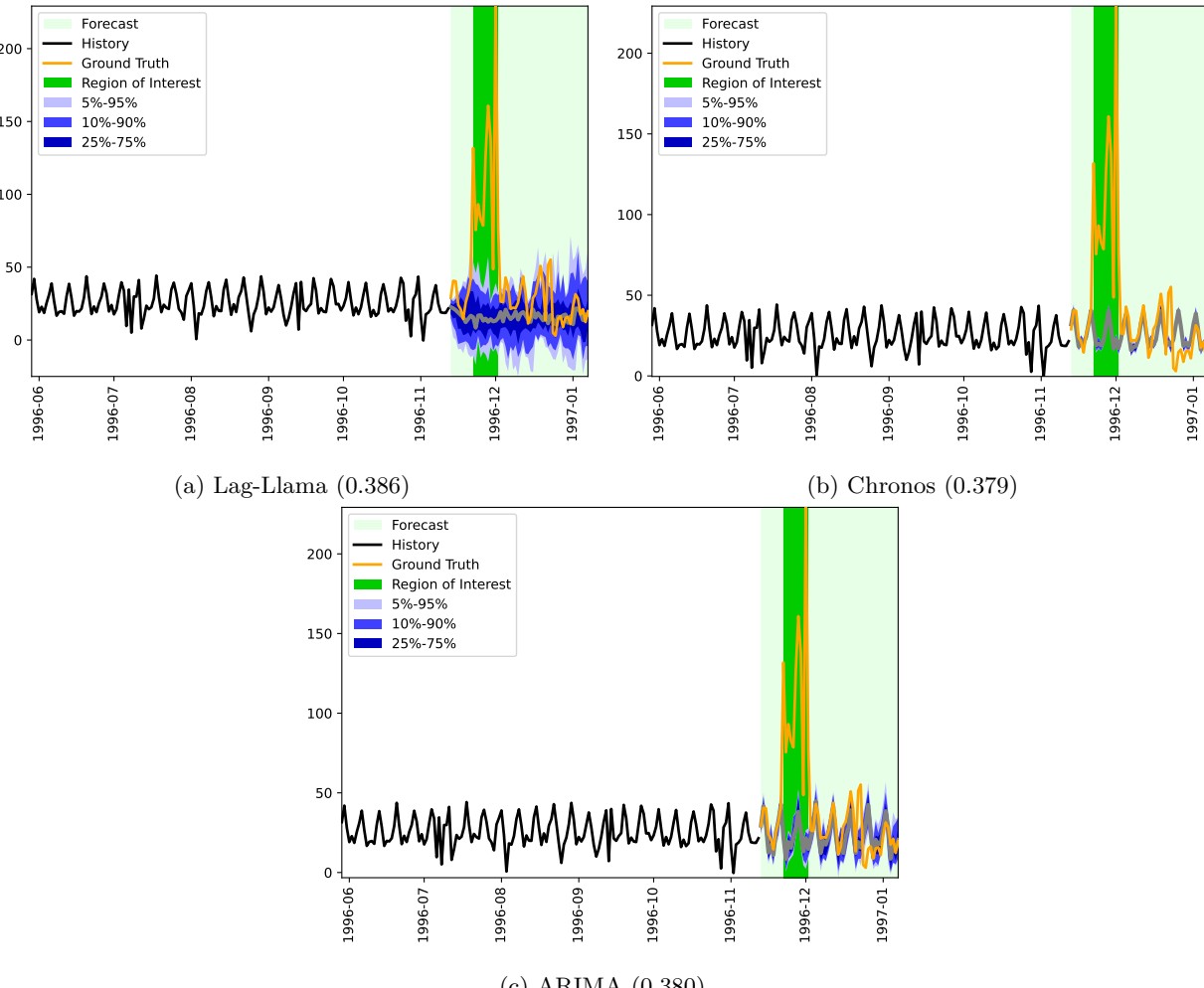

(a) Lag-Llama (0.386)

(b) Chronos (0.379)

(c) ARIMA (0.380)

Figure 20: Forecasts of Lag-Llama, Chronos, and ARIMA on the *IncreasedWithdrawalScenario* task (with RCRPS in brackets)

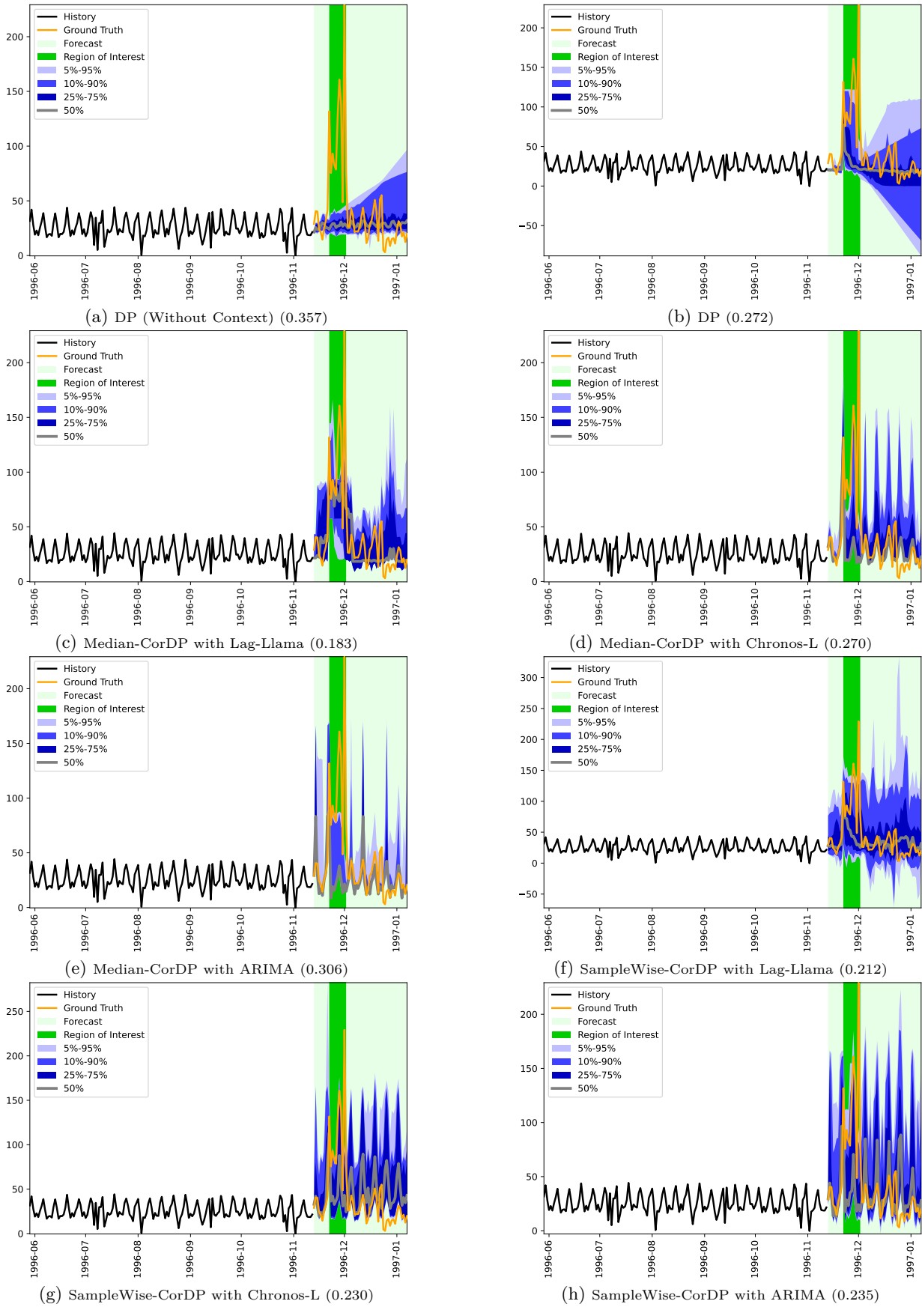

Figure 21: Forecasts of model Qwen2.5-7B-Inst on task *IncreasedWithdrawalScenario* (with RCRPS in brackets)

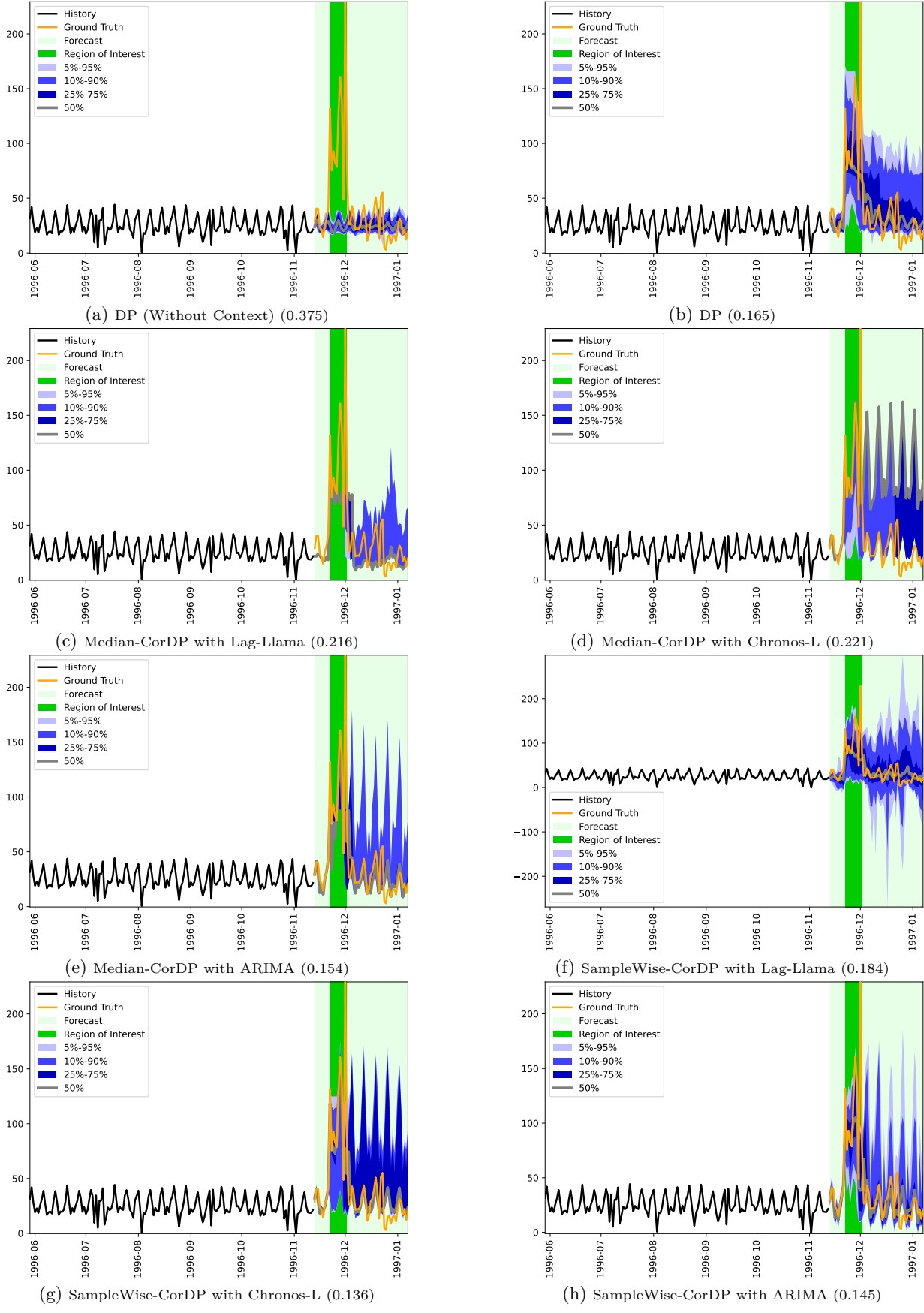

Figure 22: Forecasts of model Qwen2.5-32B-Inst on task *IncreasedWithdrawalScenario* (with RCRPS in brackets)

### D.5.3 Task: *ATMBuildingClosed*

> **Context:**
> Background: This is the number of cash withdrawals from an automated teller machine (ATM) in an arbitrary location in England.
> Constraints: None
> Scenario: Consider that the building which contains the ATM is closed from 1996-11-24 00:00:00, for 10 days.

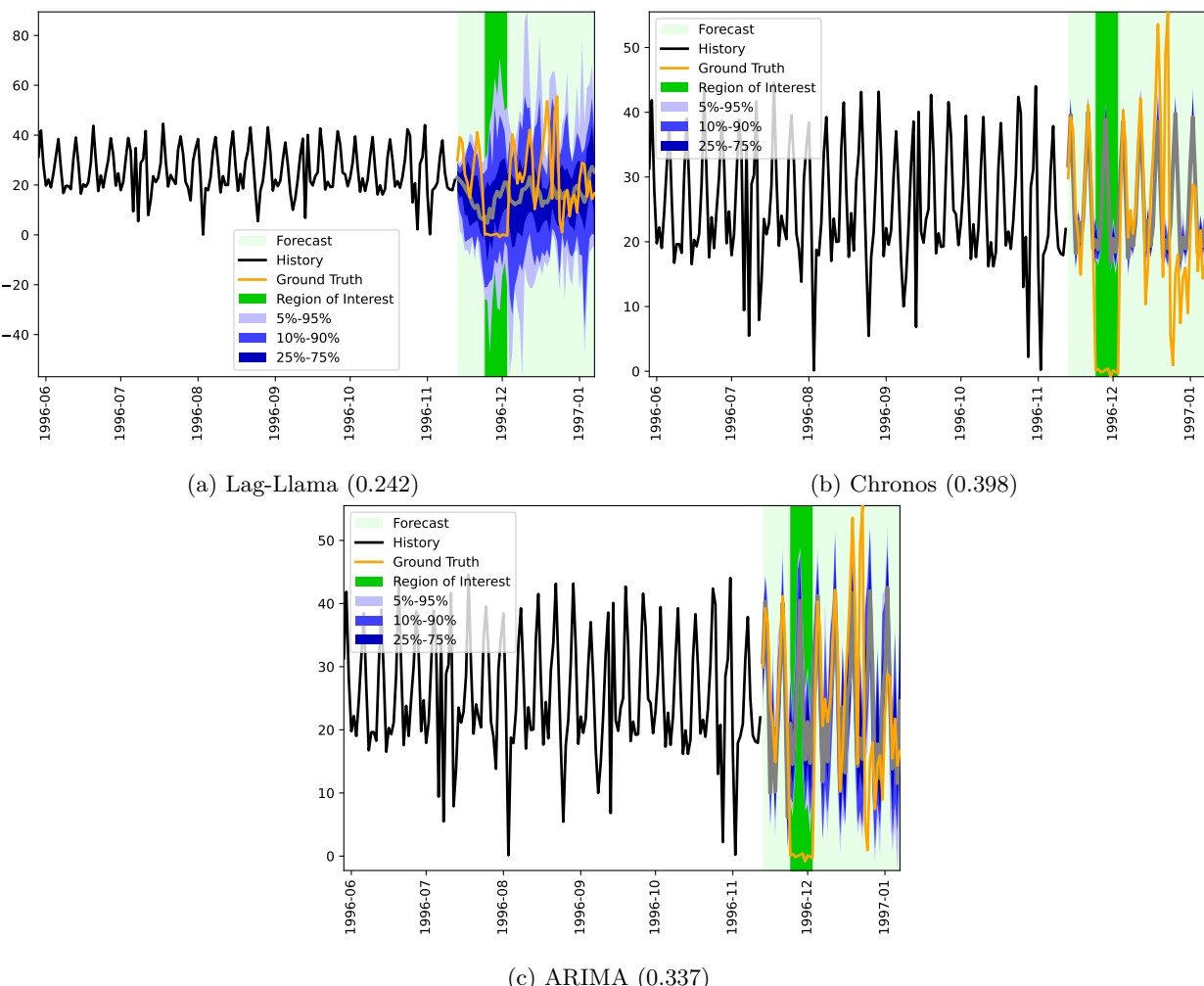

(a) Lag-Llama (0.242)

(b) Chronos (0.398)

(c) ARIMA (0.337)

Figure 23: Forecasts of Lag-Llama, Chronos, and ARIMA on the *ATMBuildingClosedTask* task (with RCRPS in brackets)

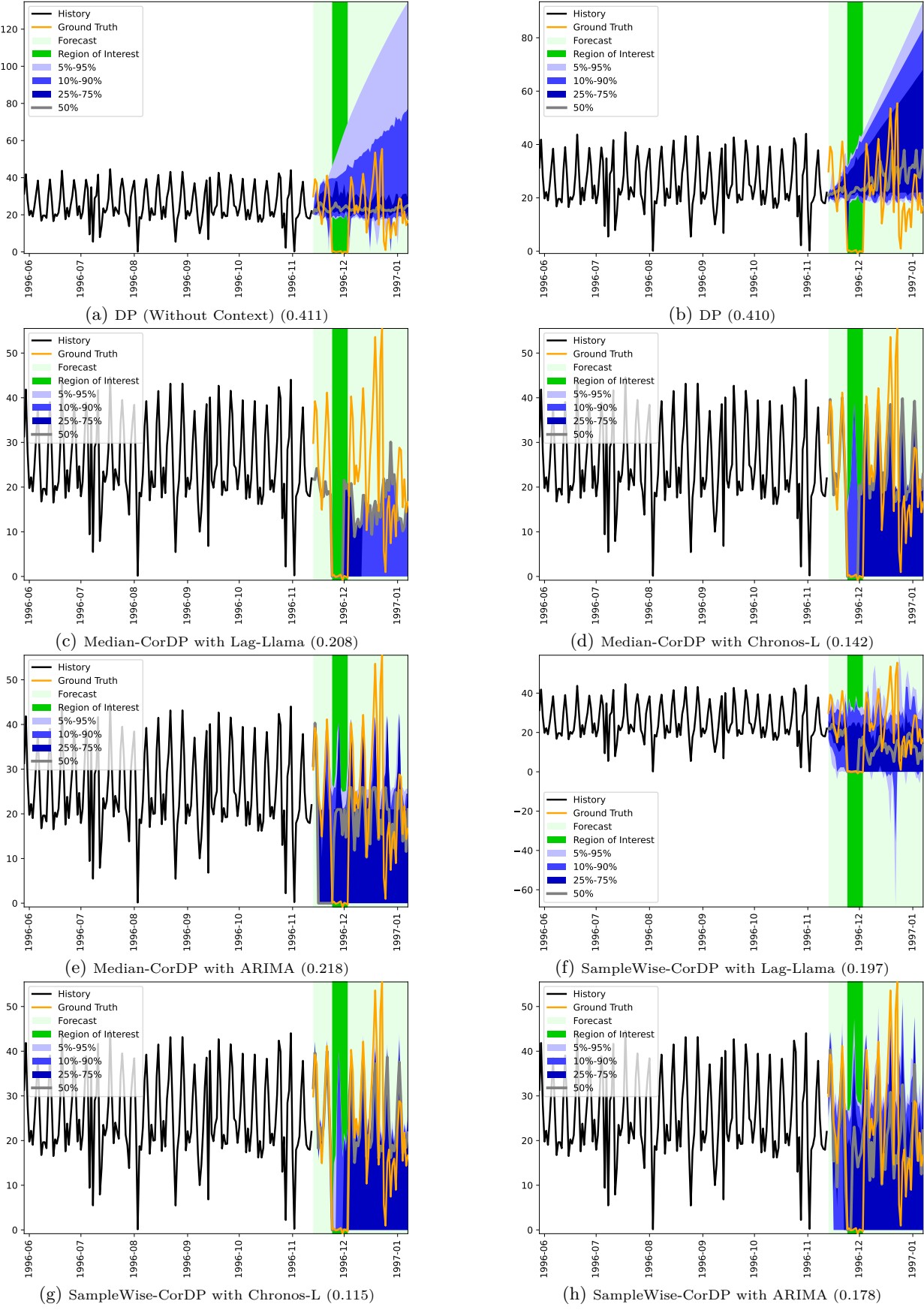

Figure 24: Forecasts of model Qwen2.5-7B-Inst on task *ATMBuildingClosedTask* (with RCRPS in brackets)

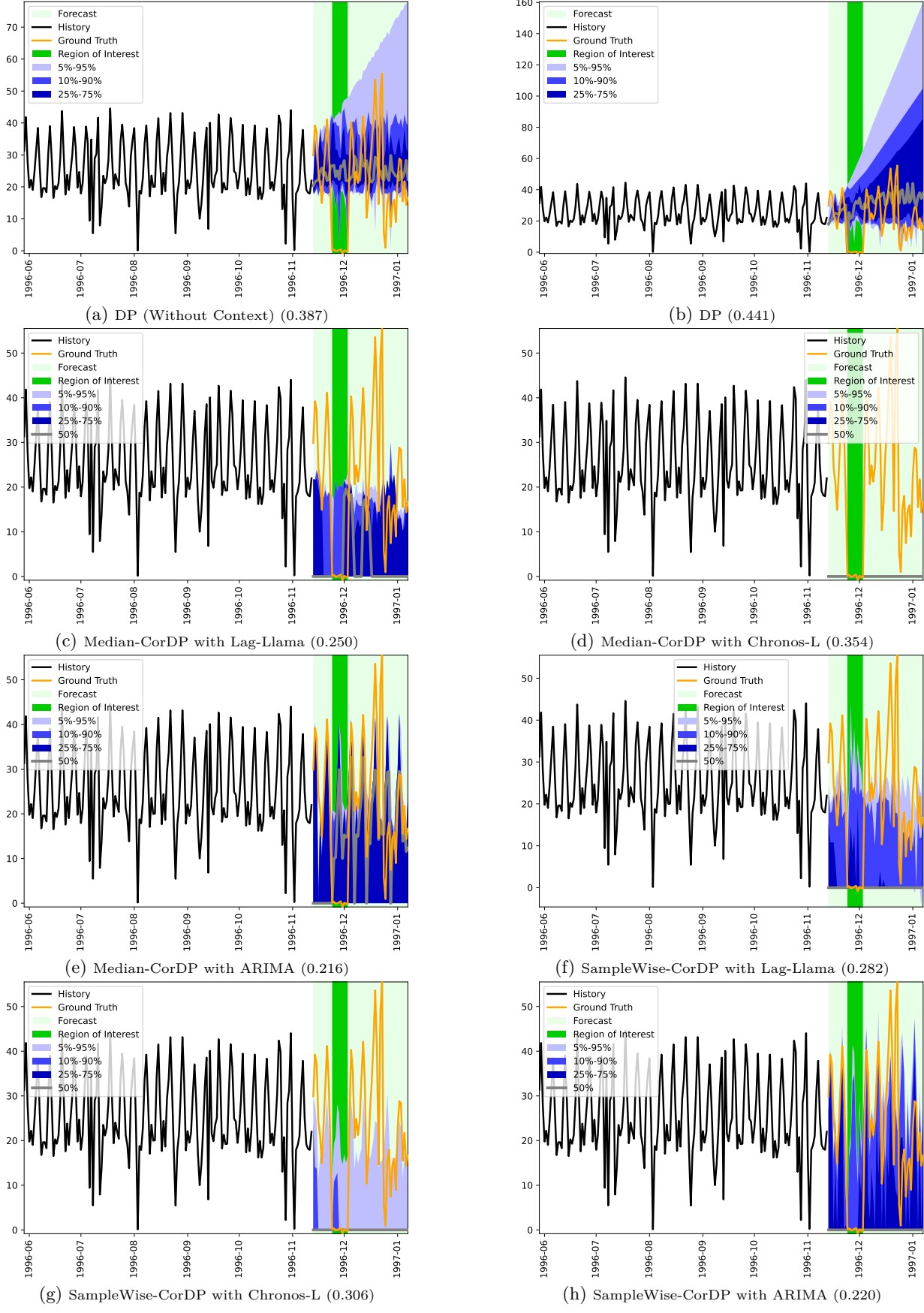

Figure 25: Forecasts of model Llama3-8B-Inst on task *ATMBuildingClosed* (with RCRPS in brackets)

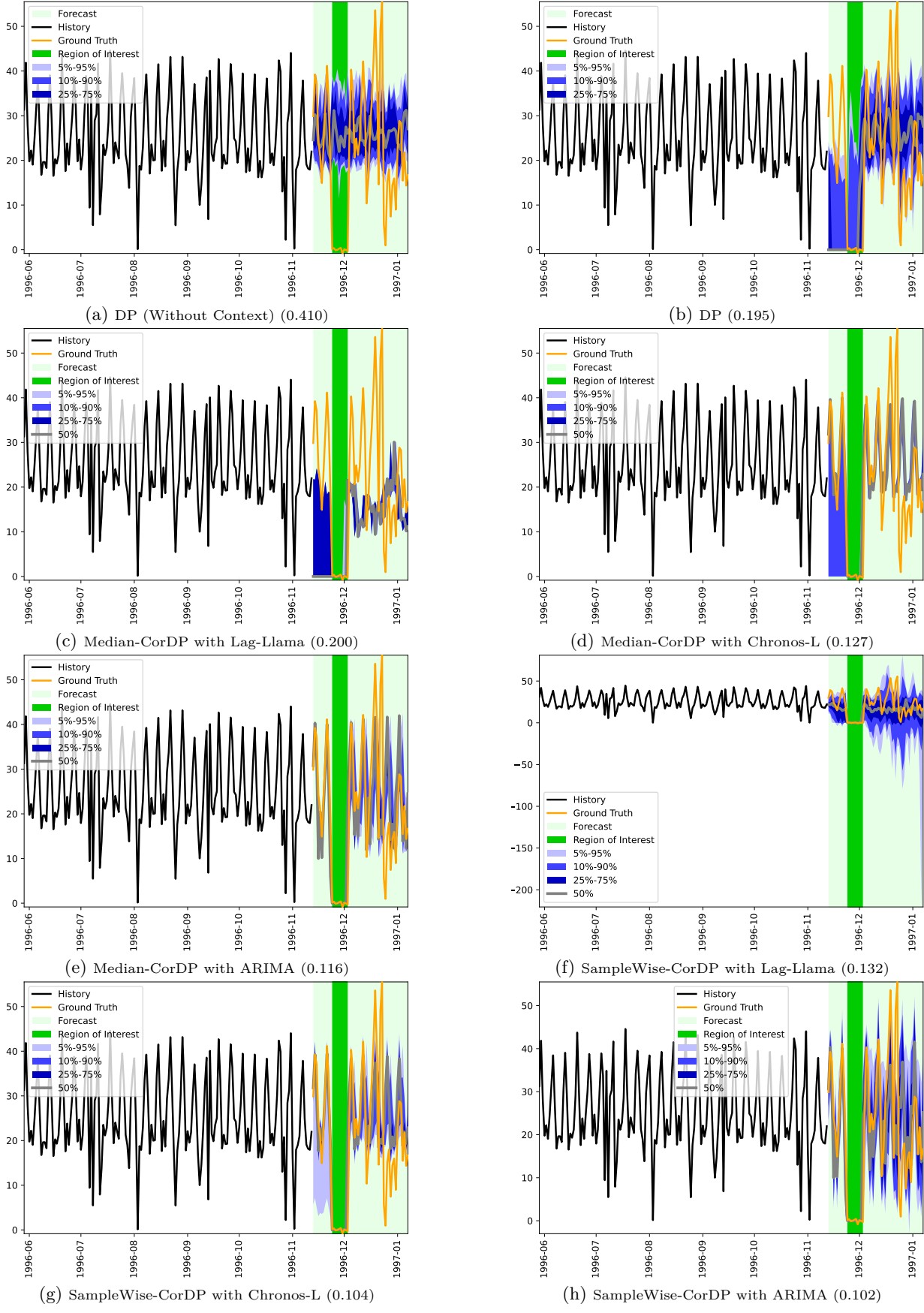

Figure 26: Forecasts of model Llama3.1-405B-Inst on task *ATMBuildingClosed* (with RCRPS in brackets)

**D.5.4   Task:  *ZenithInfoHalfDaySolarForecastTask***

**Context:**
Background: This series contains the amount of sunlight (in Watts per squared meter) arriving on a horizontal surface, for a location in Florida, United States. Over the previous 90 days, the maximum sunlight happened on average at 12:25:33.
Constraints: None
Scenario: None

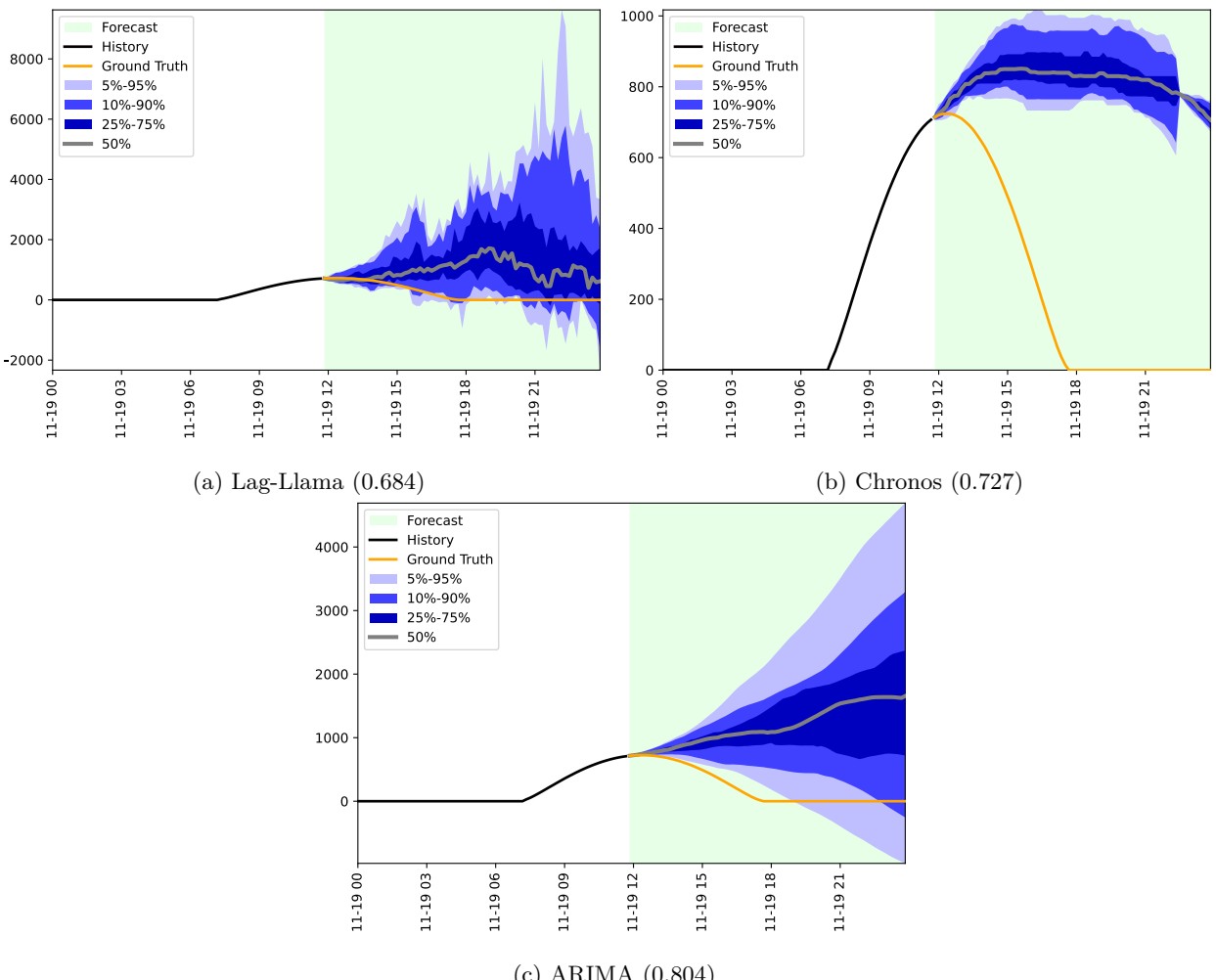

(a) Lag-Llama (0.684)

(b) Chronos (0.727)

(c) ARIMA (0.804)

Figure 27: Forecasts of Lag-Llama, Chronos, and ARIMA on the *ZenithInfoHalfDaySolarForecastTask* task (with RCRPS in brackets)

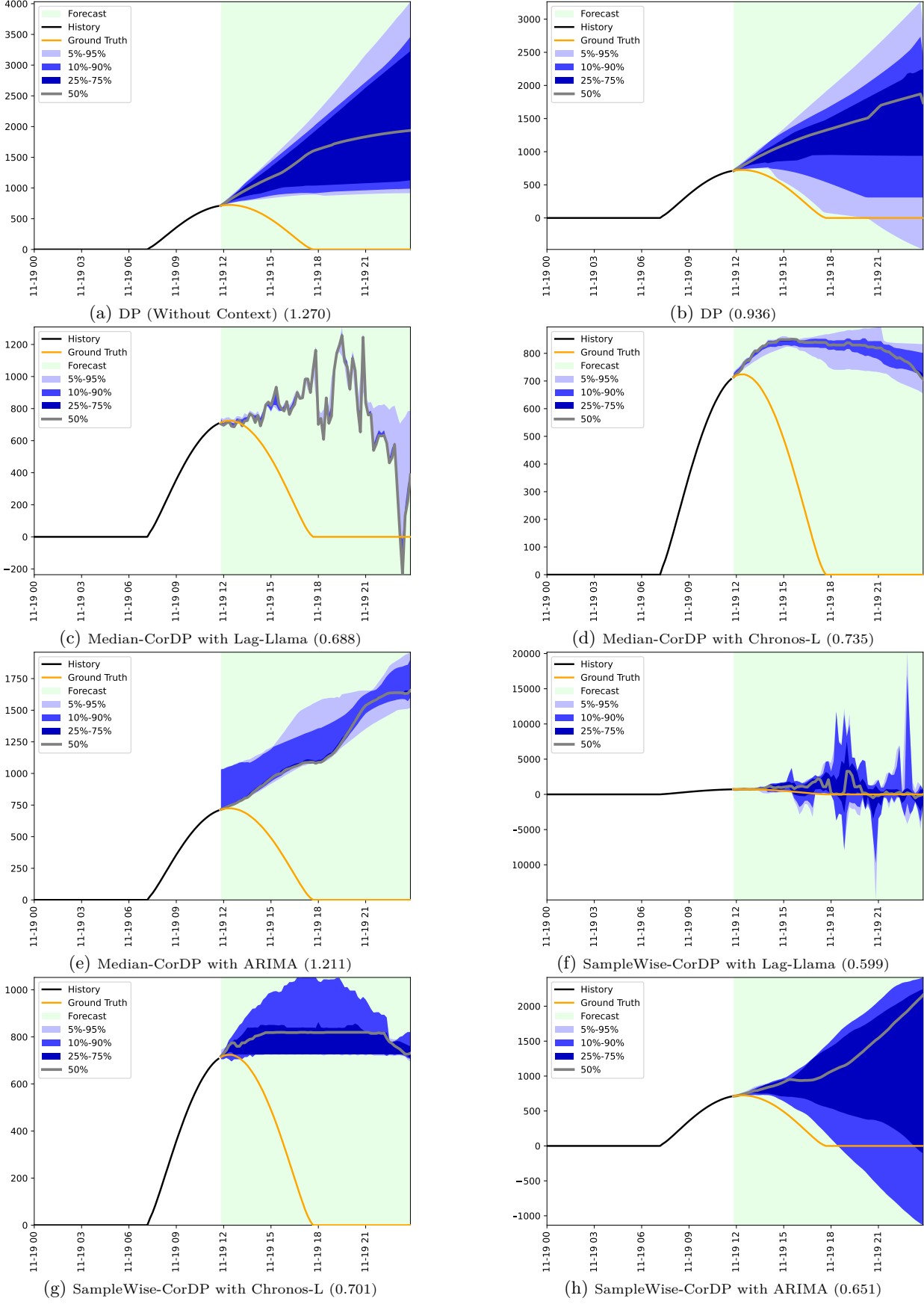

Figure 28: Forecasts of model Llama3.2-3B-Inst on task *ZenithInfoHalfDaySolarForecastTask* (with RCRPS in brackets)

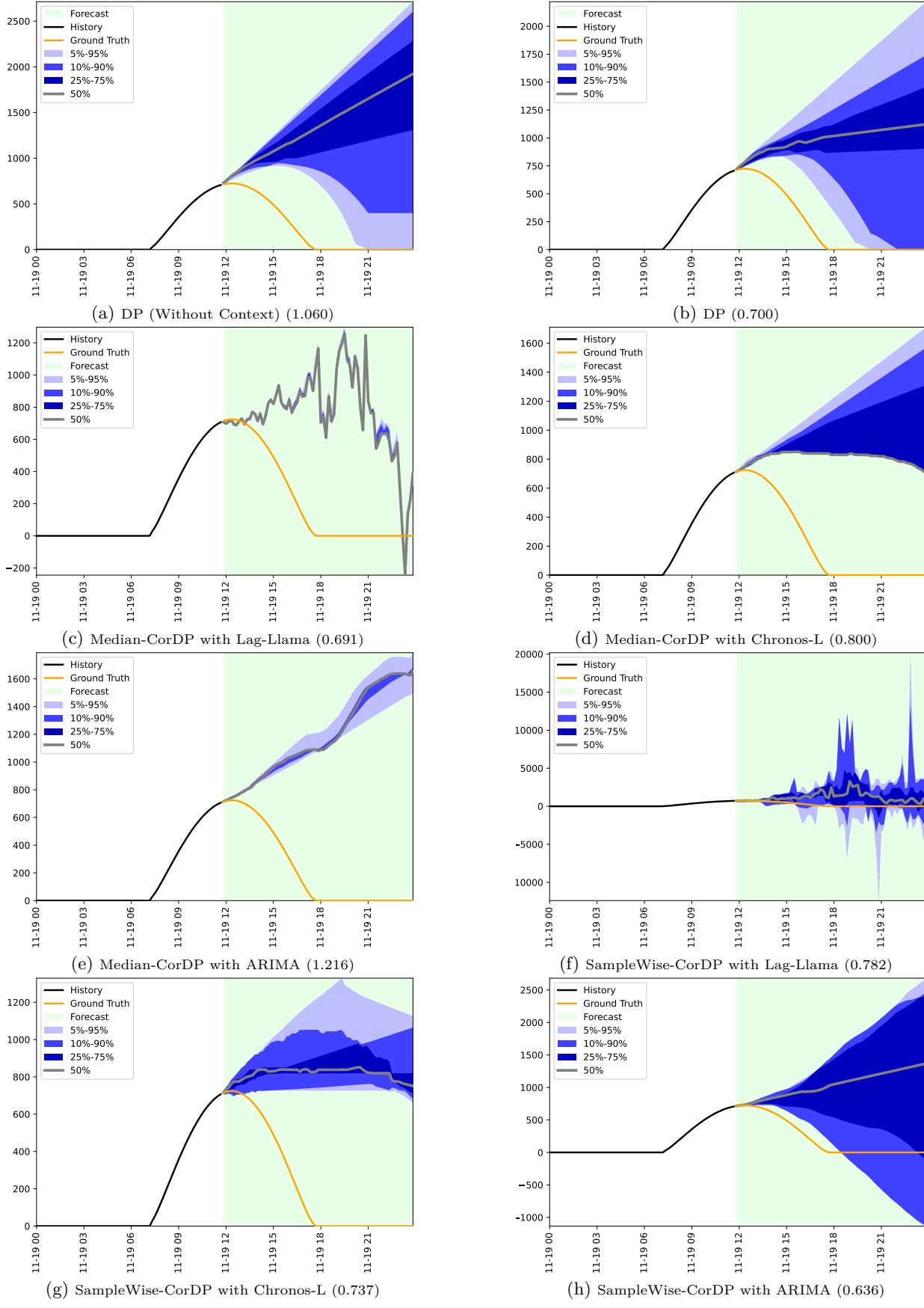

Figure 29: Forecasts of model Llama3-8B-Inst on task *ZenithInfoHalfDaySolarForecastTask* (with RCRPS in brackets)

### D.5.5 Task: *BoundedPredConstraintsBasedOnPredQuantilesTask*

**Context:**
Background: None
Constraints: Suppose that in the forecast, the values are bounded above by 6.29.
Scenario: None

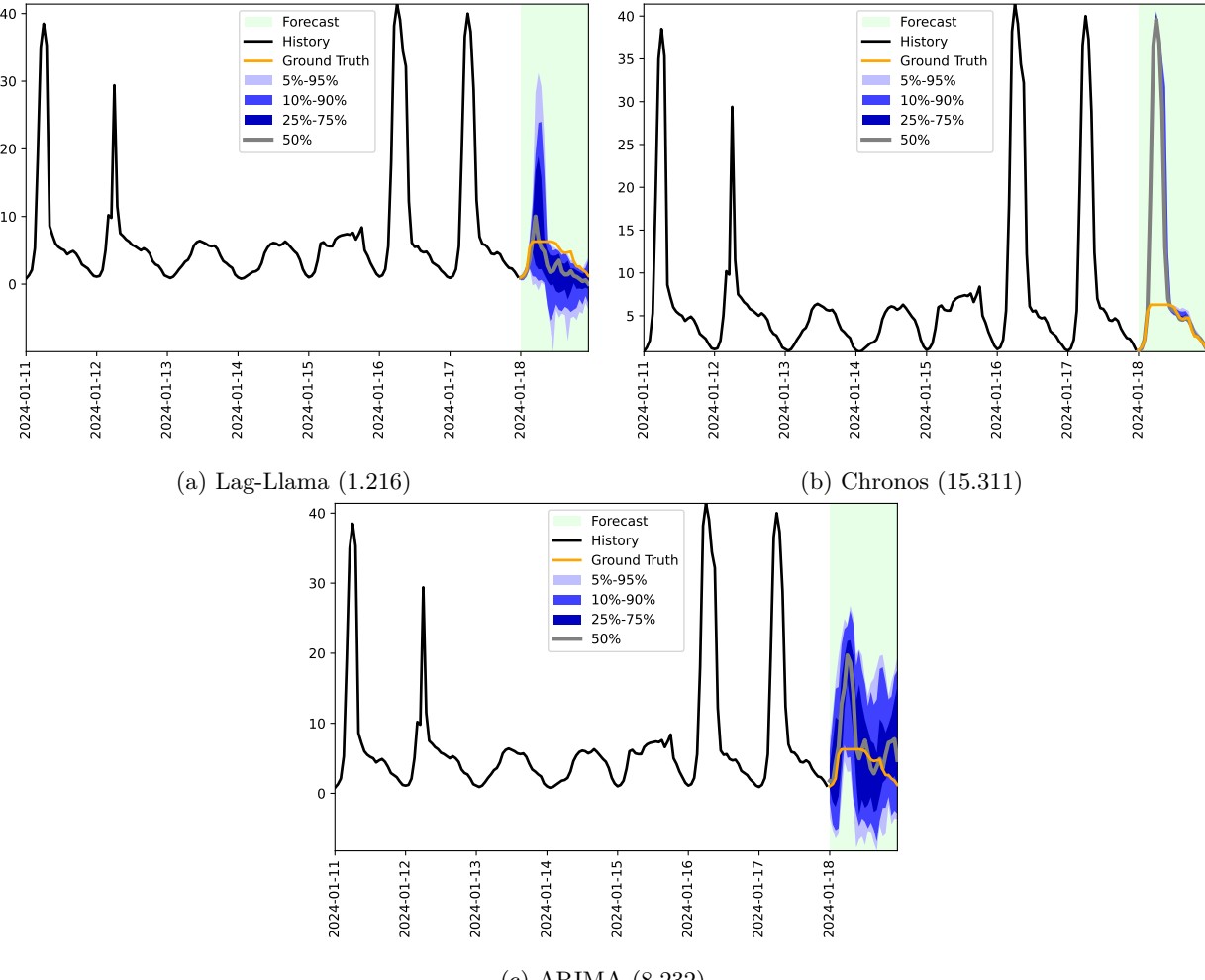

(a) Lag-Llama (1.216)

(b) Chronos (15.311)

(c) ARIMA (8.232)

Figure 30: Forecasts of Lag-Llama, Chronos, and ARIMA on the *BoundedPredConstraintsBasedOnPredQuantilesTask* task (with RCRPS in brackets)

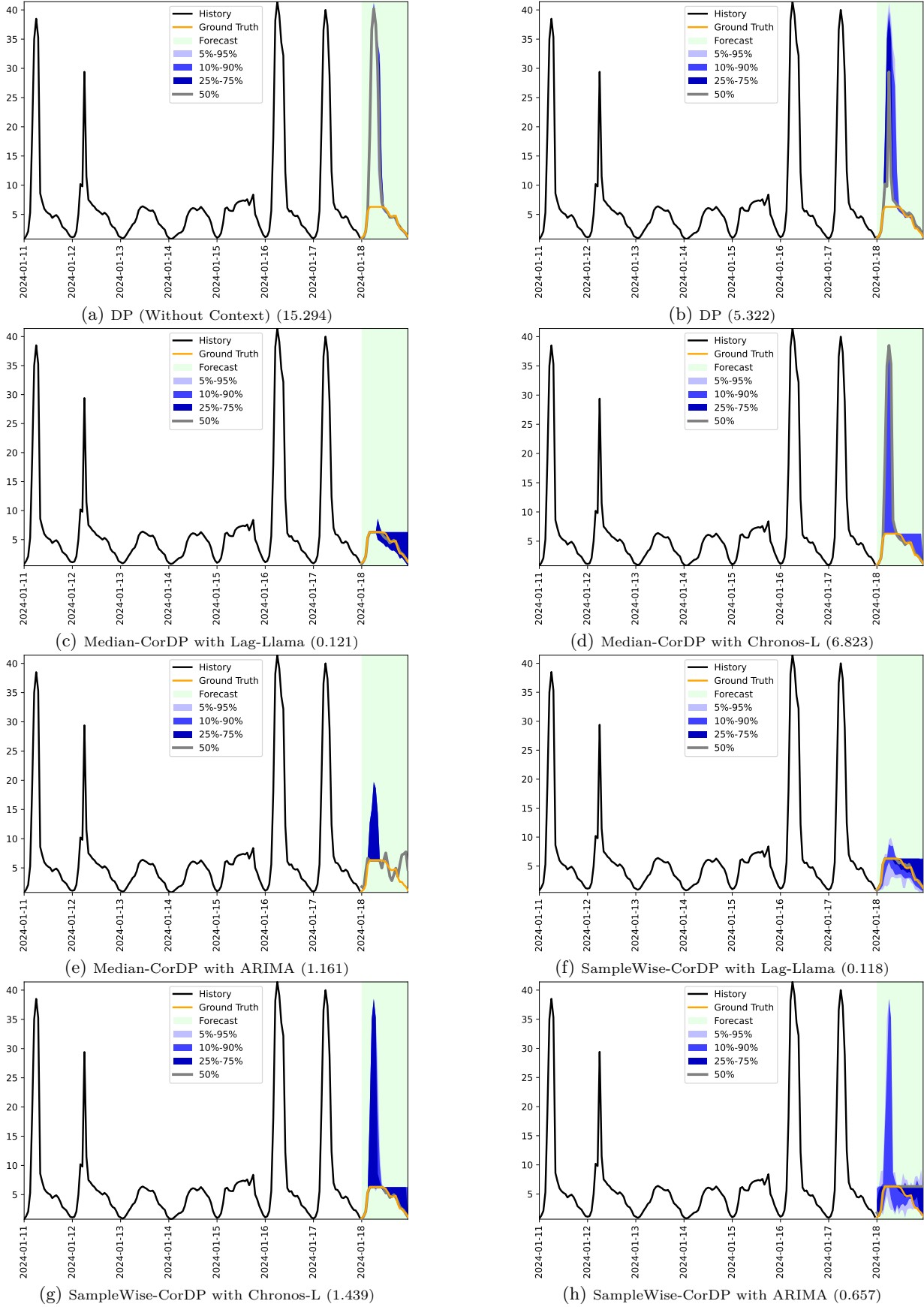

Figure 31: Forecasts of model Qwen2.5-7B-Inst on task *BoundedPredConstraintsBasedOnPredQuantilesTask* (with RCRPS in brackets)

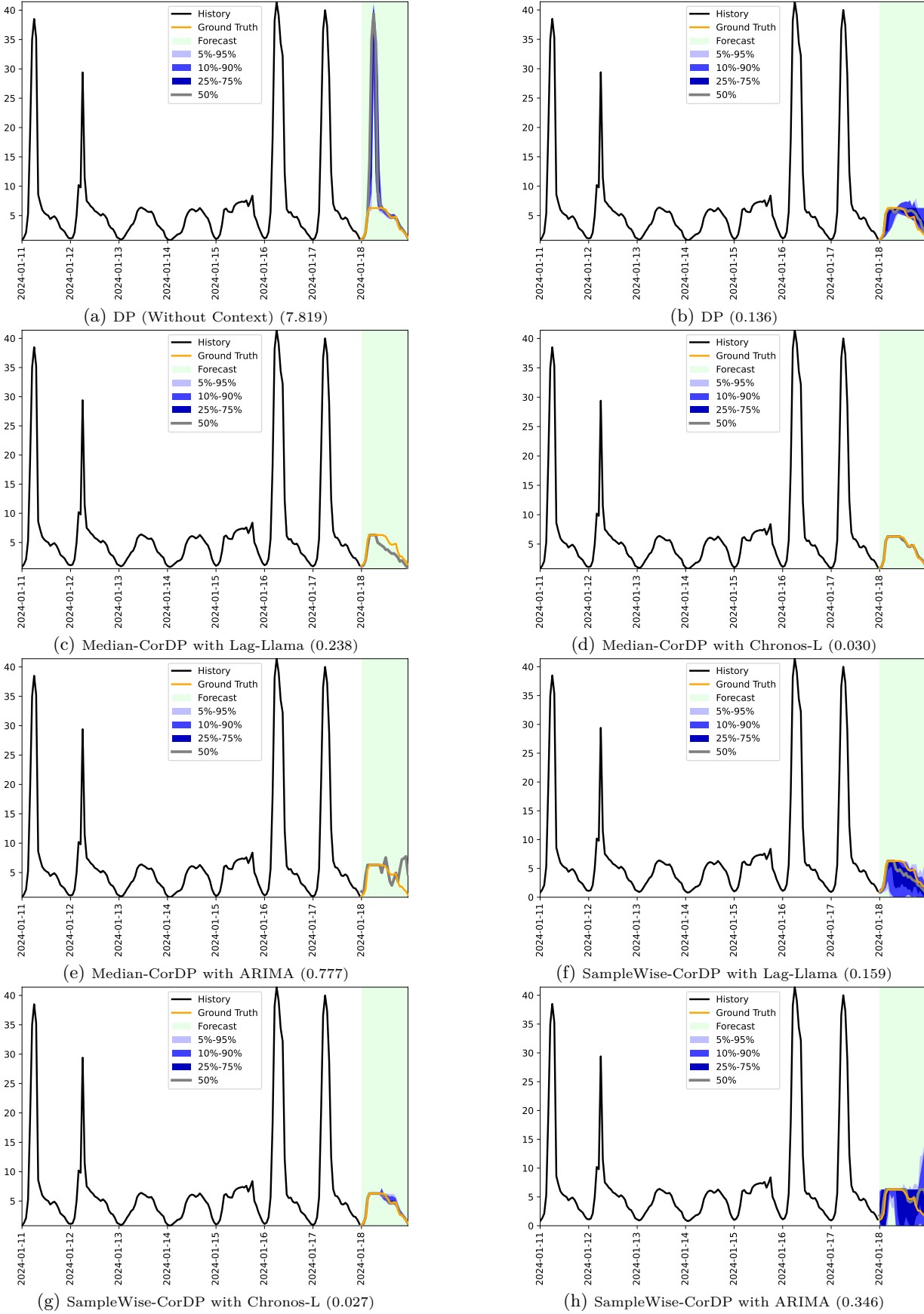

Figure 32: Forecasts of model Qwen2.5-14B-Inst on task *BoundedPredConstraintsBasedOnPredQuantilesTask* (with RCRPS in brackets)

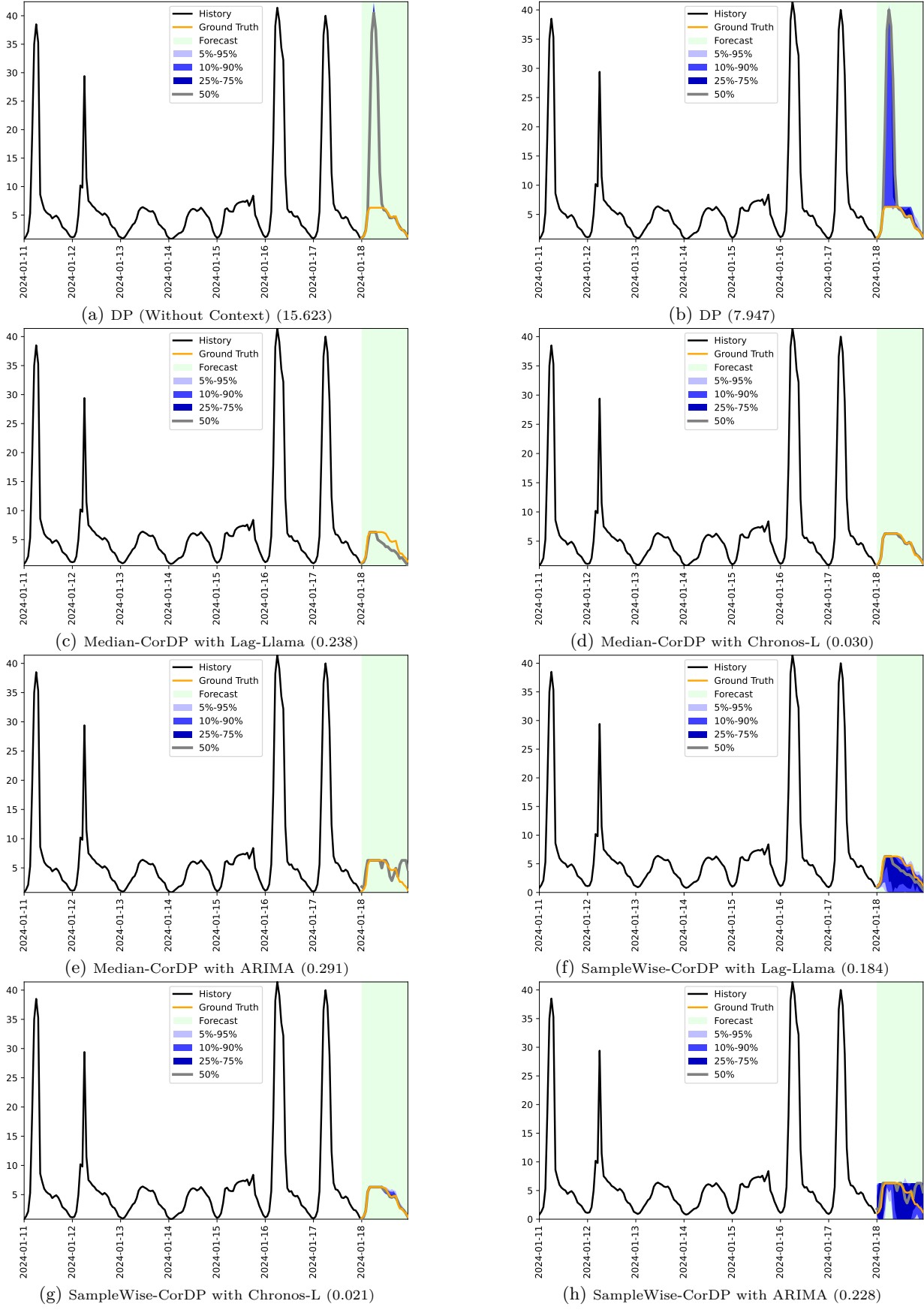

Figure 33: Forecasts of model Qwen2.5-32B-Inst on task *BoundedPredConstraintsBasedOnPredQuantilesTask* (with RCRPS in brackets)

# E  Additional Details on IC-DP

## E.1  IC-DP Prompt

We use the following prompt for the IC-DP method, where **{example_task_instance.background}**, **{example_task_instance.constraints}**, and **{example_task_instance.scenario}** are replaced by the background, constraints and scenario portions of the context of the example task respectively. **{example_task_history}**, **{example_pred_time}** and **{example_task_future}** are replaced by the history, prediction timestamps and the ground truth future values of the examples tasks respectively. **{history}** is replaced by the respective numerical history for the task instance in the format (timestamp, value), and **{context}** is replaced by the respective textual context for the task instance, and **((pred_time))** is replaced with the prediction timesteps. Although this prompt is specialized to the CiK benchmark where contexts are made up of background, constraints and scenario parts, the prompt can be generalized to use any kind of text context.

```
I have a context-aided time series forecasting task for you, where you will be given the
    history of a time series and additional context information, and prediction timesteps for
    which a forecast is required. You are expected to factor in any background knowledge,
satisfy any constraints, and respect any scenarios given in the context, and output the
    forecast.
in (timestamp, value) format in between <forecast> and </forecast> tags. You are to not
    include any other information (e.g., comments) in the forecast.

Here is the prompt for an example task:

Here is the context:
<context>\nBackground: {example_task_instance.background}\nConstraints:
    {example_task_instance.constraints}\nScenario:
    {example_task_instance.scenario}\n\n</context>\n\nHere is a historical time series in
    (timestamp, value) format:\n<history>{example_task_history}</history>\n\nNow please
    predict the value at the following timestamps: {example_pred_time}.\n

The expected output would be:
<forecast>{example_task_future}</forecast>

Note how the context was incorporated in the forecast. You are expected to do the same.
Here is the problem for which you need to return a forecast:

Here is some context about the task.
<context>
{context}
</context>

Here is a historical time series in (timestamp, value) format:
<history>
{history}
</history>

Now please predict the value at the following timestamps: {pred_time}.

Return the forecast in (timestamp, value) format in between <forecast> and </forecast> tags.
Do not include any other information (e.g., comments) in the forecast.
```

### E.2 Aggregate Results

Table 14 displays the aggregate results of models on CiK, comparing IC-DP and DP.

| Model | DP | IC-DP |
|---|---|---|
| Llama3.2-1B-Inst | $0.396 \pm 0.027$ | $\mathbf{0.337 \pm 0.009}$ |
| Llama3.2-3B-Inst | $0.687 \pm 0.025$ | $\mathbf{0.476 \pm 0.018}$ |
| Qwen2.5-0.5B-Inst | $0.592 \pm 0.027$ | $\mathbf{0.305 \pm 0.006}$ |
| Qwen2.5-1.5B-Inst | $0.616 \pm 0.018$ | $\mathbf{0.273 \pm 0.008}$ |
| Qwen2.5-3B-Inst | $0.424 \pm 0.017$ | $\mathbf{0.298 \pm 0.011}$ |
| Qwen2.5-7B-Inst | $0.401 \pm 0.006$ | $\mathbf{0.264 \pm 0.012}$ |
| Qwen2.5-14B-Inst | $\mathbf{0.247 \pm 0.006}$ | $0.270 \pm 0.005$ |
| Qwen2.5-32B-Inst | $0.397 \pm 0.008$ | $\mathbf{0.245 \pm 0.027}$ |
| Qwen2.5-72B-Inst | $0.202 \pm 0.009$ | $\mathbf{0.180 \pm 0.014}$ |
| Llama3.3-70B-Inst | $0.230 \pm 0.006$ | $\mathbf{0.168 \pm 0.006}$ |
| Llama3.1-405B-Inst | $0.173 \pm 0.003$ | $\mathbf{0.129 \pm 0.004}$ |
| GPT-4o | $0.317 \pm 0.009$ | $\mathbf{0.164 \pm 0.005}$ |
| GPT-4o-mini | $0.389 \pm 0.010$ | $\mathbf{0.253 \pm 0.004}$ |
| GPT-5.2 | $0.271 \pm 0.001$ | $\mathbf{0.260 \pm 0.001}$ |
| Gemini-2.5-Pro | $0.108 \pm 0.002$ | $\mathbf{0.100 \pm 0.001}$ |
| Claude-Sonnet-4.5 | $\mathbf{0.114 \pm 0.001}$ | $\mathbf{0.114 \pm 0.001}$ |

Table 14: Results of models with IC-DP on CiK. The best-performing method with each model is in **bold**.

### E.3 Results partitioned by task type

### E.4 Example Forecasts

| Model | RoI RCRPS | | non-RoI RCRPS | | RCRPS of tasks with full RoI | | Constraints RCRPS | |
|---|---|---|---|---|---|---|---|---|
| | DP | IC-DP | DP | IC-DP | DP | IC-DP | DP | IC-DP |
| Llama3.2-1B-Inst | $0.336 \pm 0.026$ | $\mathbf{0.218 \pm 0.006}$ | $0.248 \pm 0.026$ | $\mathbf{0.187 \pm 0.006}$ | $0.467 \pm 0.041$ | $\mathbf{0.428 \pm 0.015}$ | $0.275 \pm 0.092$ | $\mathbf{0.007 \pm 0.031}$ |
| Llama3.2-3B-Inst | $0.281 \pm 0.013$ | $\mathbf{0.147 \pm 0.005}$ | $\mathbf{0.162 \pm 0.013}$ | $0.209 \pm 0.005$ | $1.004 \pm 0.040$ | $\mathbf{0.679 \pm 0.031}$ | $1.030 \pm 0.090$ | $\mathbf{0.163 \pm 0.068}$ |
| Llama3.3-70B-Inst | $\mathbf{0.105 \pm 0.003}$ | $0.134 \pm 0.003$ | $0.182 \pm 0.003$ | $\mathbf{0.122 \pm 0.003}$ | $0.289 \pm 0.011$ | $\mathbf{0.194 \pm 0.010}$ | $\mathbf{0.000 \pm 0.024}$ | $0.025 \pm 0.020$ |
| Llama3.1-405B-Inst | $0.126 \pm 0.004$ | $\mathbf{0.094 \pm 0.004}$ | $0.150 \pm 0.004$ | $\mathbf{0.115 \pm 0.004}$ | $0.196 \pm 0.005$ | $\mathbf{0.146 \pm 0.006}$ | $0.004 \pm 0.009$ | $\mathbf{0.000 \pm 0.012}$ |
| Qwen2.5-0.5B-Inst | $0.339 \pm 0.010$ | $\mathbf{0.288 \pm 0.004}$ | $\mathbf{0.129 \pm 0.010}$ | $0.209 \pm 0.004$ | $0.836 \pm 0.046$ | $\mathbf{0.343 \pm 0.010}$ | $0.243 \pm 0.103$ | $\mathbf{0.005 \pm 0.020}$ |
| Qwen2.5-1.5B-Inst | $0.317 \pm 0.020$ | $\mathbf{0.224 \pm 0.008}$ | $0.224 \pm 0.020$ | $\mathbf{0.163 \pm 0.009}$ | $0.851 \pm 0.026$ | $\mathbf{0.327 \pm 0.011}$ | $0.706 \pm 0.147$ | $\mathbf{0.023 \pm 0.023}$ |
| Qwen2.5-3B-Inst | $0.269 \pm 0.015$ | $\mathbf{0.265 \pm 0.009}$ | $0.186 \pm 0.015$ | $\mathbf{0.180 \pm 0.009}$ | $0.558 \pm 0.027$ | $\mathbf{0.349 \pm 0.017}$ | $0.234 \pm 0.056$ | $\mathbf{0.031 \pm 0.039}$ |
| Qwen2.5-7B-Inst | $0.285 \pm 0.006$ | $\mathbf{0.164 \pm 0.007}$ | $\mathbf{0.164 \pm 0.006}$ | $0.187 \pm 0.007$ | $0.521 \pm 0.009$ | $\mathbf{0.325 \pm 0.020}$ | $0.470 \pm 0.078$ | $\mathbf{0.063 \pm 0.045}$ |
| Qwen2.5-14B-Inst | $0.162 \pm 0.005$ | $\mathbf{0.099 \pm 0.003}$ | $\mathbf{0.146 \pm 0.005}$ | $0.148 \pm 0.003$ | $\mathbf{0.310 \pm 0.010}$ | $0.369 \pm 0.008$ | $\mathbf{0.039 \pm 0.015}$ | $0.455 \pm 0.009$ |
| Qwen2.5-32B-Inst | $\mathbf{0.116 \pm 0.001}$ | $0.129 \pm 0.003$ | $0.140 \pm 0.001$ | $\mathbf{0.133 \pm 0.003}$ | $0.580 \pm 0.013$ | $\mathbf{0.323 \pm 0.045}$ | $0.479 \pm 0.019$ | $\mathbf{0.186 \pm 0.103}$ |
| Qwen2.5-72B-Inst | $\mathbf{0.115 \pm 0.004}$ | $0.125 \pm 0.003$ | $0.138 \pm 0.004$ | $\mathbf{0.113 \pm 0.003}$ | $0.253 \pm 0.015$ | $\mathbf{0.221 \pm 0.023}$ | $\mathbf{0.032 \pm 0.028}$ | $0.068 \pm 0.052$ |
| GPT-4o | $\mathbf{0.123 \pm 0.004}$ | $0.125 \pm 0.004$ | $\mathbf{0.106 \pm 0.004}$ | $0.120 \pm 0.004$ | $0.455 \pm 0.014$ | $\mathbf{0.192 \pm 0.007}$ | $0.455 \pm 0.029$ | $\mathbf{0.004 \pm 0.014}$ |
| GPT-4o-mini | $0.263 \pm 0.005$ | $\mathbf{0.207 \pm 0.003}$ | $0.150 \pm 0.005$ | $\mathbf{0.167 \pm 0.003}$ | $0.513 \pm 0.017$ | $\mathbf{0.297 \pm 0.006}$ | $0.001 \pm 0.032$ | $\mathbf{0.000 \pm 0.010}$ |
| GPT-5.2 | $0.104 \pm 0.001$ | $\mathbf{0.095 \pm 0.001}$ | $\mathbf{0.095 \pm 0.001}$ | $0.099 \pm 0.001$ | $0.387 \pm 0.001$ | $\mathbf{0.372 \pm 0.001}$ | $\mathbf{0.000 \pm 0.002}$ | $0.470 \pm 0.004$ |
| Gemini-2.5-Pro | $\mathbf{0.112 \pm 0.001}$ | $0.116 \pm 0.001$ | $0.081 \pm 0.001$ | $\mathbf{0.071 \pm 0.001}$ | $0.116 \pm 0.003$ | $\mathbf{0.104 \pm 0.001}$ | $\mathbf{0.000 \pm 0.005}$ | $0.462 \pm 0.006$ |
| Claude-Sonnet-4.5 | $\mathbf{0.108 \pm 0.001}$ | $\mathbf{0.108 \pm 0.001}$ | $\mathbf{0.090 \pm 0.001}$ | $0.101 \pm 0.001$ | $0.124 \pm 0.002$ | $\mathbf{0.121 \pm 0.001}$ | $\mathbf{0.000 \pm 0.003}$ | $0.488 \pm 0.010$ |

Table 15: Results of models with IC-DP in various groups of tasks in CiK. The best-performing method with each model in every group is in **bold**.

### E.4.1   Task: *ElectricityIncreaseInPredictionWithDistractorWithDates*

Background: This is the electricity consumption recorded in Kilowatt (kW) in city A.
Constraints: None.
Scenario: There was a festival in neighbouring cities B and C that resulted in 10 times the usual electricity being consumed there from 2013-05-28 12:00:00 for 2 hours. But this did not affect electricity consumption in city A. Suppose that there is a heat wave in city A from 2013-05-28 12:00:00 for 2 hours, leading to excessive use of air conditioning, and 4 times the usual electricity being consumed.

Figure 34: Context

Background: This is the electricity consumption recorded in Kilowatt (kW) in city A.
Constraints: None
Scenario: A brief technical issue in the electricity grid in a nearby city caused a major dip of 75% from 2014-03-24 13:00:00 for 2 hours. This issue has affected many nearby cities, but not this city.Suppose that there is a heat wave in city A from 2014-03-24 13:00:00 for 2 hours, leading to excessive use of air conditioning, and 4 times the usual electricity being consumed.

(a) Context of the In-Context Example Task used with IC-DP experiments

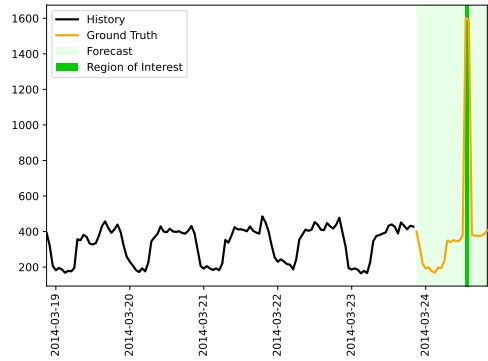

(b) Historical and Future Data of the In-Context Example Task used with IC-DP experiments

Figure 35: In-Context Example Task used with IC-DP experiments

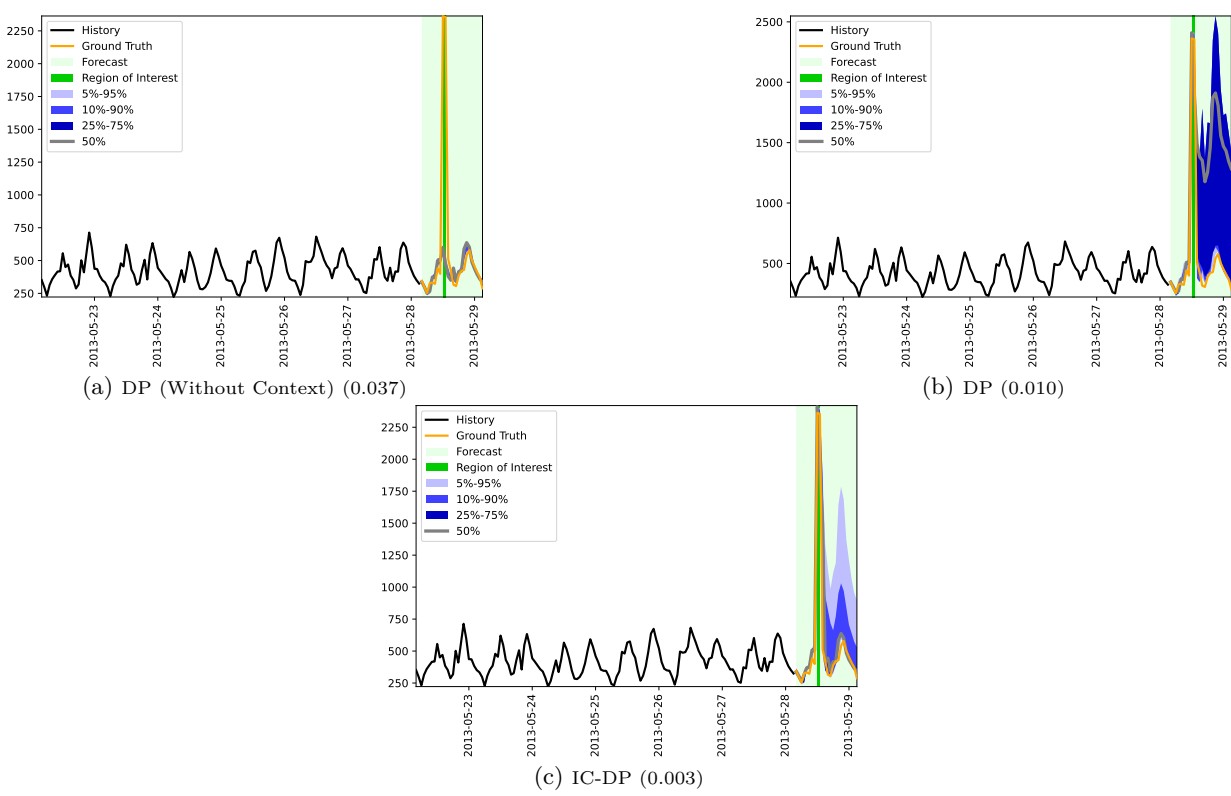

Figure 36: Forecasts of model Qwen2.5-72B-Inst on task *ElectricityIncreaseInPredictionWithDistractorWith-Dates* (with RCRPS in brackets)

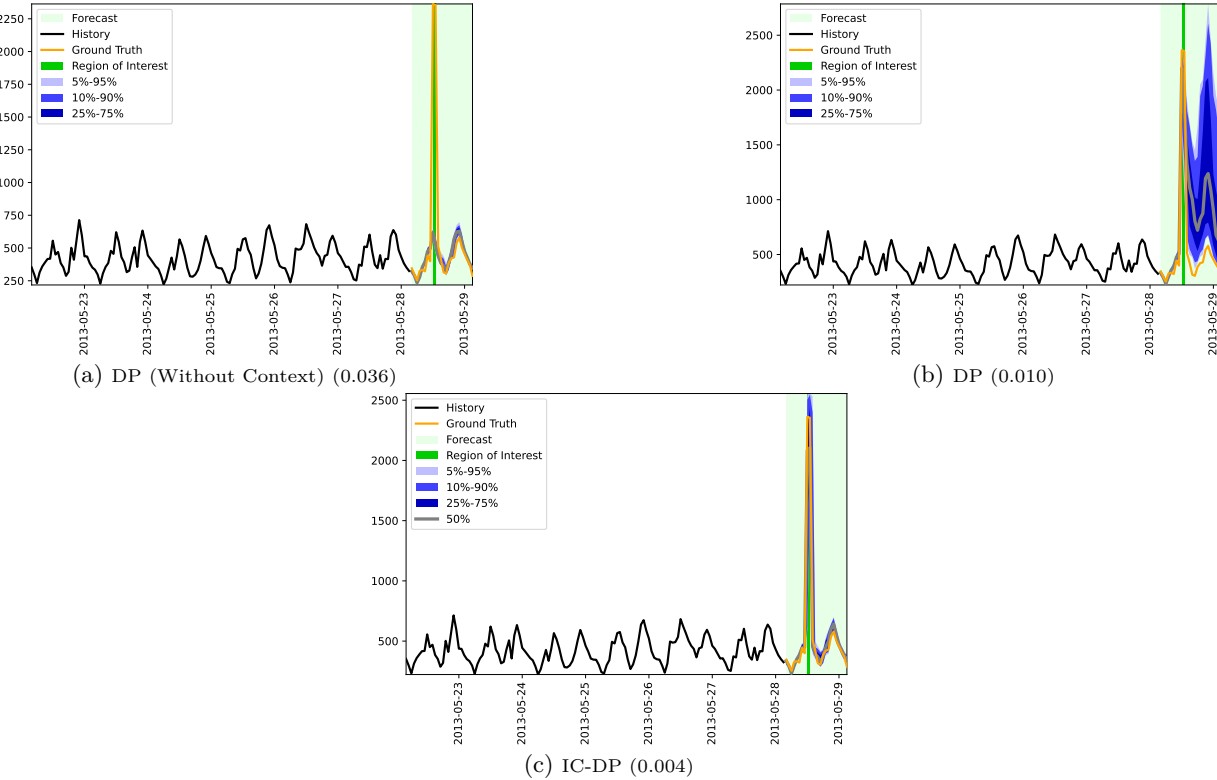

Figure 37: Forecasts of model Llama3.1-405B-Inst on task *ElectricityIncreaseInPredictionWithDistractor-WithDates* (with RCRPS in brackets)

### E.4.2 Task: *SensorTrendAccumulationTask*

Background: This series represents the occupancy rate (%) captured by a highway sensor. The sensor had a calibration problem starting from 2024-01-11 12:00:00 which resulted in an additive trend in the series that increases by 0.0874 at every hour. At timestep 2024-01-18 00:00:00, the sensor was repaired and this additive trend will disappear.
Constraints: None
Scenario: None

Figure 38: Context

Background: This series represents the occupancy rate (%) captured by a highway sensor. The sensor had a calibration problem starting from 2024-02-12 13:00:00 which resulted in an additive trend in the series that increases by 0.0489 at every hour. At timestep 2024-02-16 20:00:00, the sensor was repaired and this additive trend will disappear.
Constraints: None
Scenario: None

(a) Context of the In-Context Example Task used with IC-DP experiments

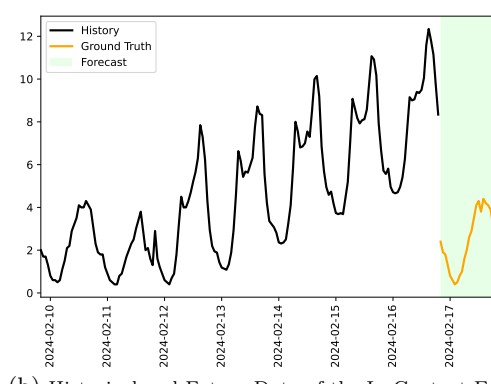

(b) Historical and Future Data of the In-Context Example Task used with IC-DP experiments

Figure 39: In-Context Example Task used with IC-DP experiments

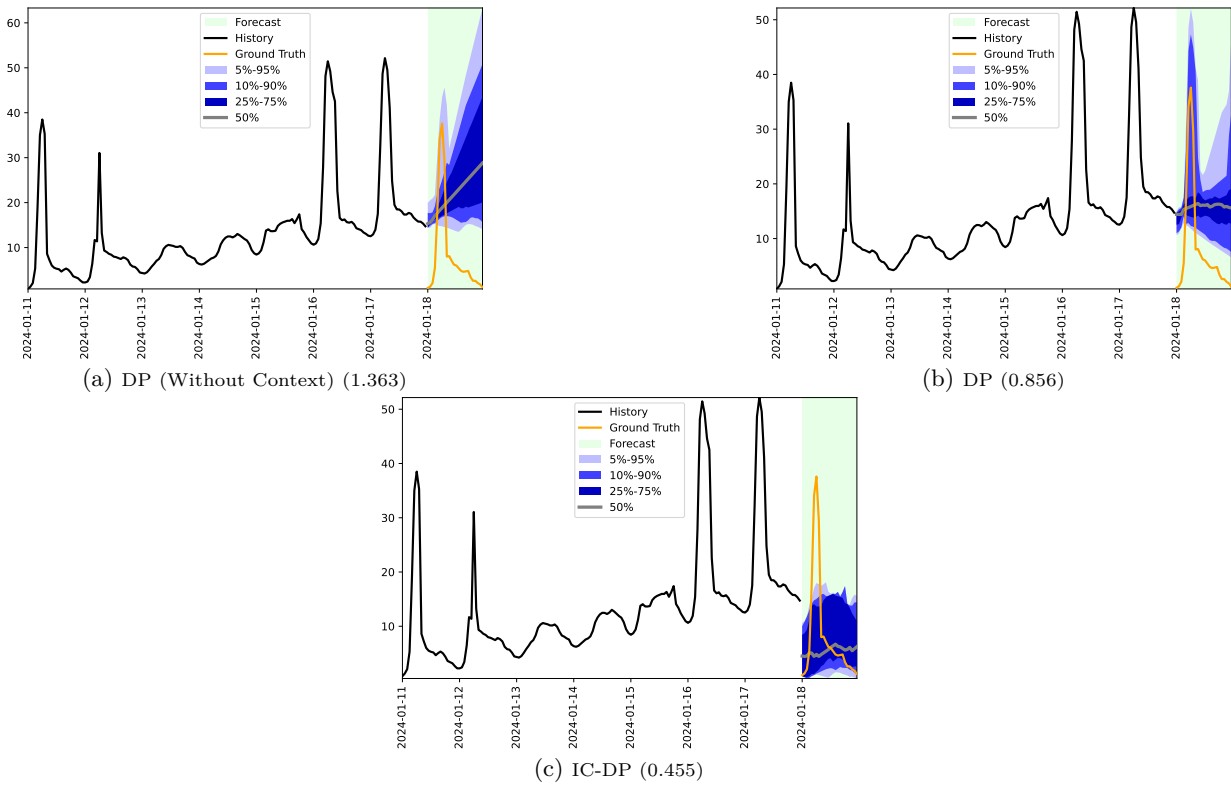

Figure 40: Forecasts of model Llama3-8B-Inst on task *SensorTrendAccumulationTask* (with RCRPS in brackets)

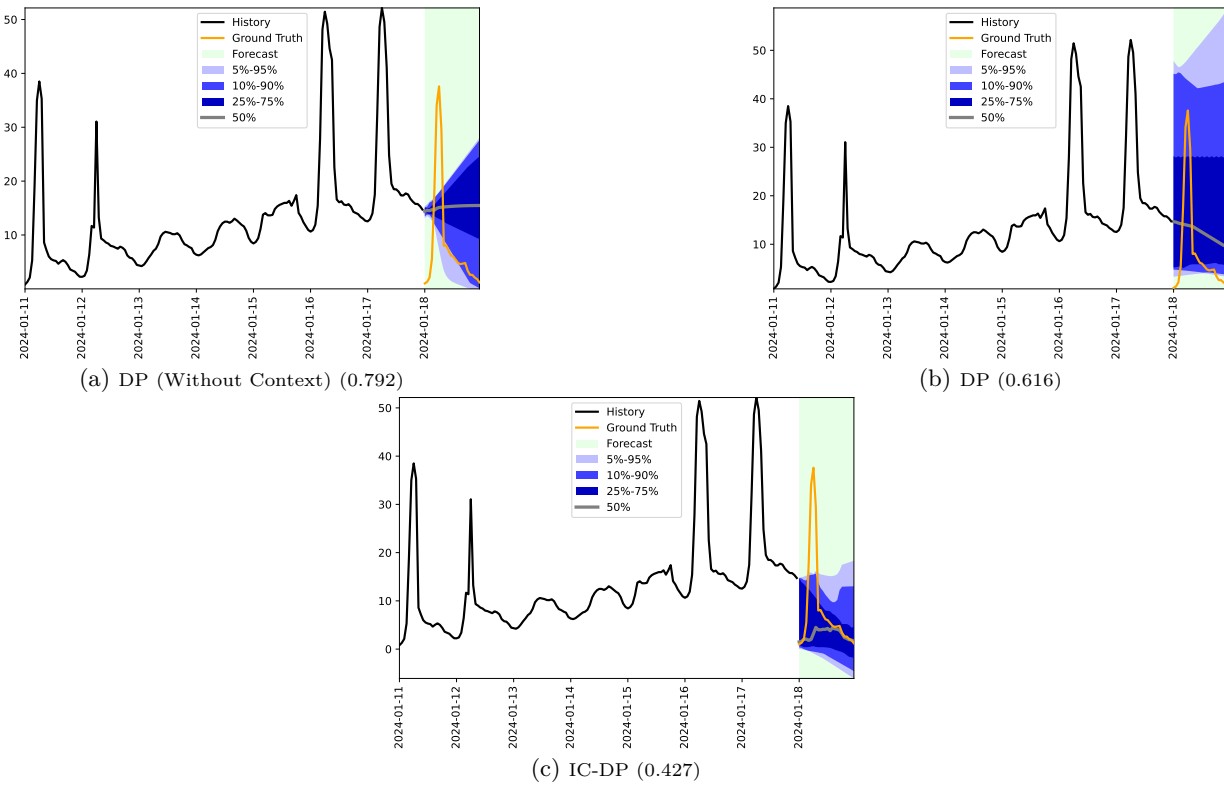

Figure 41: Forecasts of model Qwen2.5-3B-Inst on task *SensorTrendAccumulationTask* (with RCRPS in brackets)

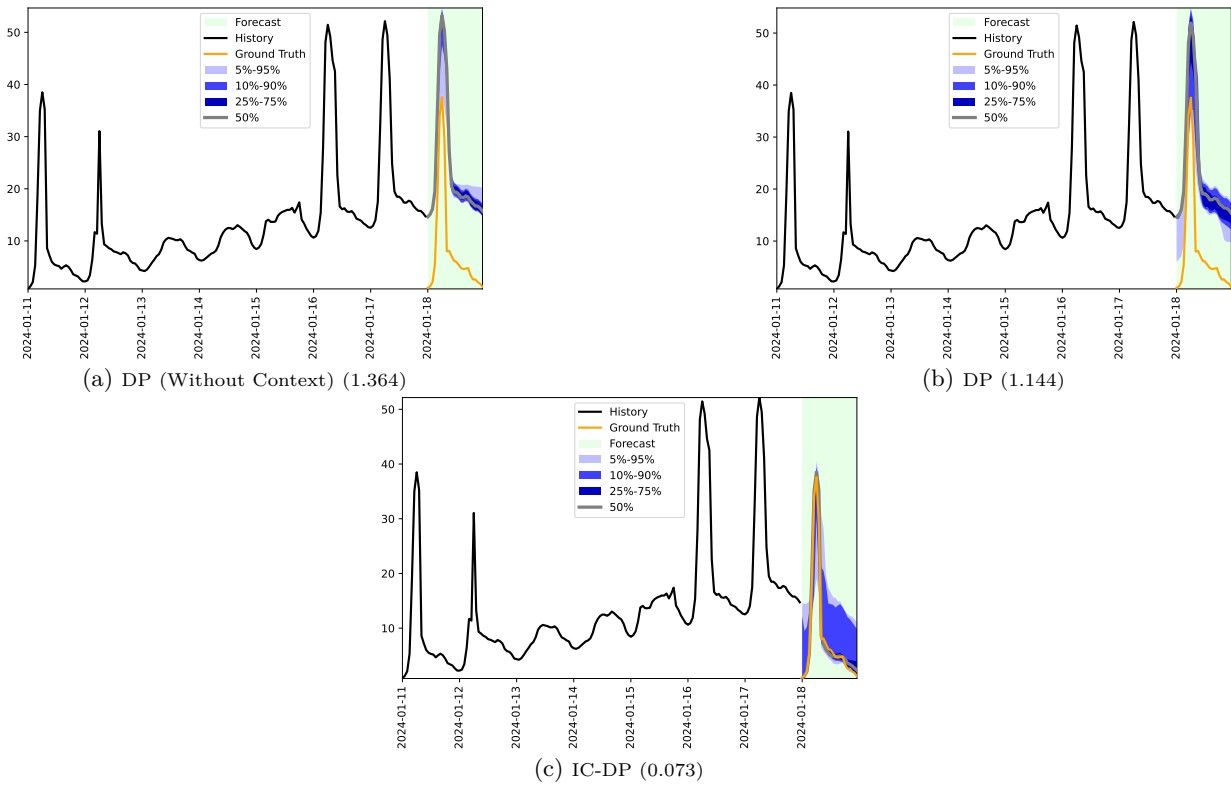

(a) DP (Without Context) (1.364)

(b) DP (1.144)

(c) IC-DP (0.073)

Figure 42: Forecasts of model Llama3.1-405B-Inst on task *SensorTrendAccumulationTask* (with RCRPS in brackets)

### E.4.3 Task: *BoundedPredConstraintsBasedOnPredQuantilesTask*

Background: None
Constraints: Suppose that in the forecast, the values are bounded above by 6.29.
Scenario: None

Figure 43: Context

Background: None
Constraints: Suppose that in the forecast, the values are bounded below by 1.57, the values are bounded above by 3.53.
Scenario: None

(a) Context of the In-Context Example Task used with IC-DP experiments

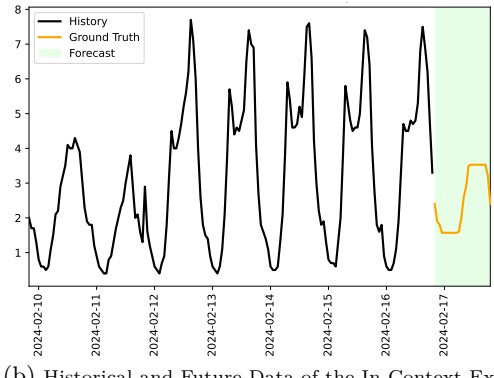

(b) Historical and Future Data of the In-Context Example Task used with IC-DP experiments

Figure 44: In-Context Example Task used with IC-DP experiments

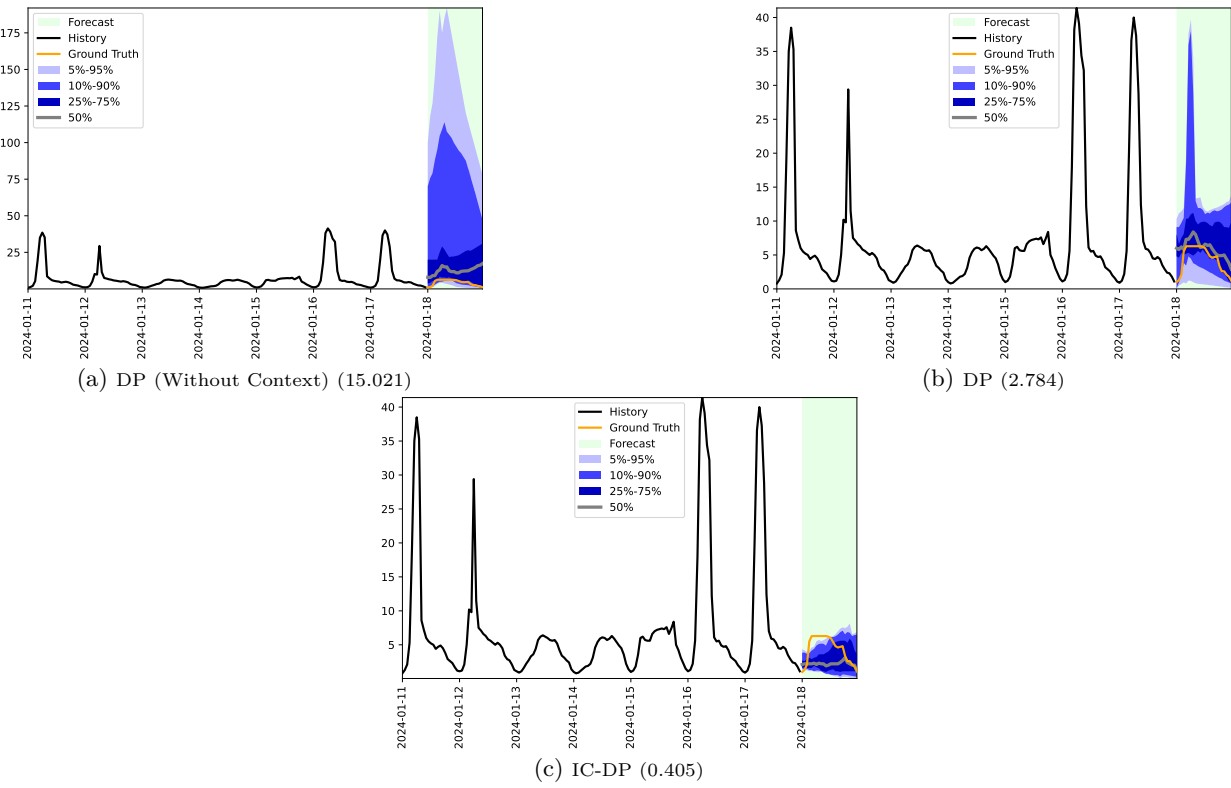

Figure 45: Forecasts of model Llama3.2-1B-Inst on task *BoundedPredConstraintsBasedOnPredQuantilesTask* (with RCRPS in brackets)

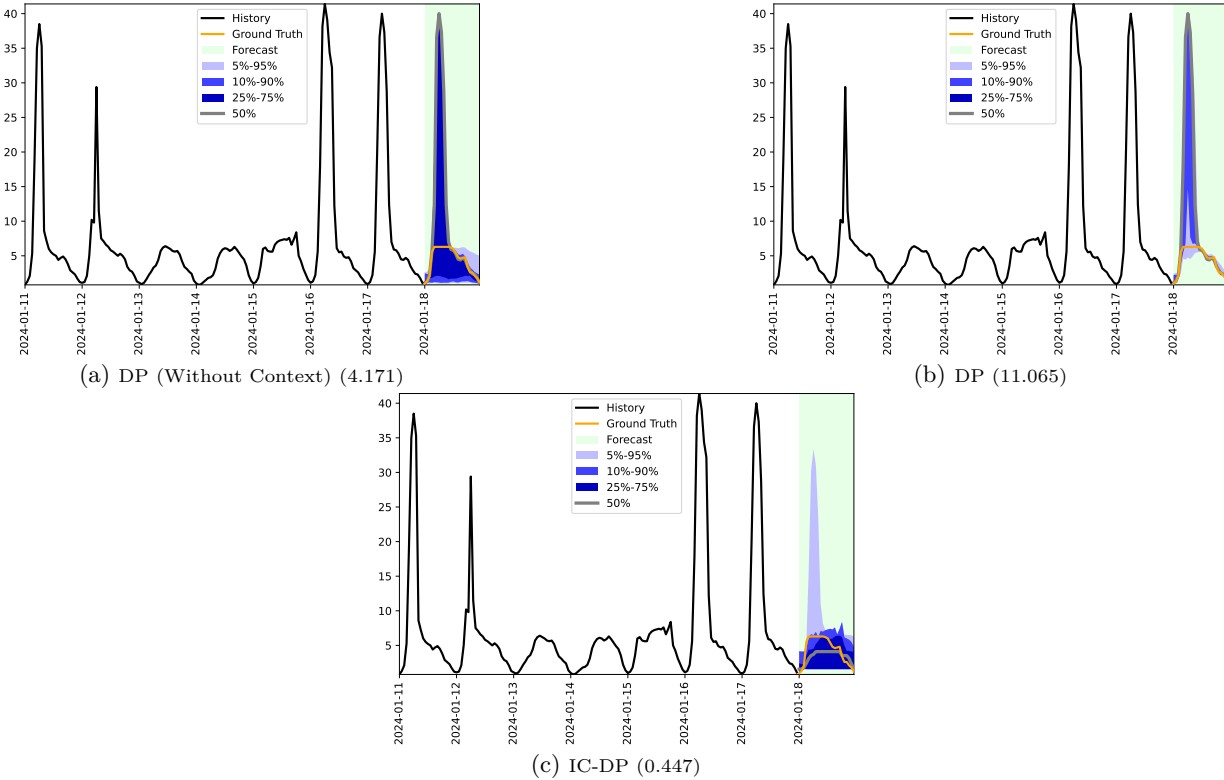

Figure 46: Forecasts of model Qwen2.5-1.5B-Inst on task *BoundedPredConstraintsBasedOnPredQuantilesTask* (with RCRPS in brackets)

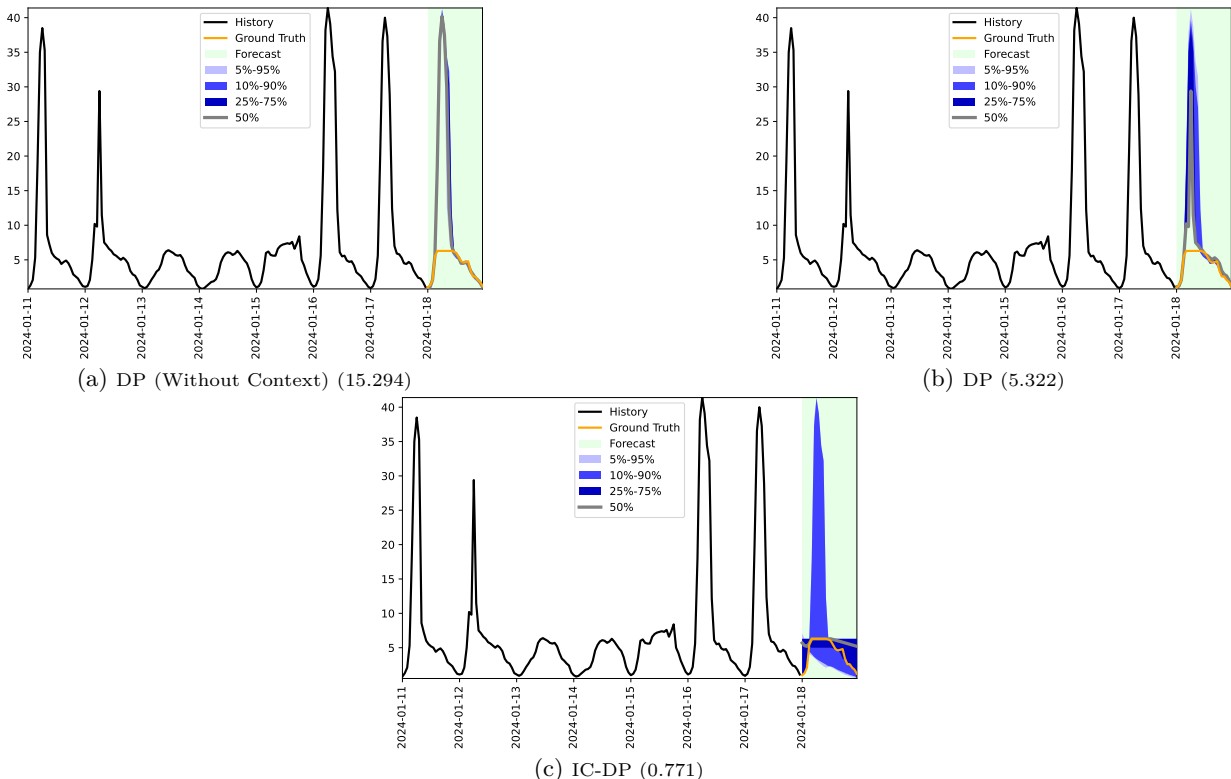

Figure 47: Forecasts of model Qwen2.5-7B-Inst on task *BoundedPredConstraintsBasedOnPredQuantilesTask* (with RCRPS in brackets)

# F    Additional Details on RouteDP

## F.1    RouteDP Prompt

To predict the difficulty of a task, we use the below prompt, where **{direct_prompt}** is replaced by the instantiated Direct Prompt (DP) prompt, which contains the context, historical time series and prediction timesteps of the task, as used in Williams et al. (2025).

```
{direct_prompt}
You are given a forecasting task with full contextual information.
Please rate the task as easy or hard.
Difficulty:
```

Given all 71 tasks from the CiK benchmark, we first run the designated Router LLM to predict the difficulty of a task. In particular, we use constrained decoding to limit the outputs to either "easy" or "hard".

Then, to route tasks, given a $k$ number of tasks to send to the large model, we dispatch the top-$k$ tasks considered hardest according to $P(hard)$ to the larger LLM, and dispatch the rest to the small model.

## F.2    Per-Task Routing via Threshold

While we evaluate RouteDP in a batch setting, the setup naturally extends to per-task routing. Since the router outputs a continuous score $P(hard) \in [0, 1]$, an operator can choose a threshold $\tau$ and route any task scoring above $\tau$ to the large model. This per-task threshold setting and the batch setting are equivalent: for a given routing budget, they route the same tasks and achieve the same RCRPS. The "fraction of tasks routed" columns in Table 2 correspond directly to exploring different percentile-based thresholds (20%, 40%, 60%, 80% quantiles of each router's score distribution).

Score ranges vary meaningfully across router sizes, as shown in Figure 48. Larger routers (e.g., Qwen2.5-32B: [0.03, 0.39], Qwen2.5-72B: [0.05, 0.95]) spread scores across their range, while smaller routers collapse tasks into a narrow band (e.g., Qwen2.5-0.5B: [0.76, 0.84]). Once a router is chosen, calibrating $\tau$ would require estimating the relevant percentile of that router's score distribution on a held-out set.

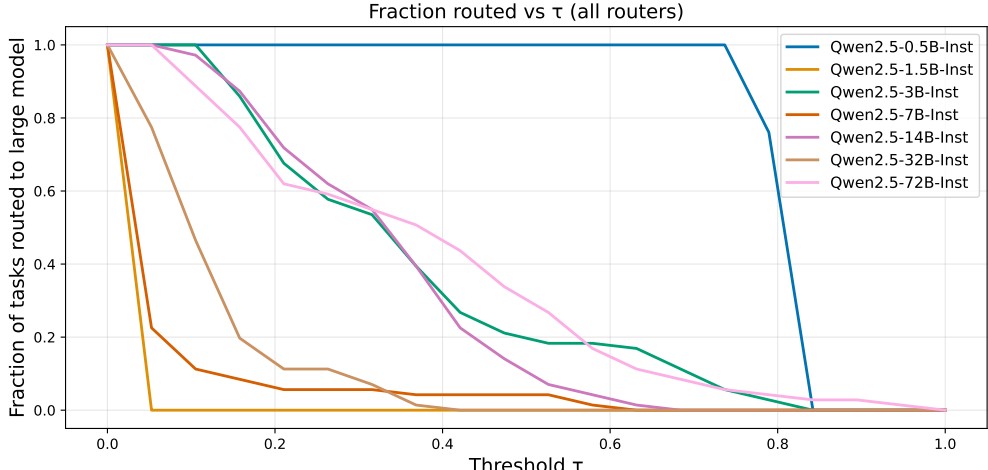

Figure 48: Fraction of tasks routed vs. threshold $\tau$ for different router models. Larger routers spread scores meaningfully across their range, while smaller routers collapse tasks into a narrow band.

## F.3 Extended Results

### F.4 Plots showcasing the area captured by different router models

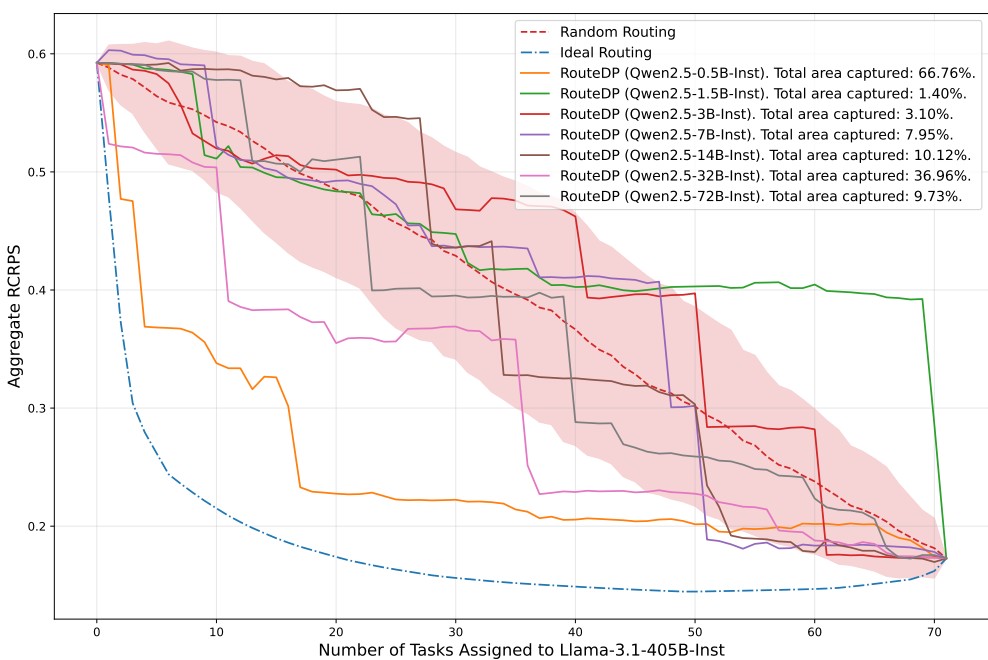

Figure 49: Random, ideal and router curves with Qwen2.5-0.5B-Inst as the small model. The shaded region represents the distribution of 100 random assignment trajectories.

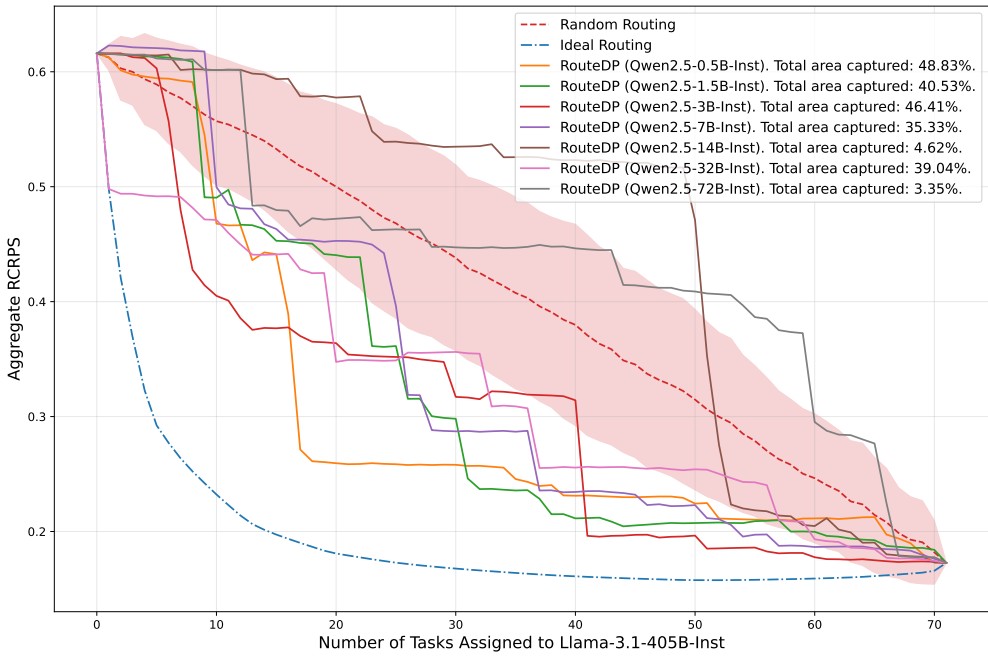

Figure 50: Random, ideal and router curves with Qwen2.5-1.5B-Inst as the small model. The shaded region represents the distribution of 100 random assignment trajectories.

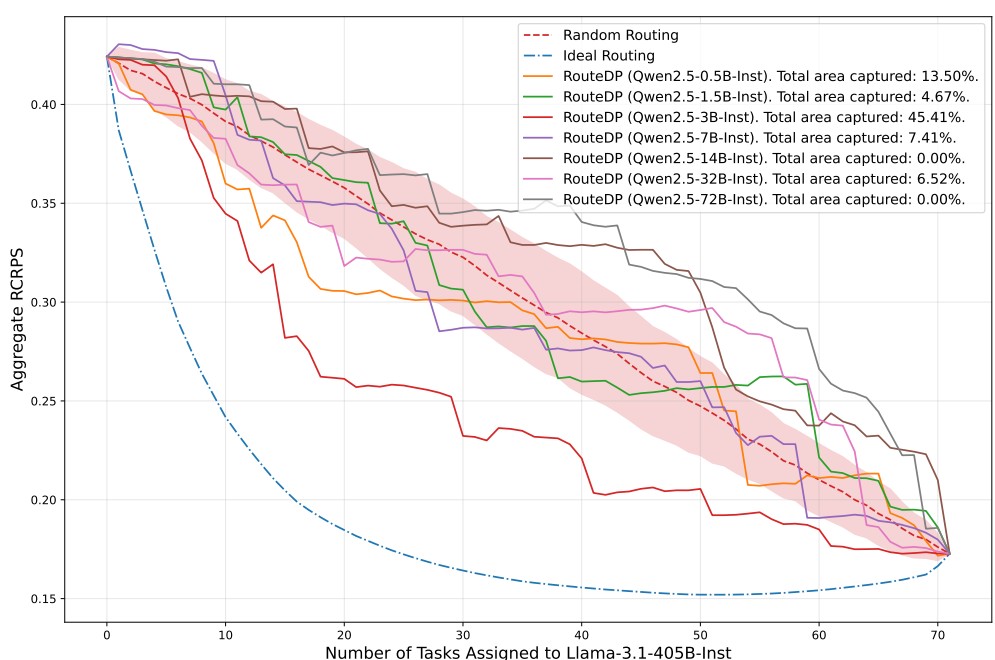

Figure 51: Random, ideal and router curves with Qwen2.5-3B-Inst as the small model. The shaded region represents the distribution of 100 random assignment trajectories.

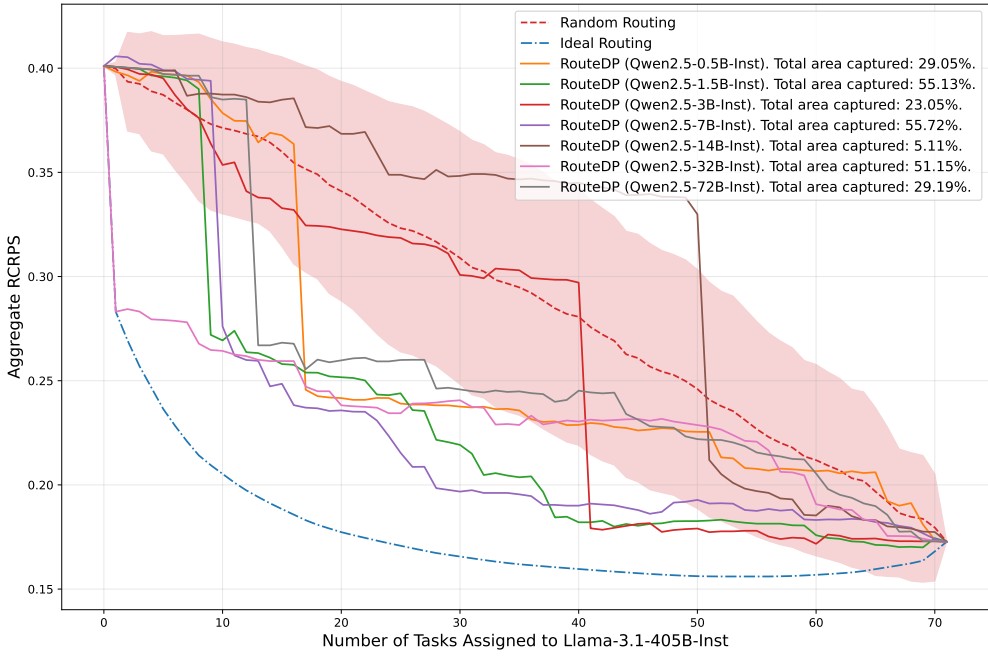

Figure 52: Random, ideal and router curves with Qwen2.5-7B-Inst as the small model. The shaded region represents the distribution of 100 random assignment trajectories.

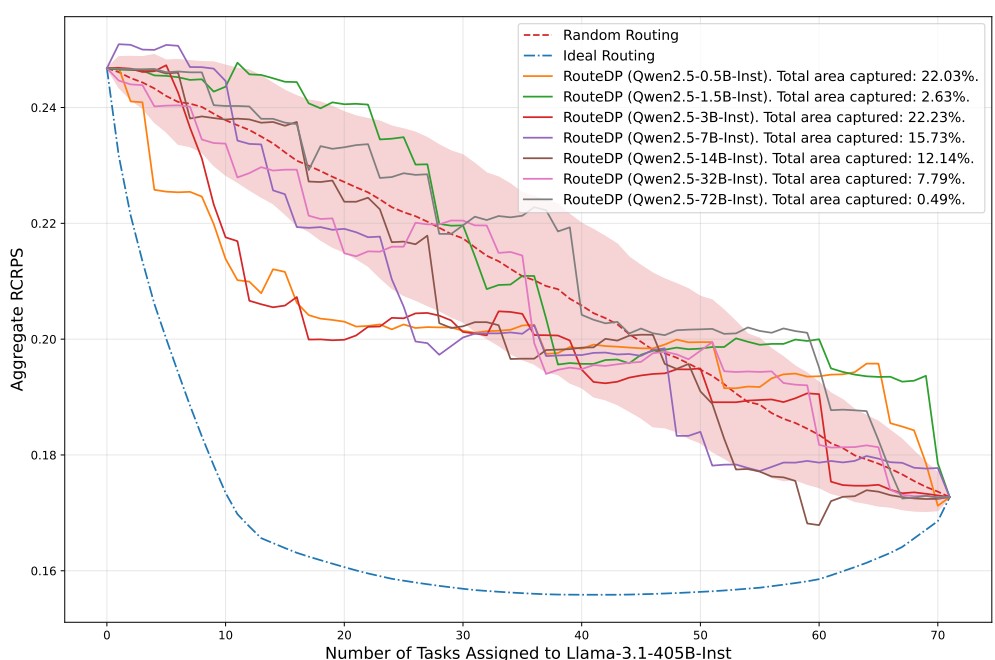

Figure 53: Random, ideal and router curves with Qwen2.5-14B-Inst as the small model. The shaded region represents the distribution of 100 random assignment trajectories.

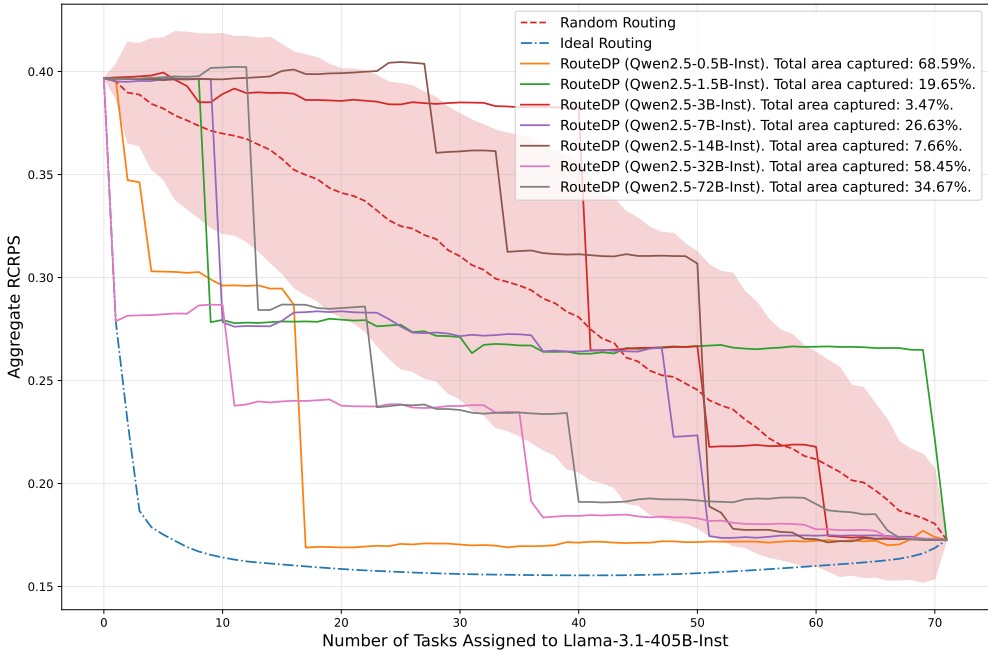

Figure 54: Random, ideal and router curves with Qwen2.5-32B-Inst as the small model. The shaded region represents the distribution of 100 random assignment trajectories.

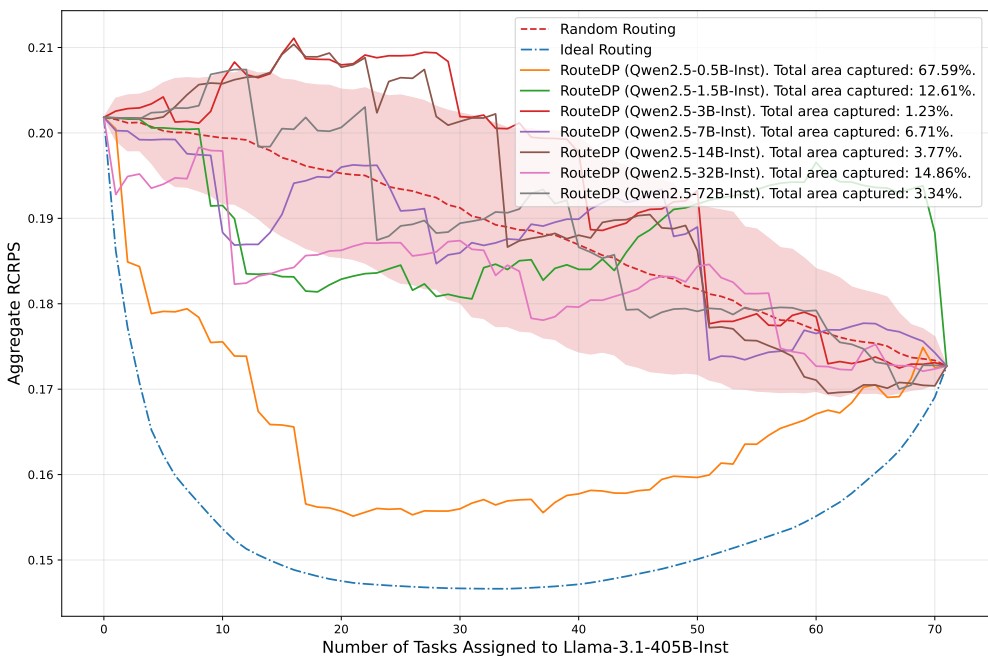

Figure 55: Random, ideal and router curves with Qwen2.5-72B-Inst as the small model. The shaded region represents the distribution of 100 random assignment trajectories.

## F.5 RouteDP with other methods

The advantage of the proposed RouteDP method is that the difficulty scores predicted by the router can, in principle, be used route tasks to models irrespective of what method the downstream model uses to obtain forecasts. To evaluate if RouteDP empirically improves the performance of downstream models for which methods other than DP were used to obtain forecasts, we test it with models that used IC-DP and CorDP to obtain forecasts. We call these methods Route-IC-DP and Route-Cor-DP respectively, indicating that the routing is done on IC-DP and CorDP forecasts respectively. For CorDP, we evaluate it with the SampleWiseCorDP method using Lag-Llama as the base forecaster.

Results with IC-DP are in Table 18, and results with CorDP are in Table 19. We observe improvements similar to with DP, across several router models and small models. This shows that the difficulties predicted by the router model may not depend on the downstream strategy (DP, IC-DP, CorDP etc.) employed. Example routing curves with Qwen2.5-0.5B-Inst as the small model and the router are provided in Figure 56 and Figure 57 respectively.

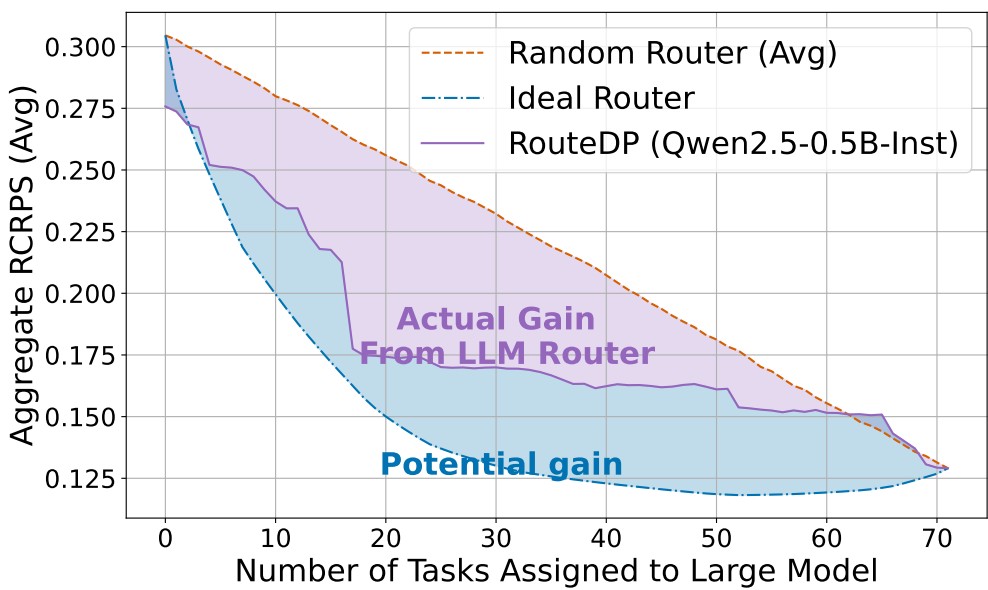

Figure 56: Route-IC-DP: Random, ideal and router curves with Qwen2.5-0.5B-Inst as the small model. The shaded region represents the distribution of 100 random assignment trajectories.

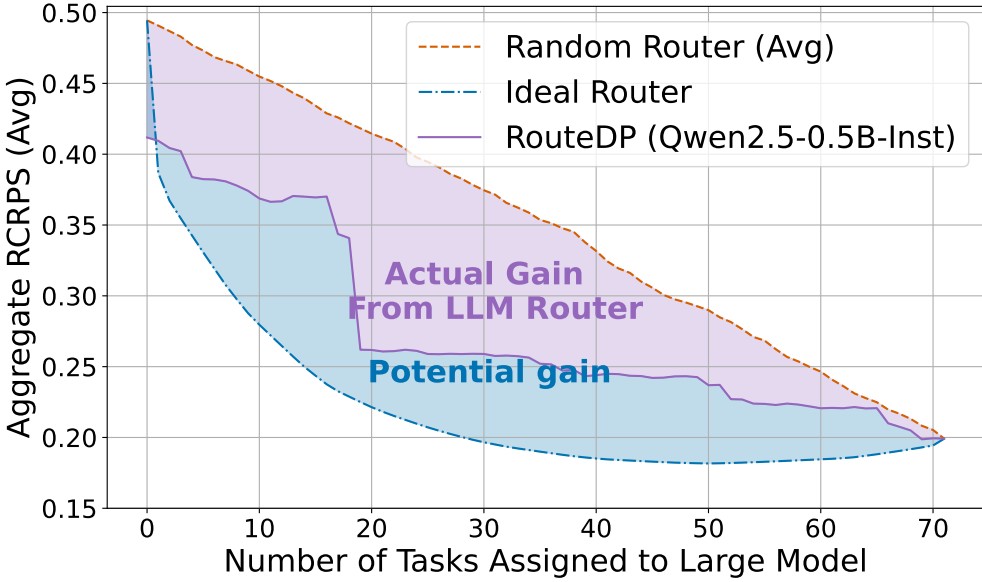

Figure 57: Route-CorDP: Random, ideal and router curves with Qwen2.5-0.5B-Inst as the small model. The shaded region represents the distribution of 100 random assignment trajectories.

| Small Model | Router | Percentage of tasks sent to large model | | | | | |
|---|---|---|---|---|---|---|---|
| | | 0% | 20% | 40% | 60% | 80% | 100% |
| Qwen2.5-0.5B-Inst | | | | | | | |
| | Qwen2.5-0.5B-Inst | 0.305 ± 0.006 | **0.225 ± 0.005** | **0.181 ± 0.004** | 0.160 ± 0.004 | 0.154 ± 0.004 | 0.129 ± 0.004 |
| | Qwen2.5-1.5B-Inst | 0.305 ± 0.006 | 0.250 ± 0.006 | 0.198 ± 0.005 | 0.171 ± 0.004 | 0.167 ± 0.004 | 0.129 ± 0.004 |
| | Qwen2.5-3B-Inst | 0.305 ± 0.006 | 0.241 ± 0.004 | 0.224 ± 0.004 | **0.157 ± 0.004** | **0.143 ± 0.004** | 0.129 ± 0.004 |
| | Qwen2.5-7B-Inst | 0.305 ± 0.006 | 0.265 ± 0.006 | 0.198 ± 0.005 | 0.185 ± 0.004 | **0.143 ± 0.004** | 0.129 ± 0.004 |
| | Qwen2.5-14B-Inst | 0.305 ± 0.006 | 0.297 ± 0.006 | 0.239 ± 0.006 | 0.222 ± 0.006 | 0.160 ± 0.004 | 0.129 ± 0.004 |
| | Qwen2.5-32B-Inst | 0.305 ± 0.006 | 0.244 ± 0.006 | 0.222 ± 0.005 | 0.191 ± 0.004 | 0.146 ± 0.004 | 0.129 ± 0.004 |
| | Qwen2.5-72B-Inst | 0.305 ± 0.006 | 0.274 ± 0.006 | 0.254 ± 0.006 | 0.236 ± 0.006 | 0.172 ± 0.006 | 0.129 ± 0.004 |
| Qwen2.5-1.5B-Inst | | | | | | | |
| | Qwen2.5-0.5B-Inst | 0.273 ± 0.008 | 0.216 ± 0.007 | 0.186 ± 0.005 | 0.171 ± 0.005 | 0.160 ± 0.004 | 0.129 ± 0.004 |
| | Qwen2.5-1.5B-Inst | 0.273 ± 0.008 | 0.243 ± 0.007 | 0.193 ± 0.005 | 0.165 ± 0.004 | 0.166 ± 0.004 | 0.129 ± 0.004 |
| | Qwen2.5-3B-Inst | 0.273 ± 0.008 | **0.197 ± 0.004** | **0.184 ± 0.005** | **0.153 ± 0.004** | 0.142 ± 0.004 | 0.129 ± 0.004 |
| | Qwen2.5-7B-Inst | 0.273 ± 0.008 | 0.237 ± 0.007 | 0.189 ± 0.005 | 0.178 ± 0.004 | **0.138 ± 0.004** | 0.129 ± 0.004 |
| | Qwen2.5-14B-Inst | 0.273 ± 0.008 | 0.260 ± 0.007 | 0.212 ± 0.007 | 0.197 ± 0.006 | 0.155 ± 0.003 | 0.129 ± 0.004 |
| | Qwen2.5-32B-Inst | 0.273 ± 0.008 | 0.216 ± 0.006 | 0.198 ± 0.005 | 0.171 ± 0.003 | 0.148 ± 0.003 | 0.129 ± 0.004 |
| | Qwen2.5-72B-Inst | 0.273 ± 0.008 | 0.244 ± 0.006 | 0.207 ± 0.006 | 0.196 ± 0.006 | 0.161 ± 0.005 | 0.129 ± 0.004 |
| Qwen2.5-3B-Inst | | | | | | | |
| | Qwen2.5-0.5B-Inst | 0.298 ± 0.011 | 0.247 ± 0.010 | 0.222 ± 0.009 | 0.220 ± 0.009 | 0.197 ± 0.009 | 0.129 ± 0.004 |
| | Qwen2.5-1.5B-Inst | 0.298 ± 0.011 | 0.255 ± 0.009 | 0.211 ± 0.009 | 0.151 ± 0.004 | 0.148 ± 0.004 | 0.129 ± 0.004 |
| | Qwen2.5-3B-Inst | 0.298 ± 0.011 | **0.223 ± 0.011** | 0.194 ± 0.007 | **0.148 ± 0.004** | **0.138 ± 0.004** | 0.129 ± 0.004 |
| | Qwen2.5-7B-Inst | 0.298 ± 0.011 | 0.250 ± 0.010 | **0.153 ± 0.004** | **0.148 ± 0.004** | 0.135 ± 0.004 | 0.129 ± 0.004 |
| | Qwen2.5-14B-Inst | 0.298 ± 0.011 | 0.274 ± 0.011 | 0.205 ± 0.010 | 0.194 ± 0.010 | 0.161 ± 0.009 | 0.129 ± 0.004 |
| | Qwen2.5-32B-Inst | 0.298 ± 0.011 | 0.250 ± 0.009 | 0.223 ± 0.009 | 0.213 ± 0.009 | 0.183 ± 0.009 | 0.129 ± 0.004 |
| | Qwen2.5-72B-Inst | 0.298 ± 0.011 | 0.265 ± 0.010 | 0.207 ± 0.010 | 0.201 ± 0.009 | 0.169 ± 0.009 | 0.129 ± 0.004 |
| Qwen2.5-7B-Inst | | | | | | | |
| | Qwen2.5-0.5B-Inst | 0.264 ± 0.012 | 0.209 ± 0.011 | 0.169 ± 0.003 | 0.164 ± 0.003 | 0.148 ± 0.003 | 0.129 ± 0.004 |
| | Qwen2.5-1.5B-Inst | 0.264 ± 0.012 | 0.218 ± 0.007 | 0.171 ± 0.005 | 0.147 ± 0.004 | 0.140 ± 0.004 | 0.129 ± 0.004 |
| | Qwen2.5-3B-Inst | 0.264 ± 0.012 | **0.197 ± 0.011** | 0.188 ± 0.011 | **0.143 ± 0.004** | **0.133 ± 0.004** | 0.129 ± 0.004 |
| | Qwen2.5-7B-Inst | 0.264 ± 0.012 | 0.215 ± 0.007 | **0.164 ± 0.005** | 0.151 ± 0.004 | 0.137 ± 0.004 | 0.129 ± 0.004 |
| | Qwen2.5-14B-Inst | 0.264 ± 0.012 | 0.246 ± 0.012 | 0.208 ± 0.012 | 0.196 ± 0.012 | 0.140 ± 0.003 | 0.129 ± 0.004 |
| | Qwen2.5-32B-Inst | 0.264 ± 0.012 | 0.231 ± 0.007 | 0.201 ± 0.006 | 0.168 ± 0.003 | 0.146 ± 0.002 | 0.129 ± 0.004 |
| | Qwen2.5-72B-Inst | 0.264 ± 0.012 | 0.242 ± 0.007 | 0.206 ± 0.007 | 0.196 ± 0.007 | 0.166 ± 0.006 | 0.129 ± 0.004 |
| Qwen2.5-14B-Inst | | | | | | | |
| | Qwen2.5-0.5B-Inst | 0.270 ± 0.005 | 0.258 ± 0.003 | **0.131 ± 0.003** | 0.132 ± 0.003 | 0.132 ± 0.003 | 0.129 ± 0.004 |
| | Qwen2.5-1.5B-Inst | 0.270 ± 0.005 | 0.142 ± 0.005 | 0.134 ± 0.005 | 0.134 ± 0.005 | 0.135 ± 0.005 | 0.129 ± 0.004 |
| | Qwen2.5-3B-Inst | 0.270 ± 0.005 | 0.258 ± 0.005 | 0.260 ± 0.005 | 0.132 ± 0.005 | 0.136 ± 0.005 | 0.129 ± 0.004 |
| | Qwen2.5-7B-Inst | 0.270 ± 0.005 | **0.138 ± 0.005** | 0.135 ± 0.005 | 0.133 ± 0.005 | 0.130 ± 0.004 | 0.129 ± 0.004 |
| | Qwen2.5-14B-Inst | 0.270 ± 0.005 | 0.267 ± 0.005 | 0.255 ± 0.003 | 0.254 ± 0.003 | 0.132 ± 0.003 | 0.129 ± 0.004 |
| | Qwen2.5-32B-Inst | 0.270 ± 0.005 | 0.153 ± 0.005 | 0.141 ± 0.005 | **0.129 ± 0.003** | **0.128 ± 0.003** | 0.129 ± 0.004 |
| | Qwen2.5-72B-Inst | 0.270 ± 0.005 | 0.148 ± 0.005 | 0.141 ± 0.005 | 0.133 ± 0.003 | 0.131 ± 0.003 | 0.129 ± 0.004 |
| Qwen2.5-32B-Inst | | | | | | | |
| | Qwen2.5-0.5B-Inst | 0.245 ± 0.027 | 0.189 ± 0.027 | **0.128 ± 0.002** | 0.129 ± 0.003 | 0.124 ± 0.002 | 0.129 ± 0.004 |
| | Qwen2.5-1.5B-Inst | 0.245 ± 0.027 | **0.179 ± 0.003** | 0.177 ± 0.003 | 0.183 ± 0.004 | 0.182 ± 0.004 | 0.129 ± 0.004 |
| | Qwen2.5-3B-Inst | 0.245 ± 0.027 | 0.241 ± 0.027 | 0.243 ± 0.027 | 0.177 ± 0.004 | 0.178 ± 0.004 | 0.129 ± 0.004 |
| | Qwen2.5-7B-Inst | 0.245 ± 0.027 | 0.181 ± 0.003 | 0.183 ± 0.004 | 0.182 ± 0.004 | 0.131 ± 0.004 | 0.129 ± 0.004 |
| | Qwen2.5-14B-Inst | 0.245 ± 0.027 | 0.241 ± 0.027 | 0.187 ± 0.027 | 0.184 ± 0.027 | 0.132 ± 0.002 | 0.129 ± 0.004 |
| | Qwen2.5-32B-Inst | 0.245 ± 0.027 | 0.193 ± 0.003 | 0.180 ± 0.003 | 0.130 ± 0.002 | **0.123 ± 0.002** | 0.129 ± 0.004 |
| | Qwen2.5-72B-Inst | 0.245 ± 0.027 | 0.193 ± 0.003 | 0.181 ± 0.003 | **0.126 ± 0.002** | **0.123 ± 0.002** | 0.129 ± 0.004 |

Table 18: Average RCRPS with Route-IC-DP (IC-DP combined with RouteDP) as the percentage of tasks sent to the large model increases. Grayed columns (0% and 100%) are baselines; **bold** indicates best router at each budget within each small-model block. Means ± standard errors. **Conclusions (hold for all small models):** (1) Routing improves over the small-model baseline (0%) across all small models and routers. (2) Each small model achieves best routed performance with a particular router at each budget; self-routing or same-family routers often perform well. (3) Combining IC-DP with routing preserves the benefits of both strategies.

| Small Model | Router | Percentage of tasks sent to large model | | | | | |
|---|---|---|---|---|---|---|---|
| | | 0% | 20% | 40% | 60% | 80% | 100% |
| Qwen2.5-0.5B-Inst | | | | | | | |
| | Qwen2.5-0.5B-Inst | 0.494 ± 0.008 | **0.332 ± 0.009** | **0.295 ± 0.007** | **0.269 ± 0.007** | 0.235 ± 0.006 | 0.199 ± 0.006 |
| | Qwen2.5-1.5B-Inst | 0.494 ± 0.008 | 0.467 ± 0.006 | 0.399 ± 0.006 | 0.364 ± 0.006 | 0.356 ± 0.006 | 0.199 ± 0.006 |
| | Qwen2.5-3B-Inst | 0.494 ± 0.008 | 0.408 ± 0.008 | 0.366 ± 0.008 | 0.325 ± 0.005 | 0.313 ± 0.005 | 0.199 ± 0.006 |
| | Qwen2.5-7B-Inst | 0.494 ± 0.008 | 0.445 ± 0.006 | 0.398 ± 0.006 | 0.387 ± 0.007 | **0.230 ± 0.006** | 0.199 ± 0.006 |
| | Qwen2.5-14B-Inst | 0.494 ± 0.008 | 0.477 ± 0.008 | 0.348 ± 0.009 | 0.323 ± 0.008 | 0.243 ± 0.007 | 0.199 ± 0.006 |
| | Qwen2.5-32B-Inst | 0.494 ± 0.008 | 0.420 ± 0.005 | 0.390 ± 0.005 | 0.270 ± 0.006 | 0.242 ± 0.006 | 0.199 ± 0.006 |
| | Qwen2.5-72B-Inst | 0.494 ± 0.008 | 0.454 ± 0.006 | 0.419 ± 0.006 | 0.306 ± 0.006 | 0.270 ± 0.006 | 0.199 ± 0.006 |
| Qwen2.5-1.5B-Inst | | | | | | | |
| | Qwen2.5-0.5B-Inst | 0.522 ± 0.018 | 0.475 ± 0.018 | 0.295 ± 0.007 | 0.267 ± 0.007 | 0.234 ± 0.006 | 0.199 ± 0.006 |
| | Qwen2.5-1.5B-Inst | 0.522 ± 0.018 | 0.355 ± 0.007 | 0.290 ± 0.006 | 0.257 ± 0.007 | 0.250 ± 0.007 | 0.199 ± 0.006 |
| | Qwen2.5-3B-Inst | 0.522 ± 0.018 | 0.439 ± 0.018 | 0.395 ± 0.017 | **0.215 ± 0.006** | **0.207 ± 0.005** | 0.199 ± 0.006 |
| | Qwen2.5-7B-Inst | 0.522 ± 0.018 | 0.449 ± 0.007 | 0.409 ± 0.007 | 0.281 ± 0.007 | 0.231 ± 0.006 | 0.199 ± 0.006 |
| | Qwen2.5-14B-Inst | 0.522 ± 0.018 | 0.507 ± 0.018 | 0.484 ± 0.018 | 0.465 ± 0.018 | 0.364 ± 0.007 | 0.199 ± 0.006 |
| | Qwen2.5-32B-Inst | 0.522 ± 0.018 | **0.308 ± 0.006** | **0.282 ± 0.006** | 0.268 ± 0.006 | 0.244 ± 0.007 | 0.199 ± 0.006 |
| | Qwen2.5-72B-Inst | 0.522 ± 0.018 | 0.459 ± 0.007 | 0.430 ± 0.007 | 0.304 ± 0.007 | 0.267 ± 0.006 | 0.199 ± 0.006 |
| Qwen2.5-3B-Inst | | | | | | | |
| | Qwen2.5-0.5B-Inst | 0.398 ± 0.028 | 0.374 ± 0.028 | 0.267 ± 0.006 | 0.251 ± 0.006 | 0.231 ± 0.006 | 0.199 ± 0.006 |
| | Qwen2.5-1.5B-Inst | 0.398 ± 0.028 | 0.300 ± 0.006 | 0.244 ± 0.006 | 0.220 ± 0.006 | 0.220 ± 0.006 | 0.199 ± 0.006 |
| | Qwen2.5-3B-Inst | 0.398 ± 0.028 | 0.322 ± 0.028 | 0.304 ± 0.028 | **0.201 ± 0.006** | **0.201 ± 0.006** | 0.199 ± 0.006 |
| | Qwen2.5-7B-Inst | 0.398 ± 0.028 | 0.274 ± 0.006 | 0.240 ± 0.006 | 0.237 ± 0.007 | 0.205 ± 0.006 | 0.199 ± 0.006 |
| | Qwen2.5-14B-Inst | 0.398 ± 0.028 | 0.380 ± 0.028 | 0.360 ± 0.029 | 0.349 ± 0.028 | 0.226 ± 0.006 | 0.199 ± 0.006 |
| | Qwen2.5-32B-Inst | 0.398 ± 0.028 | **0.260 ± 0.005** | **0.232 ± 0.004** | 0.227 ± 0.005 | 0.224 ± 0.006 | 0.199 ± 0.006 |
| | Qwen2.5-72B-Inst | 0.398 ± 0.028 | 0.286 ± 0.006 | 0.254 ± 0.006 | 0.252 ± 0.006 | 0.225 ± 0.005 | 0.199 ± 0.006 |
| Qwen2.5-7B-Inst | | | | | | | |
| | Qwen2.5-0.5B-Inst | 0.382 ± 0.007 | 0.374 ± 0.007 | 0.251 ± 0.008 | 0.235 ± 0.007 | 0.214 ± 0.006 | 0.199 ± 0.006 |
| | Qwen2.5-1.5B-Inst | 0.382 ± 0.007 | 0.263 ± 0.007 | 0.238 ± 0.006 | 0.234 ± 0.007 | 0.230 ± 0.007 | 0.199 ± 0.006 |
| | Qwen2.5-3B-Inst | 0.382 ± 0.007 | 0.351 ± 0.006 | 0.334 ± 0.006 | **0.210 ± 0.006** | **0.208 ± 0.006** | 0.199 ± 0.006 |
| | Qwen2.5-7B-Inst | 0.382 ± 0.007 | 0.362 ± 0.006 | 0.352 ± 0.006 | 0.236 ± 0.007 | 0.219 ± 0.007 | 0.199 ± 0.006 |
| | Qwen2.5-14B-Inst | 0.382 ± 0.007 | 0.374 ± 0.006 | 0.369 ± 0.006 | 0.361 ± 0.006 | 0.345 ± 0.007 | 0.199 ± 0.006 |
| | Qwen2.5-32B-Inst | 0.382 ± 0.007 | **0.245 ± 0.006** | **0.228 ± 0.006** | 0.229 ± 0.007 | 0.225 ± 0.007 | 0.199 ± 0.006 |
| | Qwen2.5-72B-Inst | 0.382 ± 0.007 | 0.380 ± 0.007 | 0.360 ± 0.006 | 0.236 ± 0.006 | 0.224 ± 0.006 | 0.199 ± 0.006 |
| Qwen2.5-14B-Inst | | | | | | | |
| | Qwen2.5-0.5B-Inst | 0.364 ± 0.006 | 0.358 ± 0.006 | 0.246 ± 0.007 | 0.243 ± 0.007 | 0.231 ± 0.007 | 0.199 ± 0.006 |
| | Qwen2.5-1.5B-Inst | 0.364 ± 0.006 | 0.239 ± 0.006 | 0.223 ± 0.006 | 0.206 ± 0.006 | 0.204 ± 0.006 | 0.199 ± 0.006 |
| | Qwen2.5-3B-Inst | 0.364 ± 0.006 | 0.334 ± 0.007 | 0.325 ± 0.006 | **0.196 ± 0.006** | 0.198 ± 0.006 | 0.199 ± 0.006 |
| | Qwen2.5-7B-Inst | 0.364 ± 0.006 | 0.348 ± 0.006 | 0.314 ± 0.005 | 0.203 ± 0.006 | 0.198 ± 0.006 | 0.199 ± 0.006 |
| | Qwen2.5-14B-Inst | 0.364 ± 0.006 | 0.351 ± 0.006 | 0.323 ± 0.006 | 0.318 ± 0.005 | 0.324 ± 0.007 | 0.199 ± 0.006 |
| | Qwen2.5-32B-Inst | 0.364 ± 0.006 | **0.227 ± 0.006** | **0.212 ± 0.006** | 0.222 ± 0.006 | 0.219 ± 0.007 | 0.199 ± 0.006 |
| | Qwen2.5-72B-Inst | 0.364 ± 0.006 | 0.351 ± 0.006 | 0.326 ± 0.006 | 0.204 ± 0.005 | **0.196 ± 0.005** | 0.199 ± 0.006 |
| Qwen2.5-32B-Inst | | | | | | | |
| | Qwen2.5-0.5B-Inst | 0.310 ± 0.005 | 0.311 ± 0.006 | 0.199 ± 0.006 | 0.204 ± 0.006 | 0.199 ± 0.006 | 0.199 ± 0.006 |
| | Qwen2.5-1.5B-Inst | 0.310 ± 0.005 | **0.190 ± 0.005** | 0.192 ± 0.005 | 0.200 ± 0.006 | 0.201 ± 0.006 | 0.199 ± 0.006 |
| | Qwen2.5-3B-Inst | 0.310 ± 0.005 | 0.309 ± 0.005 | 0.314 ± 0.005 | 0.195 ± 0.006 | 0.198 ± 0.006 | 0.199 ± 0.006 |
| | Qwen2.5-7B-Inst | 0.310 ± 0.005 | 0.298 ± 0.004 | 0.306 ± 0.005 | 0.194 ± 0.005 | 0.195 ± 0.006 | 0.199 ± 0.006 |
| | Qwen2.5-14B-Inst | 0.310 ± 0.005 | 0.305 ± 0.004 | 0.299 ± 0.004 | 0.294 ± 0.004 | 0.304 ± 0.005 | 0.199 ± 0.006 |
| | Qwen2.5-32B-Inst | 0.310 ± 0.005 | **0.190 ± 0.005** | **0.184 ± 0.005** | 0.188 ± 0.006 | 0.195 ± 0.006 | 0.199 ± 0.006 |
| | Qwen2.5-72B-Inst | 0.310 ± 0.005 | 0.307 ± 0.005 | 0.296 ± 0.004 | **0.173 ± 0.004** | **0.180 ± 0.004** | 0.199 ± 0.006 |

Table 19: Average RCRPS with Route-CorDP (SampleWise-CorDP, Lag-Llama as base forecaster) as the percentage of tasks sent to the large model increases. Grayed columns (0% and 100%) are baselines; **bold** indicates best router at each budget within each small-model block. Means ± standard errors. **Conclusions (hold for all small models):** (1) Routing improves over the small-model-only baseline (0%) across all small models and routers. (2) Each small model achieves best routed performance with a particular router at each budget; 32B and 72B routers often perform well for larger small models. (3) Combining CorDP with routing preserves the benefits of both strategies.

# G Implementation Details

To evaluate our models on the CiK benchmark, we use the official codebase of CiK at `https://github.com/ServiceNow/context-is-key-forecasting`. We use the same codebase to run model on the Direct Prompt (DP) method and the quantitative baselines benchmarked for CorDP. For completeness, we provide

the details here. Code for all proposed methods will be released on acceptance, with instructions to reproduce all experiments.

### G.1 LLMs

We self-host the respective official HuggingFace models: Llama3.2-1B-Inst (`https://huggingface.co/meta-Llama/Llama3.2-1B-Inst`), Llama3.2-3B-Inst (`https://huggingface.co/meta-Llama/Llama3.2-3B-Inst`), Llama3-8B-Inst (`https://huggingface.co/meta-Llama/Meta-Llama3-8B-Inst`), Qwen2.5-0.5B-Inst (`https://huggingface.co/Qwen2.5-0.5B-Inst`), Qwen2.5-1.5B-Inst (`https://huggingface.co/Qwen2.5-1.5B-Inst`), Qwen2.5-3B-Inst (`https://huggingface.co/Qwen2.5-3B-Inst`), Qwen2.5-7B-Inst (`https://huggingface.co/Qwen2.5-7B-Inst`), Qwen2.5-14B-Inst (`https://huggingface.co/Qwen2.5-14B-Inst`), Qwen2.5-32B-Inst (`https://huggingface.co/Qwen2.5-32B-Inst`). We use an appropriate number of H100 GPUs for each model. This ranged from 1 GPU (Models below 7B), 2 GPUs (7B, 14B Models) and 4 GPUs (32B Models).

Due to compute restrictions, for all our experiments involving Llama3.1-405B-Inst, Llama3.3-70B-Inst and Qwen2.5-72B-Inst, we use OpenRouter endpoints at `https://openrouter.ai/meta-Llama/Llama3.1-405b-Inst`, `https://openrouter.ai/meta-Llama/Llama3.3-70b-Inst` and `https://openrouter.ai/Qwen2.5-72b-Inst` respectively.

For all other models such as GPT-5.2, Claude-Sonnet-4.5, Gemini-2.5-Pro, we use the respective OpenRouter endpoints.

For all the above LLMs, we use the below prompt for the Direct Prompt method, as given in `https://github.com/ServiceNow/context-is-key-forecasting`. **context**, **history** and **pred_time** are replaced by the respective textual context, numerical history and timestamps for which a forecast is required.

```
I have a time series forecasting task for you.

Here is some context about the task. Make sure to factor in any background knowledge,
satisfy any constraints, and respect any scenarios.
<context>
((context))
</context>

Here is a historical time series in (timestamp, value) format:
<history>
((history))
</history>

Now please predict the value at the following timestamps: ((pred_time)).

Return the forecast in (timestamp, value) format in between <forecast> and </forecast> tags.
Do not include any other information (e.g., comments) in the forecast.

Example:
<history>
(t1, v1)
(t2, v2)
(t3, v3)
</history>
<forecast>
(t4, v4)
(t5, v5)
</forecast>
```

### G.2 Lag-Llama

We use the publicly available implementation of Lag-Llama (Rasul et al., 2023) following the instructions at `https://github.com/time-series-foundation-models/`, on a single H100 GPU.

### G.3 Chronos

We use the publicly available implementation of Chronos-Large (Ansari et al., 2024) following the instructions at at `https://github.com/amazon-science/chronos-forecasting` on a single H100 GPU.

### G.4 ARIMA

We used the implementation of ARIMA from the `forecast` R package, using `rpy2`. Results are computed using the `auto.arima` method. We reran the model with restricted parameter and disabled seasonality if the ARIMA fit failed.

## H  Cost and Inference Time of Methods

We report the inference time and cost of all experiments below. Note that cost only applies to models that were run using LLM APIs such as OpenRouter and OpenAI, and does not apply for models that were hosted locally.

## H.1 DP

| Metric | Total time taken | | Total Token Cost (USD) | | |
|---|---|---|---|---|---|
| | Average | Total | Average | Total | |
| Model | | | | | |
| With Context | | | | | |
| GPT-4o | 1m 26s | 28m 45s | 0.26 | 4.89 | |
| GPT-4o-mini | 2m 37s | 52m 22s | 0.03 | 0.50 | |
| GPT-5.2 | 12m 2s | 14h 14m | 0.54 | 38.53 | |
| Claude-Sonnet-4.5 | 4m 14s | 5h | 0.43 | 30.88 | |
| Gemini-2.5-Pro | 14m 28s | 17h 7m | 1.42 | 101.10 | |
| Llama3-8B-Inst | 2m 10s | 43m 25s | - | - | |
| Llama3.1-405B-Inst | 2m 57s | 59m 2s | - | - | |
| Llama3.2-1B-Inst | 1m 33s | 31m 2s | - | - | |
| Llama3.2-3B-Inst | 2m 5s | 41m 44s | - | - | |
| Llama3.3-70B-Inst | 4m 39s | 1h 33m | - | - | |
| Qwen2.5-72B-Inst | 19m 22s | 6h 27m | 0.02 | 0.45 | |
| Qwen2.5-0.5B-Inst | 14m 55s | 4h 58m | - | - | |
| Qwen2.5-1.5B-Inst | 17m 34s | 5h 51m | - | - | |
| Qwen2.5-14B-Inst | 15m 23s | 5h 7m | - | - | |
| Qwen2.5-32B-Inst | 18m 9s | 6h 3m | - | - | |
| Qwen2.5-3B-Inst | 18m 15s | 6h 5m | - | - | |
| Qwen2.5-7B-Inst | 21m 43s | 7h 14m | - | - | |
| Without Context | | | | | |
| GPT-4o | 1m 25s | 28m 26s | 0.23 | 4.67 | |
| GPT-4o-mini | 2m 35s | 51m 43s | 0.03 | 0.57 | |
| GPT-5.2 | 11m 37s | 13h 45m | 0.34 | 24.25 | |
| Claude-Sonnet-4.5 | 4m 23s | 5h 11m | 0.43 | 30.37 | |
| Gemini-2.5-Pro | 9m 16s | 10h 58m | 1.45 | 103.28 | |
| Llama3-8B-Inst | 2m 11s | 43m 54s | - | - | |
| Llama3.1-405B-Inst | 2m 55s | 58m 33s | - | - | |
| Llama3.2-1B-Inst | 1m 35s | 31m 59s | - | - | |
| Llama3.2-3B-Inst | 2m 8s | 42m 50s | - | - | |
| Llama3.3-70B-Inst | 4m 31s | 1h 30m | - | - | |
| Qwen2.5-72B-Inst | 16m 13s | 5h 24m | 0.02 | 0.43 | |
| Qwen2.5-0.5B-Inst | 15m 0s | 5h 0m | - | - | |
| Qwen2.5-1.5B-Inst | 17m 30s | 5h 50m | - | - | |
| Qwen2.5-3B-Inst | 21m 43s | 7h 14m | - | - | |
| Qwen2.5-7B-Inst | 10m 27s | 3h 29m | - | - | |
| Qwen2.5-14B-Inst | 15m 4s | 5h 1m | - | - | |
| Qwen2.5-32B-Inst | 16m 38s | 5h 32m | - | - | |

Table 20: Cost of performing inference with the **DP** method. "Total" values represent the time (or) cost of running the models on all tasks in the CiK benchmark (Williams et al., 2025), "Average" values represent the average time (or) cost of running the models on a single task from the benchmark.

## H.2 FxDP

To perform the comparison of each model's forecast effect explanation trace with the ground truth, we use three judge models (Claude-Sonnet-4.5, Gemini-2.5-Pro, and GPT-5.2), incurring a total cost of **USD 5.3411** across all judges. The cost per-model for each judge is provided in Table 21, Table 22, and Table 23, and the combined cost across all judges is in Table 24.

| Model | Total Cost (USD) |
|---|---|
| Llama3.3-70B-Inst | 0.2563 |
| Qwen2.5-72B-Inst | 0.2473 |
| Llama3.1-405B-Inst | 0.2471 |
| Qwen2.5-7B-Inst | 0.2469 |
| Qwen2.5-14B-Inst | 0.2370 |
| Qwen2.5-32B-Inst | 0.2280 |
| Llama3.2-3B-Inst | 0.2219 |
| Gemini-2.5-Pro | 0.2087 |
| Qwen2.5-3B-Inst | 0.2021 |
| GPT-5.2 | 0.2016 |
| Claude-Sonnet-4.5 | 0.1910 |

Table 21: Total cost per model for reasoning correctness evaluation (Claude-Sonnet-4.5 as judge). Total: USD 2.4881.

| Model | Total Cost (USD) |
|---|---|
| GPT-5.2 | 0.2331 |
| Claude-Sonnet-4.5 | 0.2308 |
| Qwen2.5-14B-Inst | 0.2304 |
| Qwen2.5-32B-Inst | 0.2273 |
| Gemini-2.5-Pro | 0.2261 |
| Qwen2.5-72B-Inst | 0.2240 |
| Llama3.3-70B-Inst | 0.2225 |
| Llama3.1-405B-Inst | 0.2216 |
| Qwen2.5-7B-Inst | 0.2186 |
| Llama3.2-3B-Inst | 0.2122 |
| Qwen2.5-3B-Inst | 0.1738 |

Table 22: Total cost per model for reasoning correctness evaluation (Gemini-2.5-Pro as judge). Total: USD 2.4204.

| Model | Total Cost (USD) |
|---|---|
| Llama3.2-3B-Inst | 0.0486 |
| Qwen2.5-72B-Inst | 0.0482 |
| Llama3.3-70B-Inst | 0.0466 |
| Qwen2.5-7B-Inst | 0.0459 |
| Qwen2.5-14B-Inst | 0.0442 |
| Llama3.1-405B-Inst | 0.0438 |
| Qwen2.5-32B-Inst | 0.0432 |
| Qwen2.5-3B-Inst | 0.0422 |
| Claude-Sonnet-4.5 | 0.0353 |
| GPT-5.2 | 0.0347 |

Table 23: Total cost per model for reasoning correctness evaluation (GPT-5.2 as judge). Total: USD 0.4326.

| Model | Total Cost (USD) |
|---|---|
| Llama3.3-70B-Inst | 0.5255 |
| Qwen2.5-72B-Inst | 0.5195 |
| Llama3.1-405B-Inst | 0.5125 |
| Qwen2.5-14B-Inst | 0.5115 |
| Qwen2.5-7B-Inst | 0.5114 |
| Qwen2.5-32B-Inst | 0.4986 |
| Llama3.2-3B-Inst | 0.4827 |
| GPT-5.2 | 0.4694 |
| Claude-Sonnet-4.5 | 0.4571 |
| Gemini-2.5-Pro | 0.4348 |
| Qwen2.5-3B-Inst | 0.4181 |

Table 24: Combined total cost per model across all three judges (Claude-Sonnet-4.5, Gemini-2.5-Pro, GPT-5.2). Total: USD 5.3411.

We report the time taken and cost incurred per model to produce forecasts using the FxDP method, in Table 25.

| Metric | Total time | | Total cost (USD) | |
|---|---|---|---|---|
| Stat | Avg | Total | Avg | Total |
| Model | | | | |
| Llama3.2-3B-Inst | 17m 59s | 20h 59m | - | - |
| Llama3.1-405B-Inst | 15m 55s | 18h 34m | 21.03 | 1471.86 |
| Llama3.1-8B-Inst | 12m 17s | 13h 43m | - | - |
| Llama3.3-70B-Inst | 7m 15s | 8h 28m | - | - |
| Qwen2.5-72B-Inst | 21m 3s | 24h 55m | - | - |
| Qwen2.5-7B-Inst | 4m 33s | 5h 23m | - | - |
| Qwen2.5-14B-Inst | 9m 16s | 10h 59m | - | - |
| Qwen2.5-32B-Inst | 13m 6s | 15h 31m | - | - |
| Qwen2.5-3B-Inst | 41m 11s | 45h 59m | - | - |
| Claude-Sonnet-4.5 | 5m 1s | 1h 40m | 0.69 | 13.84 |
| Gemini-2.5-Pro | 8m 44s | 2h 54m | 0.83 | 16.57 |
| GPT-5.2 | 7m 25s | 2h 28m | 0.51 | 10.21 |

Table 25: Cost per model for producing a forecast using the FxDP method.

## H.3   IC-DP

We report the time taken and cost incurred per model to produce forecasts using the ICDP method, in Table 26.

| Metric | Total time taken | | Total Token Cost (USD) | | |
|---|---|---|---|---|---|
| Model | Average | Total | Average | Total | |
| GPT-4o | 0m 55s | 1h 5m | 2.01 | 142.83 | |
| GPT-4o-mini | 1m 19s | 1h 34m | 0.01 | 0.95 | |
| GPT-5.2 | 11m 24s | 13h 29m | 0.60 | 42.25 | |
| Claude-Sonnet-4.5 | 4m 0s | 4h 44m | 0.64 | 45.17 | |
| Gemini-2.5-Pro | 16m 29s | 19h 30m | 1.38 | 98.27 | |
| Llama3.1-405B-Inst | 8m 34s | 10h 9m | 0.13 | 9.57 | |
| Llama3.3-70B-Inst | 6m 10s | 7h 18m | 0.02 | 1.15 | |
| Llama3-8B-Inst | 0m 45s | 53m 26s | - | - | |
| Llama3.2-1B-Inst | 3m 27s | 4h 5m | - | - | |
| Llama3.2-3B-Inst | 3m 43s | 4h 24m | - | - | |
| Qwen2.5-72B-Inst | 14m 8s | 16h 43m | 0.03 | 1.95 | |
| Qwen2.5-0.5B-Inst | 6m 20s | 7h 30m | - | - | |
| Qwen2.5-1.5B-Inst | 7m 0s | 8h 17m | - | - | |
| Qwen2.5-14B-Inst | 13m 59s | 16h 33m | - | - | |
| Qwen2.5-32B-Inst | 17m 1s | 20h 8m | - | - | |
| Qwen2.5-3B-Inst | 7m 59s | 9h 27m | - | - | |
| Qwen2.5-7B-Inst | 8m 40s | 10h 15m | - | - | |

Table 26: Cost of performing inference with the **IC-DP** method. "Total" values represent the time (or) cost of running the models on all tasks in the CiK benchmark (Williams et al., 2025), "Average" values represent the average time (or) cost of running the models on a single task from the benchmark. Compared to DP (with context), IC-DP is only 1.1–1.5× more expensive in USD for the main API models (GPT-5.2, Claude-Sonnet-4.5, Gemini-2.5-Pro); total runtime is similar (within 1.0–1.15×)

## H.4 CorDP

We report the time taken and cost incurred per model to produce forecasts using the CorDP method, in Table 27 and Table 28. CorDP methods incur similar cost and time as DP.

| Metric | Total time taken | | Total Token Cost (USD) | | |
|---|---|---|---|---|---|
| Model | Average | Total | Average | Total | |
| Base Forecaster: Arima | | | | | |
| Llama3.1-405B-Inst | 8m 31s | 10h 4m | 0.03 | 2.39 | |
| Llama3.3-70B-Inst | 4m 51s | 5h 44m | - | - | |
| Llama3-8B-Inst | 2m 0s | 2h 22m | - | - | |
| Llama3.2-1B-Inst | 0m 59s | 1h 10m | - | - | |
| Llama3.2-3B-Inst | 1m 29s | 1h 46m | - | - | |
| Qwen2.5-72B-Inst | 14m 24s | 17h 2m | - | - | |
| Qwen2.5-0.5B-Inst | 1m 34s | 1h 51m | - | - | |
| Qwen2.5-1.5B-Inst | 1m 59s | 2h 21m | - | - | |
| Qwen2.5-14B-Inst | 6m 15s | 7h 24m | - | - | |
| Qwen2.5-32B-Inst | 8m 33s | 10h 7m | - | - | |
| Qwen2.5-3B-Inst | 3m 15s | 3h 51m | - | - | |
| Qwen2.5-7B-Inst | 2m 51s | 3h 19m | - | - | |
| GPT-4o | 0m 50s | 59m 39s | 0.07 | 4.96 | |
| GPT-4o-mini | 1m 15s | 1h 29m | 0.00 | 0.32 | |
| GPT-5.2 | 10m 49s | 12h 48m | 0.63 | 45.07 | |
| Claude-Sonnet-4.5 | 5m 30s | 6h 31m | 0.52 | 36.59 | |
| Gemini-2.5-Pro | 6m 19s | 7h 28m | 0.50 | 35.77 | |
| Base Forecaster: Chronos-Large | | | | | |
| Llama3.1-405B-Inst | 8m 21s | 9h 53m | 0.03 | 2.47 | |
| Llama3.3-70B-Inst | 4m 52s | 5h 46m | - | - | |
| Llama3-8B-Inst | 1m 57s | 2h 19m | - | - | |
| Llama3.2-1B-Inst | 1m 1s | 1h 10m | - | - | |
| Llama3.2-3B-Inst | 1m 28s | 1h 44m | - | - | |
| Qwen2.5-72B-Inst | 17m 55s | 21h 12m | - | - | |
| Qwen2.5-0.5B-Inst | 3m 42s | 3h 46m | - | - | |
| Qwen2.5-1.5B-Inst | 3m 31s | 4h 9m | - | - | |
| Qwen2.5-14B-Inst | 8m 28s | 10h 2m | - | - | |
| Qwen2.5-32B-Inst | 10m 57s | 12h 58m | - | - | |
| Qwen2.5-3B-Inst | 5m 2s | 5h 58m | - | - | |
| Qwen2.5-7B-Inst | 2m 54s | 3h 23m | - | - | |
| GPT-4o | 0m 50s | 1h 0m | 0.07 | 5.06 | |
| GPT-4o-mini | 1m 18s | 1h 32m | 0.00 | 0.31 | |
| GPT-5.2 | 9m 8s | 10h 48m | 0.64 | 45.25 | |
| Claude-Sonnet-4.5 | 5m 25s | 6h 24m | 0.51 | 36.29 | |
| Gemini-2.5-Pro | 6m 18s | 7h 27m | 0.50 | 35.18 | |
| Base Forecaster: Lag-Llama | | | | | |
| Llama3.1-405B-Inst | 8m 16s | 9h 47m | 0.03 | 2.43 | |
| Llama3.3-70B-Inst | 4m 44s | 5h 36m | - | - | |
| Llama3-8B-Inst | 1m 56s | 2h 17m | - | - | |
| Llama3.2-1B-Inst | 0m 57s | 1h 7m | - | - | |
| Llama3.2-3B-Inst | 1m 27s | 1h 43m | - | - | |
| Qwen2.5-72B-Inst | 17m 37s | 20h 51m | - | - | |
| Qwen2.5-0.5B-Inst | 2m 57s | 3h 29m | - | - | |
| Qwen2.5-1.5B-Inst | 3m 23s | 4h 0m | - | - | |
| Qwen2.5-14B-Inst | 9m 29s | 11h 14m | - | - | |
| Qwen2.5-32B-Inst | 13m 17s | 15h 43m | - | - | |
| Qwen2.5-3B-Inst | 6m 0s | 7h 6m | - | - | |
| Qwen2.5-7B-Inst | 2m 47s | 3h 15m | - | - | |
| GPT-4o | 0m 52s | 1h 2m | 0.07 | 4.86 | |
| GPT-4o-mini | 1m 32s | 1h 49m | 0.00 | 0.33 | |
| GPT-5.2 | 8m 49s | 10h 27m | 1.71 | 121.68 | |
| Claude-Sonnet-4.5 | 5m 23s | 6h 22m | 0.50 | 35.84 | |
| Gemini-2.5-Pro | 12m 35s | 14h 54m | 2.16 | 153.11 | |

Table 27: Cost of performing inference with the **Median-CorDP** method. "Total" values represent the time (or) cost of running the models on all tasks in the CiK benchmark (Williams et al., 2025), "Average" values represent the average time (or) cost of running the models on a single task from the benchmark.

| Metric | Total time taken | | Total Token Cost (USD) | | |
|---|---|---|---|---|---|
| Stat | Avg | Total | Avg | Total | |
| Model | | | | | |
| **Base Forecaster: Arima** | | | | | |
| Llama3.1-405B-Inst | 8m 32s | 10h 6m | 0.03 | 2.38 | |
| Llama3.3-70B-Inst | 42m 39s | 50h 29m | - | - | |
| Llama3-8B-Inst | 14m 22s | 16h 46m | - | - | |
| Llama3.2-1B-Inst | 5m 25s | 6h 20m | - | - | |
| Llama3.2-3B-Inst | 8m 37s | 10h 3m | - | - | |
| Qwen2.5-72B-Inst | 1h 6m | 79h 6m | - | - | |
| Qwen2.5-0.5B-Inst | 9m 43s | 11h 30m | - | - | |
| Qwen2.5-1.5B-Inst | 11m 28s | 13h 23m | - | - | |
| Qwen2.5-14B-Inst | 31m 37s | 36h 53m | - | - | |
| Qwen2.5-32B-Inst | 45m 5s | 52h 36m | - | - | |
| Qwen2.5-3B-Inst | 21m 3s | 24h 55m | - | - | |
| Qwen2.5-7B-Inst | 13m 34s | 15h 50m | - | - | |
| GPT-4o | 2m 34s | 3h 2m | 0.14 | 9.62 | |
| GPT-4o-mini | 4m 20s | 5h 8m | 0.01 | 0.59 | |
| GPT-5.2 | 5m 50s | 6h 54m | 0.41 | 28.99 | |
| Claude-Sonnet-4.5 | 5m 32s | 6h 33m | 0.51 | 36.50 | |
| Gemini-2.5-Pro | 6m 31s | 7h 42m | 0.53 | 37.66 | |
| **Base Forecaster: Chronos-Large** | | | | | |
| Llama3.1-405B-Inst | 8m 29s | 10h 3m | 0.03 | 2.34 | |
| Llama3.3-70B-Inst | 42m 38s | 50h 27m | - | - | |
| Llama3-8B-Inst | 14m 45s | 17h 28m | - | - | |
| Llama3.2-1B-Inst | 5m 39s | 6h 19m | - | - | |
| Llama3.2-3B-Inst | 8m 45s | 10h 22m | - | - | |
| Qwen2.5-72B-Inst | 1h 8m | 80h 29m | - | - | |
| Qwen2.5-0.5B-Inst | 1m 23s | 1h 38m | - | - | |
| Qwen2.5-1.5B-Inst | 12m 5s | 14h 18m | - | - | |
| Qwen2.5-14B-Inst | 32m 6s | 37h 27m | - | - | |
| Qwen2.5-32B-Inst | 45m 42s | 53h 19m | - | - | |
| Qwen2.5-3B-Inst | 21m 21s | 25h 16m | - | - | |
| Qwen2.5-7B-Inst | 13m 48s | 16h 20m | - | - | |
| GPT-4o | 2m 30s | 2h 58m | 0.14 | 9.61 | |
| GPT-4o-mini | 4m 20s | 5h 7m | 0.01 | 0.60 | |
| GPT-5.2 | 5m 11s | 6h 9m | 0.39 | 27.49 | |
| Claude-Sonnet-4.5 | 5m 33s | 6h 34m | 0.52 | 37.03 | |
| Gemini-2.5-Pro | 6m 22s | 7h 32m | 0.53 | 37.96 | |
| **Base Forecaster: Lag-Llama** | | | | | |
| Llama3.1-405B-Inst | 8m 28s | 10h 1m | 0.03 | 2.30 | |
| Llama3.3-70B-Inst | 41m 48s | 49h 27m | - | - | |
| Llama3-8B-Inst | 14m 14s | 16h 37m | - | - | |
| Llama3.2-1B-Inst | 5m 5s | 5h 51m | - | - | |
| Llama3.2-3B-Inst | 8m 31s | 9h 57m | - | - | |
| Qwen2.5-72B-Inst | 1h 5m | 77h 24m | - | - | |
| Qwen2.5-0.5B-Inst | 9m 8s | 10h 49m | - | - | |
| Qwen2.5-1.5B-Inst | 11m 15s | 13h 8m | - | - | |
| Qwen2.5-14B-Inst | 30m 56s | 36h 6m | - | - | |
| Qwen2.5-32B-Inst | 43m 59s | 51h 19m | - | - | |
| Qwen2.5-3B-Inst | 20m 28s | 24h 13m | - | - | |
| Qwen2.5-7B-Inst | 13m 4s | 15h 15m | - | - | |
| GPT-4o | 2m 30s | 2h 58m | 0.13 | 9.39 | |
| GPT-4o-mini | 4m 10s | 4h 55m | 0.01 | 0.59 | |
| GPT-5.2 | 5m 56s | 7h 1m | 0.41 | 28.84 | |
| Claude-Sonnet-4.5 | 5m 25s | 6h 25m | 0.51 | 35.88 | |
| Gemini-2.5-Pro | 6m 9s | 7h 17m | 0.52 | 36.93 | |

Table 28: Cost of performing inference with the **SampleWise-CorDP** method. "Total" values represent the time (or) cost of running the models on all tasks in the CiK benchmark (Williams et al., 2025), "Average" values represent the average time (or) cost of running the models on a single task from the benchmark.

### H.5 RouteDP

We report the time taken and cost incurred per model to produce the difficulty score of tasks using the RouteDP method, in Table 29.

| Model | Average (s) | Total (s) |
|---|---|---|
| Qwen2.5-0.5B-Inst | 0.18 | 12.48 |
| Qwen2.5-1.5B-Inst | 0.17 | 12.10 |
| Qwen2.5-3B-Inst | 0.19 | 13.58 |
| Qwen2.5-7B-Inst | 0.23 | 16.31 |
| Qwen2.5-14B-Inst | 0.28 | 19.90 |
| Qwen2.5-32B-Inst | 0.48 | 34.35 |

Table 29: Time taken to compute the difficulty scores of tasks with the **RouteDP** method. "Total" values represent the time to compute the score for all tasks in the CiK benchmark (Williams et al., 2025), "Average" values represent the average time taken to compute the score for a single task from the benchmark.

| Small Model | Router | Percentage of tasks sent to large model | | | | | |
|---|---|---|---|---|---|---|---|
| | | 0% | 20% | 40% | 60% | 80% | 100% |
| Qwen2.5-0.5B-Inst | | | | | | | |
| | Qwen2.5-0.5B-Inst | $0.592 \pm 0.027$ | $\mathbf{0.316 \pm 0.027}$ | $\mathbf{0.222 \pm 0.005}$ | $\mathbf{0.206 \pm 0.005}$ | $0.199 \pm 0.004$ | $0.173 \pm 0.003$ |
| | Qwen2.5-1.5B-Inst | $0.592 \pm 0.027$ | $0.504 \pm 0.009$ | $0.449 \pm 0.007$ | $0.404 \pm 0.004$ | $0.407 \pm 0.004$ | $0.173 \pm 0.003$ |
| | Qwen2.5-3B-Inst | $0.592 \pm 0.027$ | $0.507 \pm 0.026$ | $0.490 \pm 0.026$ | $0.393 \pm 0.003$ | $0.282 \pm 0.003$ | $0.173 \pm 0.003$ |
| | Qwen2.5-7B-Inst | $0.592 \pm 0.027$ | $0.510 \pm 0.010$ | $0.437 \pm 0.007$ | $0.412 \pm 0.004$ | $\mathbf{0.181 \pm 0.004}$ | $0.173 \pm 0.003$ |
| | Qwen2.5-14B-Inst | $0.592 \pm 0.027$ | $0.581 \pm 0.027$ | $0.439 \pm 0.027$ | $0.324 \pm 0.027$ | $0.187 \pm 0.004$ | $0.173 \pm 0.003$ |
| | Qwen2.5-32B-Inst | $0.592 \pm 0.027$ | $0.383 \pm 0.010$ | $0.368 \pm 0.008$ | $0.230 \pm 0.006$ | $0.196 \pm 0.004$ | $0.173 \pm 0.003$ |
| | Qwen2.5-72B-Inst | $0.592 \pm 0.027$ | $0.509 \pm 0.010$ | $0.395 \pm 0.009$ | $0.287 \pm 0.009$ | $0.243 \pm 0.009$ | $0.173 \pm 0.003$ |
| Qwen2.5-1.5B-Inst | | | | | | | |
| | Qwen2.5-0.5B-Inst | $0.616 \pm 0.018$ | $0.436 \pm 0.016$ | $\mathbf{0.258 \pm 0.005}$ | $0.231 \pm 0.005$ | $0.210 \pm 0.004$ | $0.173 \pm 0.003$ |
| | Qwen2.5-1.5B-Inst | $0.616 \pm 0.018$ | $0.466 \pm 0.016$ | $0.300 \pm 0.016$ | $0.212 \pm 0.005$ | $0.210 \pm 0.005$ | $0.173 \pm 0.003$ |
| | Qwen2.5-3B-Inst | $0.616 \pm 0.018$ | $\mathbf{0.375 \pm 0.009}$ | $0.349 \pm 0.009$ | $\mathbf{0.196 \pm 0.004}$ | $\mathbf{0.181 \pm 0.004}$ | $0.173 \pm 0.003$ |
| | Qwen2.5-7B-Inst | $0.616 \pm 0.018$ | $0.481 \pm 0.018$ | $0.288 \pm 0.016$ | $0.235 \pm 0.005$ | $0.188 \pm 0.004$ | $0.173 \pm 0.003$ |
| | Qwen2.5-14B-Inst | $0.616 \pm 0.018$ | $0.598 \pm 0.017$ | $0.536 \pm 0.016$ | $0.523 \pm 0.015$ | $0.214 \pm 0.004$ | $0.173 \pm 0.003$ |
| | Qwen2.5-32B-Inst | $0.616 \pm 0.018$ | $0.441 \pm 0.017$ | $0.356 \pm 0.017$ | $0.256 \pm 0.009$ | $0.210 \pm 0.004$ | $0.173 \pm 0.003$ |
| | Qwen2.5-72B-Inst | $0.616 \pm 0.018$ | $0.484 \pm 0.017$ | $0.448 \pm 0.017$ | $0.445 \pm 0.017$ | $0.375 \pm 0.015$ | $0.173 \pm 0.003$ |
| Qwen2.5-3B-Inst | | | | | | | |
| | Qwen2.5-0.5B-Inst | $0.424 \pm 0.017$ | $0.338 \pm 0.014$ | $0.301 \pm 0.011$ | $0.281 \pm 0.011$ | $0.208 \pm 0.004$ | $0.173 \pm 0.003$ |
| | Qwen2.5-1.5B-Inst | $0.424 \pm 0.017$ | $0.383 \pm 0.014$ | $0.309 \pm 0.012$ | $0.260 \pm 0.012$ | $0.262 \pm 0.012$ | $0.173 \pm 0.003$ |
| | Qwen2.5-3B-Inst | $0.424 \pm 0.017$ | $\mathbf{0.315 \pm 0.015}$ | $\mathbf{0.254 \pm 0.011}$ | $\mathbf{0.203 \pm 0.006}$ | $\mathbf{0.188 \pm 0.005}$ | $0.173 \pm 0.003$ |
| | Qwen2.5-7B-Inst | $0.424 \pm 0.017$ | $0.382 \pm 0.015$ | $0.285 \pm 0.012$ | $0.276 \pm 0.012$ | $0.228 \pm 0.010$ | $0.173 \pm 0.003$ |
| | Qwen2.5-14B-Inst | $0.424 \pm 0.017$ | $0.402 \pm 0.017$ | $0.340 \pm 0.015$ | $0.329 \pm 0.015$ | $0.246 \pm 0.011$ | $0.173 \pm 0.003$ |
| | Qwen2.5-32B-Inst | $0.424 \pm 0.017$ | $0.359 \pm 0.014$ | $0.326 \pm 0.013$ | $0.295 \pm 0.012$ | $0.262 \pm 0.011$ | $0.173 \pm 0.003$ |
| | Qwen2.5-72B-Inst | $0.424 \pm 0.017$ | $0.392 \pm 0.015$ | $0.345 \pm 0.015$ | $0.338 \pm 0.014$ | $0.289 \pm 0.013$ | $0.173 \pm 0.003$ |
| Qwen2.5-7B-Inst | | | | | | | |
| | Qwen2.5-0.5B-Inst | $0.401 \pm 0.006$ | $0.364 \pm 0.005$ | $0.238 \pm 0.004$ | $0.229 \pm 0.004$ | $0.208 \pm 0.004$ | $0.173 \pm 0.003$ |
| | Qwen2.5-1.5B-Inst | $0.401 \pm 0.006$ | $0.263 \pm 0.006$ | $0.222 \pm 0.005$ | $0.183 \pm 0.004$ | $0.181 \pm 0.004$ | $0.173 \pm 0.003$ |
| | Qwen2.5-3B-Inst | $0.401 \pm 0.006$ | $0.338 \pm 0.004$ | $0.314 \pm 0.004$ | $\mathbf{0.179 \pm 0.004}$ | $\mathbf{0.174 \pm 0.004}$ | $0.173 \pm 0.003$ |
| | Qwen2.5-7B-Inst | $0.401 \pm 0.006$ | $\mathbf{0.260 \pm 0.006}$ | $\mathbf{0.199 \pm 0.005}$ | $0.191 \pm 0.004$ | $0.188 \pm 0.004$ | $0.173 \pm 0.003$ |
| | Qwen2.5-14B-Inst | $0.401 \pm 0.006$ | $0.384 \pm 0.006$ | $0.351 \pm 0.006$ | $0.343 \pm 0.005$ | $0.194 \pm 0.004$ | $0.173 \pm 0.003$ |
| | Qwen2.5-32B-Inst | $0.401 \pm 0.006$ | $\mathbf{0.260 \pm 0.006}$ | $0.240 \pm 0.005$ | $0.231 \pm 0.004$ | $0.206 \pm 0.004$ | $0.173 \pm 0.003$ |
| | Qwen2.5-72B-Inst | $0.401 \pm 0.006$ | $0.267 \pm 0.006$ | $0.246 \pm 0.006$ | $0.244 \pm 0.006$ | $0.214 \pm 0.005$ | $0.173 \pm 0.003$ |
| Qwen2.5-14B-Inst | | | | | | | |
| | Qwen2.5-0.5B-Inst | $0.247 \pm 0.006$ | $0.208 \pm 0.004$ | $0.202 \pm 0.004$ | $0.199 \pm 0.004$ | $0.194 \pm 0.004$ | $0.173 \pm 0.003$ |
| | Qwen2.5-1.5B-Inst | $0.247 \pm 0.006$ | $0.246 \pm 0.006$ | $0.220 \pm 0.006$ | $0.196 \pm 0.006$ | $0.199 \pm 0.006$ | $0.173 \pm 0.003$ |
| | Qwen2.5-3B-Inst | $0.247 \pm 0.006$ | $\mathbf{0.206 \pm 0.006}$ | $0.204 \pm 0.006$ | $\mathbf{0.192 \pm 0.006}$ | $0.189 \pm 0.004$ | $0.173 \pm 0.003$ |
| | Qwen2.5-7B-Inst | $0.247 \pm 0.006$ | $0.234 \pm 0.007$ | $\mathbf{0.197 \pm 0.006}$ | $0.198 \pm 0.006$ | $0.179 \pm 0.003$ | $0.173 \pm 0.003$ |
| | Qwen2.5-14B-Inst | $0.247 \pm 0.006$ | $0.237 \pm 0.006$ | $0.203 \pm 0.006$ | $0.200 \pm 0.004$ | $\mathbf{0.176 \pm 0.003}$ | $0.173 \pm 0.003$ |
| | Qwen2.5-32B-Inst | $0.247 \pm 0.006$ | $0.230 \pm 0.005$ | $0.220 \pm 0.005$ | $0.195 \pm 0.003$ | $0.193 \pm 0.003$ | $0.173 \pm 0.003$ |
| | Qwen2.5-72B-Inst | $0.247 \pm 0.006$ | $0.238 \pm 0.006$ | $0.218 \pm 0.005$ | $0.203 \pm 0.004$ | $0.202 \pm 0.004$ | $0.173 \pm 0.003$ |
| Qwen2.5-32B-Inst | | | | | | | |
| | Qwen2.5-0.5B-Inst | $0.397 \pm 0.008$ | $0.296 \pm 0.005$ | $\mathbf{0.171 \pm 0.003}$ | $\mathbf{0.172 \pm 0.004}$ | $\mathbf{0.172 \pm 0.003}$ | $0.173 \pm 0.003$ |
| | Qwen2.5-1.5B-Inst | $0.397 \pm 0.008$ | $0.278 \pm 0.008$ | $0.272 \pm 0.008$ | $0.264 \pm 0.007$ | $0.266 \pm 0.007$ | $0.173 \pm 0.003$ |
| | Qwen2.5-3B-Inst | $0.397 \pm 0.008$ | $0.390 \pm 0.007$ | $0.384 \pm 0.007$ | $0.265 \pm 0.007$ | $0.218 \pm 0.006$ | $0.173 \pm 0.003$ |
| | Qwen2.5-7B-Inst | $0.397 \pm 0.008$ | $0.276 \pm 0.008$ | $0.273 \pm 0.008$ | $0.265 \pm 0.007$ | $0.175 \pm 0.003$ | $0.173 \pm 0.003$ |
| | Qwen2.5-14B-Inst | $0.397 \pm 0.008$ | $0.397 \pm 0.008$ | $0.361 \pm 0.007$ | $0.310 \pm 0.005$ | $0.177 \pm 0.003$ | $0.173 \pm 0.003$ |
| | Qwen2.5-32B-Inst | $0.397 \pm 0.008$ | $\mathbf{0.240 \pm 0.007}$ | $0.237 \pm 0.007$ | $0.185 \pm 0.003$ | $0.181 \pm 0.004$ | $0.173 \pm 0.003$ |
| | Qwen2.5-72B-Inst | $0.397 \pm 0.008$ | $0.284 \pm 0.008$ | $0.236 \pm 0.007$ | $0.191 \pm 0.005$ | $0.193 \pm 0.006$ | $0.173 \pm 0.003$ |
| Qwen2.5-72B-Inst | | | | | | | |
| | Qwen2.5-0.5B-Inst | $0.202 \pm 0.009$ | $\mathbf{0.167 \pm 0.007}$ | $\mathbf{0.156 \pm 0.004}$ | $\mathbf{0.158 \pm 0.004}$ | $\mathbf{0.165 \pm 0.004}$ | $0.173 \pm 0.003$ |
| | Qwen2.5-1.5B-Inst | $0.202 \pm 0.009$ | $0.184 \pm 0.006$ | $0.181 \pm 0.006$ | $0.185 \pm 0.006$ | $0.194 \pm 0.006$ | $0.173 \pm 0.003$ |
| | Qwen2.5-3B-Inst | $0.202 \pm 0.009$ | $0.207 \pm 0.009$ | $0.210 \pm 0.009$ | $0.189 \pm 0.006$ | $0.178 \pm 0.004$ | $0.173 \pm 0.003$ |
| | Qwen2.5-7B-Inst | $0.202 \pm 0.009$ | $0.187 \pm 0.006$ | $0.185 \pm 0.006$ | $0.192 \pm 0.006$ | $0.175 \pm 0.003$ | $0.173 \pm 0.003$ |
| | Qwen2.5-14B-Inst | $0.202 \pm 0.009$ | $0.207 \pm 0.009$ | $0.202 \pm 0.009$ | $0.190 \pm 0.008$ | $0.175 \pm 0.003$ | $0.173 \pm 0.003$ |
| | Qwen2.5-32B-Inst | $0.202 \pm 0.009$ | $0.183 \pm 0.005$ | $0.186 \pm 0.005$ | $0.180 \pm 0.004$ | $0.175 \pm 0.004$ | $0.173 \pm 0.003$ |
| | Qwen2.5-72B-Inst | $0.202 \pm 0.009$ | $0.198 \pm 0.006$ | $0.188 \pm 0.005$ | $0.185 \pm 0.004$ | $0.180 \pm 0.004$ | $0.173 \pm 0.003$ |

Table 16: Average RCRPS of small models routed with different routers, as the percentage of tasks sent to the large model (Llama-3.1-405B-Inst) increases. Grayed columns (0% and 100%) are baselines identical across routers for each small model; **bold** indicates the best router at each routing budget within each small-model block. Means ± standard errors. **Conclusions (hold for all small models):** (1) Each small model achieves its best routed performance when using itself as the router (or a same-size router) at most budgets. (2) Routing a fraction of tasks to the large model consistently improves over the small-model-only baseline (0%), with gains increasing as more tasks are routed. (3) Smallest small models (e.g. 0.5B–3B) show the largest absolute gains from routing; larger small models (e.g. 14B–72B) have less headroom but still benefit from routing at low-to-moderate budgets.

| Router | Qwen2.5-0.5B-Inst | Qwen2.5 1.5B | Qwen2.5 3B | Qwen2.5 7B | Qwen2.5 14B | Qwen2.5 32B | Qwen2.5-72B-Inst |
|---|---|---|---|---|---|---|---|
| Qwen2.5-0.5B-Inst | 66.76 | 48.83 | 13.50 | 29.05 | 22.03 | 68.59 | 67.59 |
| Qwen2.5-1.5B-Inst | 1.40 | 40.53 | 4.67 | 55.13 | 2.63 | 19.65 | 12.61 |
| Qwen2.5-3B-Inst | 3.10 | 46.41 | 45.41 | 23.05 | 22.23 | 3.47 | 1.23 |
| Qwen2.5-7B-Inst | 7.95 | 35.33 | 7.41 | 55.72 | 15.73 | 26.63 | 6.71 |
| Qwen2.5-14B-Inst | 10.12 | 4.62 | 0.00 | 5.11 | 12.14 | 7.66 | 3.77 |
| Qwen2.5-32B-Inst | 36.96 | 39.04 | 6.52 | 51.15 | 7.79 | 58.45 | 14.86 |
| Qwen2.5-72B-Inst | 9.73 | 3.35 | 0.00 | 29.19 | 0.49 | 34.67 | 3.34 |

Table 17: Area captured by each router for each small model, between the small model's own random and ideal routing curves. Values of different routers across the same small model are comparable.

