# OpenReview forum: "Beyond Naïve Prompting: Strategies for Improved Context-aided Forecasting with LLMs"
_TMLR — Decision pending for TMLR_

### Review · Reviewer_krMV · 2026-04-06

**Summary Of Contributions:**

This paper examines four strategies to improve LLM-based context-aided time series forecasting: FxDP (forecast effect explanation) for improved diagnostics,  CorDP (forecast correction) and IC-DP (in-context example) for higher accuracy, and RouteDP (model routing) to balance performance and cost. These approaches are evaluated on the Context-Is-Key benchmark, consisting of 71 tasks where text-based context, in addition to historical numerical data, is critical to produce accurate predictions. Each approach is evaluated across various language models of different sizes.

**Audience:**

Yes

**Audience Explanation:**

The paper proposes approaches to improve LLM-based context-aided time-series forecasting (across a few axes). It also systematically compares a variety of language models.

**Claims And Evidence:**

Yes

**Claims Explanation:**

Mostly, with a few aspects that should be clarified.

Overall, the claim about using "the latest frontier models" is not quite true. For example, Opus 4.5 and gemini 3 are more recent than the models used in the paper.

Starting with FxDP (Direct Prompting with Forecast Effect Explanation), the authors show that some LLMs can often generate correct explanations. There can be an execution gap where the explanation is correct, but the final results aren't. One unclear aspect is whether this gap is caused by the model not applying the explanation, or whether it doesn't interpret the historical numeric data correctly.

For CorDP (Direct Prompting for Forecast Correction), the authors show that the approach can be effective in some scenarios, although it is constrained by the quality of the original predictions (e.g. with ARIMA). Besides the "Total Wins", the impact of this approach might be clearer with a side-by-side comparison of each column (in Table 1) with the direct prompting baseline. The authors show two different correction mechanism (median vs sample-wise), but it is unclear a priori which one would be effective for a task. Ideally, these variants should be combined into a single solution that consistently provides strong results.

For IC-DP (In-Context Direct Prompting) , the paper shows improved performance for 14 models out of 16, often by a significant margin. These results are based on single-shot learning. It could be interesting to examine how performance scales with additional examples. IC-DP also assumes that "the structural relationship be-tween context and forecast remains identical". It would be interesting to see how well the approach generalizes if the example is slightly out-of-domain.

Finally, for RouteDP (Direct Prompting with Model Routing), the proposed approach can consistently beat random routing, although there is still a gap towards the optimal solution. The paper convincingly shows the approach can be effective for relative rankings across many tasks. However, if there were much fewer tasks, the usefulness of the absolute scores is less clear.

As general remarks, the evaluation of combined approaches is fairly limited in the main paper (although there is a substantial appendix). While potentially acceptable, using a single benchmark limits the potential confidence in the robustness of the results.

**Requested Changes:**

(strengthen) p3 "Gruver et al. (2024) propose LLMTime, prompting an LLM to autoregressively generate each timestep; they show surprising performance versus dataset-specific time series models." How is it surprising?

(strengthen) p4 Even if it is a mostly standard metric, define CRPS more clearly.

(critical) p4 Stating the you are using the last frontier models is not quite exact. Please be more precise. Maybe they were when you started writing the paper, but model availability changes quickly.

(strengthen) p5 "Context-aided forecasting requires models to perform two sequential tasks: first, correctly reasoning how the context would affect the forecast, and second, producing an accurate quantitative forecast that applies said effects." Without FxDP, the first step can be done implicitly, correct?

(strengthen) For FxDP, are there any cases with explanations judged as incorrect, but with a correct response?

(strengthen) Can you explain in more detail the execution gap? Does the model apply the text-based context incorrectly, or makes mistakes related to the historical data?

(strengthen) Does providing explicit explanation only improve explainability, or can it also improve correctness by guiding the model reasoning?

(critical) p8 For CorDP methods, show side-by-side comparison with baseline instead of only total wins.

(strengthen) Can the two CorDP approaches be combined, or can we predict beforehand which one is more likely to succeed?

(strengthen) p10 If you have 71 tasks, how do you round/truncate to pick 20% increments?

(strengthen) For routing, if you didn't have multiple tasks to establish relative rankings, how would you decide which model to pick for a single task?

---

> ### Author Response · Authors · 2026-05-03
> **Rebuttal by the Authors**
>
> Dear Reviewer krMV,
>
> Thank you for the thorough and constructive review and for recognizing the strengths of our work. We  address each of your concerns point by point below; all corresponding updates in the paper appear in blue text. For a complete overview of changes, please see the "Changes since the last submission" section at the top of the submission.
>
> > (strengthen) p3 "Gruver et al. (2024) propose LLMTime, prompting an LLM to autoregressively generate each timestep; they show surprising performance versus dataset-specific time series models." How is it surprising?
>
> * Thank you for the question. We use the word "surprising" deliberately, mirroring the language of Gruver et al. \[1\] themselves. Their abstract states: "we find that large language models (LLMs) such as GPT-3 and LLaMA-2 can surprisingly zero-shot extrapolate time series at a level comparable to or exceeding the performance of purpose-built time series models trained on the downstream tasks" \[1\]
> * This is because LLMTime \[1\] was the first work to this result which was non-obvious at the time. Therefore, we adopt the same language to orient readers toward the origins of this research direction on LLMs for forecasting.
> * As in Paragraph 3, we already follow this statement with more recent works that have explored this direction in detail, giving readers the full trajectory of the literature.
> * To make this clearer, we have updated the sentence in the related work to read:
>   > "Gruver et al. (2024) propose LLMTime, the first work to show that LLMs can zero-shot extrapolate time series autoregressively; their then-surprising finding that LLMs match or exceed purpose-built forecasting models motivated a growing body of follow-up work, which we describe below. "
>
>
>   \[1\] Gruver, Nate, Marc Finzi, Shikai Qiu, and Andrew G. Wilson. "Large language models are zero-shot time series forecasters." *Advances in neural information processing systems* 36 (2023): 19622-19635.
>
>
> > (strengthen) p4 Even if it is a mostly standard metric, define CRPS more clearly.
>
> * Thank you for the suggestion. We have added the following definition of the CRPS metric to Appendix A.1, where the definition of the RCRPS metric is provided, and added a brief inline definition in the main text with an explicit pointer to the appendix for the full details.
>
>   >> Given a univariate forecast $\widetilde{X}$ and a ground-truth realization $x$, the Continuous Ranked  Probability Score (CRPS) can be defined in its integral as follows:
>   >>
>   >> $$\text{CRPS}(\widetilde{X}, x) = \int_{-\infty}^{\infty} dy \left[ \Phi_{\widetilde{X}}(y) - \mathbb{1}(y \ge x) \right]^2$$
>   >>
>   >> where $\Phi_{\widetilde{X}}(y)$ is the Cumulative Distribution Function of $\widetilde{X}$, and  $\mathbb{1}$ is the indicator function.
>
> > (critical) p4 Stating the you are using the last frontier models is not quite exact. Please be more precise. Maybe they were when you started writing the paper, but model availability changes quickly.
>
> * Thank you for this observation. The frontier models evaluated in this work - GPT-5.2, Gemini-2.5-Pro, and Claude-Sonnet-4.5, were the latest available at the time of writing (Feb 2026).
> * We have updated all relevant references to make this explicit, e.g. "frontier models available at the time of writing".

---

> ### Author Response · Authors · 2026-05-03
>
> > (strengthen) p5 "Context-aided forecasting requires models to perform two sequential tasks: first, correctly reasoning how the context would affect the forecast, and second, producing an accurate quantitative forecast that applies said effects." Without FxDP, the first step can be done implicitly, correct?
>
> * Thank you for the question. Yes, this is correct. Without FxDP, the model performs both steps implicitly in a single forward pass \- the contextual reasoning is internalized and never surfaced.
> * FxDP explicitly decouples these two steps by requiring the model to first verbalize how the context affects the forecast before producing the numerical output. This is analogous to the distinction between implicit and explicit reasoning in LLMs more broadly: chain-of-thought prompting \[1\] has shown that making intermediate reasoning steps explicit can both improve performance and enable diagnosis of failure modes. FxDP applies this principle specifically to context-aided forecasting, where the two steps: 1\) reasoning about forecast effects and 2\) applying them quantitatively \- are conceptually distinct and, as our results show, can fail independently. This decomposition is precisely what enables the discovery of the Execution Gap: a failure mode that is invisible when reasoning remains implicit.
> * We have made this explicit in Section 4.1 of the revised version, noting that without FxDP, both steps are performed implicitly in a single forward pass, and that FxDP externalizes this process to make intermediate failures diagnosable.
>
> \[1\] Wei, Jason, Xuezhi Wang, Dale Schuurmans, Maarten Bosma, Fei Xia, Ed Chi, Quoc V. Le, and Denny Zhou. "Chain-of-thought prompting elicits reasoning in large language models." *Advances in neural information processing systems* 35 (2022): 24824-24837.
>
> > (strengthen) For FxDP, are there any cases with explanations judged as incorrect, but with a correct response?
>
> * We thank the reviewer for the question. We find no instances of incorrect explanations leading to correct (improved) forecasts. This shows that all explanations marked as “incorrect” by the LLM judges are reliable. This is also confirmed by our human evaluation results of the LLM judge decisions (Appendix C.3.4): human annotators and LLM judges reach unanimous agreement on all incorrect explanations.
> * We have made this explicit in Sec 4.3 of the revised version.
>
> > (strengthen) Can you explain in more detail the execution gap? Does the model apply the text-based context incorrectly, or makes mistakes related to the historical data?
>
>
> * We thank the reviewer for the question. We show that the Execution Gap arises purely from the model's failure to apply its contextual reasoning to the forecast \- not from errors in interpreting the historical data.
> * This can be inferred directly from the results of the LLMs without-context given in Table 3 (App. C.1.): when models are given only the historical data and no textual context, they produce reasonable quantitative forecasts. For example, frontier models (Claude-Sonnet-4.5: 0.509, GPT-5.2: 0.504, Gemini-2.5-Pro: 0.457) all produce results in the same range as dedicated quantitative forecasters (Chronos-Large: 0.492). This demonstrates that their numerical forecasting capability i.e., their ability to interpret and extrapolate from historical data - is already present.
> * The Execution Gap therefore measures the improvement over this no-context forecast when context is provided, isolating the ability to apply contextual reasoning as the sole variable.
> * We have made this distinction explicit in Sec 4.3 of the revised version.

---

> ### Author Response · Authors · 2026-05-03
>
> > (strengthen) Does providing explicit explanation only improve explainability, or can it also improve correctness by guiding the model reasoning?
>
> * Thank you for the question. We find that LLMs providing explicit explanations does not meaningfully change forecasting accuracy. The aggregate RCRPS results of FxDP are within the standard error of DP across all evaluated models, confirming that FxDP's value lies in diagnosis rather than accuracy improvement. Representative results are shown below:
>
>   | Model | DP | FxDP |
>   |---|---|---|
>   | Llama-3.2-3B | 0.687 ± 0.025 | 0.694 ± 0.029 |
>   | Qwen-2.5-3B | 0.424 ± 0.017 | 0.431 ± 0.011 |
>   | Qwen-2.5-7B | 0.401 ± 0.006 | 0.404 ± 0.001 |
>   | Qwen-2.5-14B | 0.247 ± 0.006 | 0.250 ± 0.002 |
>   | Qwen-2.5-32B | 0.397 ± 0.008 | 0.401 ± 0.001 |
>   | Llama-3.3-70B | 0.230 ± 0.006 | 0.228 ± 0.009 |
>   | Qwen-2.5-72B | 0.202 ± 0.009 | 0.207 ± 0.007 |
>   | Llama-3.1-405B | 0.173 ± 0.003 | 0.175 ± 0.005 |
>   | GPT-5.2 | 0.271 ± 0.001 | 0.272 ± 0.001 |
>   | Gemini-2.5-Pro | 0.108 ± 0.002 | 0.109 ± 0.004 |
>   | Claude-Sonnet-4.5 | 0.114 ± 0.001 | 0.114 ± 0.008 |
>
> * This is consistent with the chain-of-thought literature, where reasoning traces do not always translate to accuracy gains without additional training [1].
> * We have added this table to Appendix C.4. in the revised version, confirming that FxDP results lie within the standard error of DP across all evaluated models.
>
> [1] Wei, Jason, Xuezhi Wang, Dale Schuurmans, Maarten Bosma, Fei Xia, Ed Chi, Quoc V. Le, and Denny Zhou. "Chain-of-thought prompting elicits reasoning in large language models." Advances in neural information processing systems 35 (2022): 24824-24837.
>
>
> > (critical) p8 For CorDP methods, show side-by-side comparison with baseline instead of only total wins.
>
> * Thank you for the suggestion. We wish to clarify that the original table already enables side-by-side comparison: DP appears as the first column and serves as the baseline against which each CorDP variant (columns 2–7) can be directly read off. We chose this layout in the main text for conciseness, as it also allows us to highlight the best method per LLM and to show that the CorDP variant and choice of base forecaster matters. The total-wins summary is provided solely as an additional aggregate view of the results.
> * That said, we provide dedicated comparison tables as requested. Due to the increased number of columns, we split by variant type for readability \- one table for Median-CorDP and one for SampleWise-CorDP \- with DP shown alongside each variant for direct comparison. We have added these tables to Appendix D.2. (Tables 11 and 12\) in the revised version, and have added a pointer to them in the main text in Sec 5.3.

---

> ### Author Response · Authors · 2026-05-03
>
> > (strengthen) Can the two CorDP approaches be combined, or can we predict beforehand which one is more likely to succeed?
>
> * Thank you for the questions. The two variants cannot be directly combined, as they are fundamentally different in how they handle forecast distributions (Sec 5.2). Median-CorDP discards the original distribution entirely \- it uses only the base forecast median as a starting point and draws new samples from the LLM to form a corrected distribution. SampleWise-CorDP, by contrast,  preserves the original probabilistic structure by correcting each individual sample from the base forecast. The two variants are instead complementary by design, each suited to different task types, as discussed in Sec 5.3.
> * On predicting which variant is more likely to succeed: the choice depends weakly on the LLM \- among the 13 models where CorDP outperforms DP,  SampleWise-CorDP is best for 7 models and Median-CorDP for the remaining 6, with no clear LLM-level predictor. However,  task type is a stronger and more actionable signal (Sec 5.3 \- paragraph titled “Performance varies by task type”) - SampleWise-CorDP excels on partial-RoI tasks, while Median-CorDP dominates full-shape and constrained tasks. This task-type specialization provides practitioners with a principled guide for variant selection.
> * We have made this distinction and the task-type guidance explicit in Sections 5.2 and 5.3 of the revised version of the paper.
>
> > (strengthen) p10 If you have 71 tasks, how do you round/truncate to pick 20% increments?
>
>
> * Thank you for the question. The reported percentages are nominal labels for readability; the actual number of tasks routed is obtained by flooring (e.g., 20% of 71 tasks gives 14 tasks, 40% gives 28, and so on). We note that Appendix F.2 already provides the full routing curve without any rounding, so the coarse increments in the main text table are purely a presentational choice for conciseness.
> * We have made this rounding convention clearer in the caption of Table 2\.
>
> > (strengthen) For routing, if you didn't have multiple tasks to establish relative rankings, how would you decide which model to pick for a single task?
>
> * We thank the reviewer for the question. As stated in Section 7.2, the router outputs a continuous numerical difficulty score between 0 (easiest) and 1 (hardest) for each task, so per-task routing is already supported: if we do not have multiple tasks to establish relative ranking, a threshold $\\tau$ can be chosen such that any task scoring above $\\tau$ is routed to the large model. This per-task setting and our batch setting are equivalent \- they route the same tasks and achieve the same RCRPS: the "fraction of tasks routed" in Table 2 simply corresponds to exploring different percentile-based thresholds (20%, 40%, 60%, 80% quantiles of the router's score distribution).
>
>
> * We explore this further and find that score ranges vary meaningfully across router sizes: larger routers (e.g., Qwen2.5-32B: \[0.03, 0.39\], Qwen2.5-72B: \[0.05, 0.95\]) spread scores across their range, while smaller routers collapse tasks into a narrow band (e.g., Qwen2.5-0.5B: \[0.76, 0.84\]). Once a router is chosen, calibrating $\\tau$ would require only a small held-out set of 5–10 tasks to estimate the relevant percentile of that router's score distribution \- a lightweight, one-time step identical to standard practice in confidence-based routing \[1, 2\].
>
> * We have updated the paper to reflect this: Section 7.2 now clarifies that our setup supports both batch and per-task routing, and a new Appendix F.2 provides a dedicated discussion with score range analysis across router sizes (Figure 48), showing that both settings yield equivalent results.
>
> \[1\] Chen, Lingjiao, Matei Zaharia, and James Zou. "FrugalGPT: How to Use Large Language Models While Reducing Cost and Improving Performance." *Transactions on Machine Learning Research*.
>
> \[2\] Ong, Isaac, Amjad Almahairi, Vincent Wu, Wei-Lin Chiang, Tianhao Wu, Joseph E. Gonzalez, M. Waleed Kadous, and Ion Stoica. "RouteLLM: Learning to Route LLMs from Preference Data." In *The Thirteenth International Conference on Learning Representations*.
>
>
> ---
> We hope these revisions fully address the raised concerns and that the updated paper provides a clearer and more complete picture of our contributions; we welcome any further questions.

---

### Review · Reviewer_6yCZ · 2026-04-13

**Summary Of Contributions:**

This paper tries to improve context-aided time series forecasting with LLMs. In detailed, the authors propose four strategies that extend the baseline Direct Prompting method along three orthogonal dimensions:
1) FxDP: it aims to elicit forecast effect explanations before predictions as the diagnostics, revealing an "Execution Gap" where models correctly reason about context effects but fail to apply them quantitatively.
2) CorDP: uses LLMs as correctors of existing quantitative forecasts, with Median and SampleWise variants.
3) IC-DP: provides one in-context example of a similar context-aided task, yielding 14–56% improvements for small models.
4) RouteDP: routes difficult tasks to the LLMs with large parameters while keeping easy tasks on a small model.

The author provides the comprehensive evaluation results, which span Qwen-family, Llama-family, and closed-source models on the Context-is-Key (CiK) benchmark.

Strengths:
1. The motivation is clear. Using LLM for time-series forecasting is a kind of challenge. The author aims to incorporate contextual information into the LLM-based time series forecasting process in a more reasonable manner. The proposed three dimensions (diagnostics, accuracy, efficiency) are orthogonal, and the paper demonstrates that they can be combined by a router.

2. The FxDP diagnostic, revealing that models can explain context effects correctly yet fail to apply them quantitatively, is an actionable insight.

3. The idea of using LLMs to correct existing quantitative forecasts is interesting. The finding that small models with CorDP can outperform their DP baselines is of practical value.

Weaknesses:
1. Although the experimental design and evaluation are very comprehensive, the proposed methods are primarily limited to prompt engineering, which may be considered insufficient given the current state of the field in 2026. As a result, the main contribution of this work appears to lie more in its systematic empirical evaluation than in methodological innovation.

2. The router is evaluated in a batch setting, so that it assumes all tasks may be known upfront. In real-world settings, the router makes per-task decisions without knowing the full distribution. Also, the binary "easy/hard" classification with constrained decoding is simplistic; a regression-based difficulty score might perform better, but no alternatives are explored or discussed.

3. Table 1 shows that CorDP performance varies depending on the base forecaster, but the paper provides limited discussion on how to select the right base forecaster.

**Audience:**

Yes

**Audience Explanation:**

Time series forecasting by using LLM is an important topic for the current research community. This paper shares several insights and practical suggestions/experiences.

**Broader Impact Concerns:**

No ethical concers.

**Claims And Evidence:**

Yes

**Claims Explanation:**

The experiment design is comprehensive. First, the paper works on recently larger benchmarks. Then, the paper evaluates 17 models, three base forecasters for CorDP, seven router configurations, and provides extensive ablations. Next, the scale of experimentation is substantial to the proposed claims and conclusions.

**Requested Changes:**

1. See in Weaknesses.

2. Figure 3 is kind of difficult to read and follow.

---

> ### Author Response · Authors · 2026-05-03
> **Rebuttal by the Authors**
>
> Dear Reviewer 6yCZ,
>
> Thank you for the thorough and constructive review, and for recognizing the strengths of our work. We address each of your concerns point by point below; all corresponding updates in the paper appear in blue text. For a complete overview of changes, please see the "Changes since the last submission" section at the top of the submission.
>
> > Although the experimental design and evaluation are very comprehensive, the proposed methods are primarily limited to prompt engineering, which may be considered insufficient given the current state of the field in 2026. As a result, the main contribution of this work appears to lie more in its systematic empirical evaluation than in methodological innovation.
>
>
> * We thank the reviewer for noting that the experimental design and evaluation are very comprehensive. We would like to highlight that prompting-based methods are still high relevant for this setting of context-aided forecasting: Specifically, Zhang et al. \[1\] include the paradigm of prompting-based MMTS (multimodal time series forecasting) methods in their taxonomy, and show that prompting-based methods such as the studied Direct Prompt method, perform strongly in this setting. Even more recently, Zheng et al. \[2\] benchmark several LLMs with the Direct Prompt method on their own context-aided forecasting datasets, and show that certain LLMs (GPT-5.2) with this method can get better performance than dedicated, trained multimodal time series forecasting methods. Our methods push the capabilities of the Direct Prompt method \- make them diagnosable, better in performance and more efficient.
> * We would also like to note that our work goes beyond prompt engineering for performance: we introduce a diagnostic framework (FxDP) backed by a human study, revealing the Execution Gap as a novel finding about model behavior, and an efficiency framework (RouteDP) for cost-effective deployment. Our contributions thus span diagnostics, and efficiency, going beyond the typical focus of prompt engineering on accuracy improvements.
> * Automatic prompt optimization \- as explored through frameworks such as APE \[4\], OPRO \[5\], and DSPy \[6\], is orthogonal to our methods and could be layered on top; we leave this to future research, including its emerging applications in self-evolving time series agents \[7\].
> * Beyond prompt engineering, recent advances in agentic methods (LLMs equipped with tool use) can also be highly relevant in this setting. Prompting-based methods such as direct prompt and our proposed improvements serve as strong baselines as shown in \[1, 2\], and our methods establish stronger, more principled baselines, setting higher standards for the setting and allowing better development of future methods.
> * To make this explicit in the paper, in the revised version, we have updated Section 9 (Discussion and Future Work) to further emphasize the connection to agentic methods and the role of our methods as principled baselines for future work.
>
>
> \[1\] Zhang, Xiyuan, Boran Han, Haoyang Fang, Abdul Fatir Ansari, Shuai Zhang, Danielle C. Maddix, Cuixiong Hu et al. "When Does Multimodality Lead to Better Time Series Forecasting?." *arXiv preprint arXiv:2506.21611* (2025).
>
> \[2\] Zheng, Vincent Zhihao, Étienne Marcotte, Arjun Ashok, Andrew Robert Williams, Lijun Sun, Alexandre Drouin, and Valentina Zantedeschi. "Overcoming the Modality Gap in Context-Aided Forecasting." *arXiv preprint arXiv:2603.12451* (2026).
>
> \[3\] Ramnath, Kiran, Kang Zhou, Sheng Guan, Soumya Smruti Mishra, Xuan Qi, Zhengyuan Shen, Shuai Wang et al. "A systematic survey of automatic prompt optimization techniques." In *Proceedings of the 2025 Conference on Empirical Methods in Natural Language Processing*, pp. 33066-33098. 2025\.
>
> \[4\] Zhou, Yongchao, Andrei Ioan Muresanu, Ziwen Han, Keiran Paster, Silviu Pitis, Harris Chan, and Jimmy Ba. "Large language models are human-level prompt engineers." In *The eleventh international conference on learning representations*. 2022\.
>
> \[5\] Yang, Chengrun, Xuezhi Wang, Yifeng Lu, Hanxiao Liu, Quoc V. Le, Denny Zhou, and Xinyun Chen. "Large language models as optimizers." In *The Twelfth International Conference on Learning Representations*. 2023\.
>
> \[6\] Khattab, Omar, Arnav Singhvi, Paridhi Maheshwari, Zhiyuan Zhang, Keshav Santhanam, Sri Vardhamanan, Saiful Haq et al. "Dspy: Compiling declarative language model calls into self-improving pipelines." *arXiv preprint arXiv:2310.03714* (2023).
>
> \[7\] Xu, Longkun, Xiaochun Zhang, Qiantu Tuo, and Rui Li. "SEA-TS: Self-Evolving Agent for Autonomous Code Generation of Time Series Forecasting Algorithms." *arXiv preprint arXiv:2603.04873* (2026).

---

> ### Author Response · Authors · 2026-05-03
>
> > The router is evaluated in a batch setting, so that it assumes all tasks may be known upfront. In real-world settings, the router makes per-task decisions without knowing the full distribution. Also, the binary "easy/hard" classification with constrained decoding is simplistic; a regression-based difficulty score might perform better, but no alternatives are explored or discussed.
>
> * We thank the reviewer for the suggestion.
> * **On regression-based difficulty scores:** As stated in Section 7.2, the router already outputs a continuous numerical difficulty score between 0 (easiest) and 1 (hardest) for each task \- this is precisely the regression-based scoring the reviewer suggests.
> * **On batch vs. per-task routing:** We agree that real-world deployment requires per-task routing decisions. We would like to clarify that our setup already supports this: since scores are numerical, an appropriate threshold $\\tau$ can be chosen such that any task scoring above $\\tau$ is routed to the large model.
>   * Crucially, this per-task threshold setting and our batch setting are equivalent \- they route the same tasks and achieve the same RCRPS: the "fraction of tasks routed" reported in Table 2 simply corresponds to exploring different percentile-based thresholds (20%, 40%, 60%, 80% quantiles of each router's score distribution), making the results directly transferable to the per-task setting.
>
> * We explore this further and find that score ranges vary meaningfully across router sizes: larger routers spread scores across their range (e.g., Qwen2.5-32B: \[0.03, 0.39\], Qwen2.5-72B: \[0.05, 0.95\]), while smaller routers collapse tasks into a narrow band (e.g., Qwen2.5-0.5B: \[0.76, 0.84\]). Once a router is chosen, calibrating $\\tau$ would require only a small held-out set of tasks to estimate the relevant percentile of that router's score distribution \- a lightweight, one-time step identical to standard practice in confidence-based routing \[1, 2\].
>
>
> * We have updated the paper to reflect this: Section 7.2 now clarifies that our setup supports both batch and per-task routing, and a new Appendix F.2 provides a dedicated discussion with score range analysis across router sizes (Figure F.2), showing that both settings yield equivalent results.
>
> \[1\] Chen, Lingjiao, Matei Zaharia, and James Zou. "FrugalGPT: How to Use Large Language Models While Reducing Cost and Improving Performance." *Transactions on Machine Learning Research*.
>
> \[2\] Ong, Isaac, Amjad Almahairi, Vincent Wu, Wei-Lin Chiang, Tianhao Wu, Joseph E. Gonzalez, M. Waleed Kadous, and Ion Stoica. "RouteLLM: Learning to Route LLMs from Preference Data." In *The Thirteenth International Conference on Learning Representations*.
>
> > Table 1 shows that CorDP performance varies depending on the base forecaster, but the paper provides limited discussion on how to select the right base forecaster.
>
> * We thank the reviewer for the question. As we discuss in the "SampleWise vs Median variants" paragraph of Sec 5.3, the performance of CorDP methods does differ depending on the base forecaster used; however, all winning methods except one use Lag-Llama as the base forecaster. This is because Lag-Llama is the best performing quantitative forecaster on this data, as shown in the base forecaster row of Table 1\.
> * This points to a clear and practical selection criterion: to choose the best available quantitative forecaster as the base model for CorDP. Crucially, this selection does not require any labeled context-aided forecasting data \- one can simply evaluate candidate base forecasters on standard held-out no-context data (a routine step in any forecasting pipeline) and use the best performer. This is also practically convenient: in real deployments, practitioners typically already have a quantitative forecasting model in place, and CorDP is designed to augment it with contextual reasoning rather than replace it. We show that LLMs with CorDP are flexible enough to use forecasts from any base model and incorporate context on top.
> * We have made this selection guideline explicit in Section 5.3 in the revised version of the paper.

---

> > ### Author Response · Authors · 2026-05-03
> >
> > > Figure 3 is kind of difficult to read and follow
> >
> > * We thank the reviewer for the feedback. We present an improved version of Figure 3 in the revised version of the paper.
> >
> > * The key changes are: the Execution Gap segment is now highlighted in orange (previously gray) to draw attention to the central finding, and bracket annotations below the x-axis explicitly group models and label the two key results. The two key results are now made explicit:
> >
> >    \- Smaller models fail to produce accurate explanations of how the context would affect the forecast
> >
> >    \- Larger models can explain forecast effects correctly, but the Execution Gap \- in applying them to produce accurate forecasts \- still persists.
> >
> > ---
> > We hope these revisions fully address the raised concerns and that the updated paper provides a clearer and more complete picture of our contributions; we welcome any further questions.

---

### Review · Reviewer_mE34 · 2026-04-21

**Summary Of Contributions:**

The paper studies the problem of forecasting with contexts where one aims to make a time-series prediction with the presence of additional information/context in the format of natural language descriptions. The paper proposes several prompting-based strategy to resolve the problem, each with a different emphasis. The paper illustrates the performance of the proposed methods on the CiK benchmark.

**Audience:**

Yes

**Audience Explanation:**

The studied problem is an important one -- like how we can incorporate additional text information in time series prediction.

**Broader Impact Concerns:**

Nan.

**Claims And Evidence:**

Yes

**Claims Explanation:**

The paper is well-written and easy to follow.

**Requested Changes:**

My main concerns are the following three aspects:

- As I understand, the time-series prediction here solely relies on the LLM with no access to tool use such as Python. The LLM indeed is a good model to utilize the context information; but it is not a good or the best time series prediction model. So the prediction generated from the LLM is more likely a qualitative than quantitative one. To this aspect, I appreciate the paper's work on thoroughly studying the setup and the proposed methods, but I personally don't think this is the right approach for tackling the problem.

- The numerical experiment is just on one data data CiK. It's unclear how robust the numerical results are, like whether the conclusions and insights drawn from the paper apply to other datasets or scenarios.

- Will the performance be improved if we can fine-tune the model?

---

> ### Author Response · Authors · 2026-05-03
> **Rebuttal by the Authors**
>
> Dear Reviewer mE34,
>
> Thank you for recognizing the paper as well-written and easy to follow, and for confirming that the claims are well-supported by convincing evidence. We address each of your concerns point by point below; all corresponding updates in the paper appear in blue text. For a complete overview of changes, please see the "Changes since the last submission" section at the top of the submission.
>
> > As I understand, the time-series prediction here solely relies on the LLM with no access to tool use such as Python. The LLM indeed is a good model to utilize the context information; but it is not a good or the best time series prediction model. So the prediction generated from the LLM is more likely a qualitative than quantitative one. To this aspect, I appreciate the paper's work on thoroughly studying the setup and the proposed methods, but I personally don't think this is the right approach for tackling the problem.
>
> * Thank you for highlighting our thorough study of the setup and the proposed methods. We would like to highlight that the studied Direct Prompt method, which simply prompts LLMs with the context and history and obtains a structured forecast, has been shown to be a strong method for context-aided forecasting in multiple other studies as well \[1, 2\]. Specifically, Zhang et al. \[1\] show that it performs strongly in this setting of context-aided forecasting, and more importantly, that the improvement from larger LLMs stems from enhanced numerical processing capabilities. Even more recently, Zheng et al. \[2\] benchmark several LLMs with the Direct Prompt method on their own context-aided forecasting datasets, and show that certain LLMs (GPT-5.2) with this method can get better performance than dedicated, trained multimodal time series forecasting methods. These results reinforce that prompting-based methods such as Direct Prompt make strong models for context-aided forecasting. Our methods push the capabilities of the Direct Prompt method \- make them diagnosable, better in performance and more efficient.
> * Nevertheless, we agree with you that recent developments in tool-use agents can motivate better methods that rely on strong forecasting models as tools. We would like to note that the paper already proposes and evaluates a strategy, CorDP (Direct Prompting for Forecast Correction) where a specialized quantitative model (Lag-Llama, Chronos, ARIMA) handles the numerical forecasting (i.e. without textual context), and the LLM is used solely for reasoning over the context and “correct” the forecast given the textual context. This mirrors what a tool-use agent would do in this setting \- call a quantitative forecasting tool, obtain its forecast, and apply contextual reasoning to refine it and produce a corrected forecast. CorDP evaluates this pipeline by design, and our results show it to be a strong and practical approach.
> * We believe prompting-based methods represent a necessary and complementary step toward agentic forecasting: without strong prompting baselines, it is unclear what tool-use agents need to improve upon, and our methods establish exactly this \- stronger, more principled baselines that make clear where prompting succeeds and where it falls short. We believe more studies are necessary to develop and improve methods that are fundamentally agentic for context-aided forecasting.
> * We already discuss this aspect in the Discussion and Future Work section (Sec 9), and have updated it in the revised version to highlight the above-mentioned connection of CorDP to agentic methods.
>
> \[1\] Zhang, Xiyuan, Boran Han, Haoyang Fang, Abdul Fatir Ansari, Shuai Zhang, Danielle C. Maddix, Cuixiong Hu et al. "When Does Multimodality Lead to Better Time Series Forecasting?." *arXiv preprint arXiv:2506.21611* (2025).
>
> \[2\] Zheng, Vincent Zhihao, Étienne Marcotte, Arjun Ashok, Andrew Robert Williams, Lijun Sun, Alexandre Drouin, and Valentina Zantedeschi. "Overcoming the Modality Gap in Context-Aided Forecasting." *arXiv preprint arXiv:2603.12451* (2026).

---

> ### Author Response · Authors · 2026-05-03
>
> > The numerical experiment is just on one data data CiK. It's unclear how robust the numerical results are, like whether the conclusions and insights drawn from the paper apply to other datasets or scenarios.
>
> * We thank the reviewer for the question. The reason for benchmarking on CiK is not arbitrary. First, it is the only benchmark where accurate forecasts cannot be achieved without incorporating the context, making it uniquely suitable for evaluating context-aided forecasting capabilities \[1\]. Second, it is the only benchmark specifically designed for zero-shot, prompt-based multimodal forecasting \- the setting our methods target, where the focus is on how well models can use unambiguous, relevant context to succeed in forecasting scenarios. Other benchmarks \[2, 3, 4, 5\] do not guarantee that context is essential for high-quality forecasts, clouding evaluation as shown in \[1\], or are proposed for a different setting of training-based multimodal forecasting \[1\]. This controlled design is also a strength: by guaranteeing that context is always necessary and unambiguous, CiK provides a clean signal on whether methods can effectively use context; a noisier benchmark would obscure the phenomena our paper studies.
> * As we state in the discussion section, evaluating the proposed strategies in more unconstrained zero-shot context-aided forecasting settings, where context may be irrelevant or excessively long, is an interesting direction, however it requires developing appropriate benchmarks first. We have added a clarifying sentence to Section 9 to make this scope explicit.
> * Finally, as in the supplementary material, we make the codebase of methods as accessible as possible, to enable benchmarking them seamlessly as appropriate benchmarks in this space are developed.
>
> \[1\] Zhang, Xiyuan, Boran Han, Haoyang Fang, Abdul Fatir Ansari, Shuai Zhang, Danielle C. Maddix, Cuixiong Hu et al. "When Does Multimodality Lead to Better Time Series Forecasting?." *arXiv preprint arXiv:2506.21611* (2025).
>
> \[2\] Mike A Merrill, Mingtian Tan, Vinayak Gupta, Tom Hartvigsen, and Tim Althoff. Language models still struggle to zero-shot reason about time series. EMNLP 2024
>
> \[3\] Haoxin Liu, Shangqing Xu, Zhiyuan Zhao, Lingkai Kong, Harshavardhan Kamarthi, Aditya B Sasanur, Megha Sharma, Jiaming Cui, Qingsong Wen, Chao Zhang, et al. Time-MMD: A new multi-domain multimodal dataset for time series analysis. NeurIPS 2024
>
> \[4\] Chengsen Wang, Qi Qi, Jingyu Wang, Haifeng Sun, Zirui Zhuang, Jinming Wu, Lei Zhang, and Jianxin Liao. Chattime: A unified multimodal time series foundation model bridging numerical and textual data. AAAI 2025
>
> \[5\] Chengsen Wang, Qi Qi, Zhongwen Rao, Lujia Pan, Jingyu Wang, and Jianxin Liao. Chronosteer: Bridging large language model and time series foundation model via synthetic data. arXiv preprint arXiv:2505.10083, 2025\.
>
> > Will the performance be improved if we can fine-tune the model?
>
> * We thank the reviewer for the question. We agree that there is strong potential for performance to be improved when models are fine-tuned with a specific method (DP, CorDP, IC-DP, or RouteDP) on a specific dataset. Importantly, our results already provide concrete guidance on where fine-tuning would help most: FxDP reveals that small models primarily fail at the reasoning stage (inaccurate explanations), while larger models fail at the execution stage (accurate explanations but failed forecasts) \- giving a clear target for what fine-tuning should address for each model class.
> * More broadly, moving beyond zero-shot and few-shot inference paradigm to the finetuning paradigm could even expand the scope of all four strategies. For instance, it could enable trained routers in RouteDP to improve routing performance, or allow CorDP to fit to the dynamics of a specific base forecasting model. Our methods open the door for all these improvements through fine-tuning.
> * We believe this is a natural and exciting direction that our methods explicitly enable, and we discuss it in the Discussion and Future Work section (Section 9), and we have emphasized these connections further in the revised version.
>
> ---
>
> We hope these revisions fully address the raised concerns and that the updated paper provides a clearer and more complete picture of our contributions; we welcome any further questions.

---

### Decision · Action_Editor_KJUn · 2026-06-17

**Recommendation:** Accept as is

**Audience:**

Yes

**Audience Explanation:**

Reviewers were unanimous on this front. They emphasized the importance of time series forecasting in the greater research landscape right now, and I believe the depth of discussion between authors and reviewers emphasizes, on a kind of meta-review level, that there really is substantial interest here.

**Claims And Evidence:**

Yes

**Claims Explanation:**

Most of the discussion provided by the reviewers (and backed up by author discussion) emphasizes the "comprehensive" (according to one reviewer) and and has actually since been expanded by the authors to include more recent models. While one review had some reservations about whether the direct prompt method was the "right" method writ large to this type of problem, the authors provided a quite nice response to the questions raised there. Moreover, the other reviewers did find the insights from the supported by a quite substantial body of evidence.